# Oligomerised RIPK1 is the main core component of the CD95 necrosome

Nikita V Ivanisenko[1,4], Corinna König[1,4], Laura K Hillert-Richter[1,4], Maria A Feoktistova[2], Sabine Pietkiewicz[1], Max Richter[1], Diana Panayotova-Dimitrova [ID][2], Thilo Kaehne[3] & Inna N Lavrik [ID][1][✉]

## Abstract

**The necrosome is the key macromolecular signaling platform initiating necroptosis, i.e., a RIPK1/RIPK3-dependent program of cell death with an important role in the control of inflammation in multicellular organisms. However, the composition and structure of the necrosome remain incompletely understood. Here we use biochemical assays, quantitative mass spectrometry, and AlphaFold modeling to decipher the composition and derive a structural model of the CD95L/BV6-induced necrosome. We identify RIPK1 as the central component of the necrosome, forming the core of this complex. In addition, AlphaFold modeling provides insights into the structural mechanisms underlying RIPK1 oligomerization, highlighting the critical role of type-II interactions between the Death Domains (DDs) of FADD and RIPK1 in the assembly of RIPK1-mediated complexes. The role of type-II DD interactions in necroptosis induction is further validated through structure-guided site-directed mutagenesis. Our findings could be useful for the pharmacological targeting of the necroptosis network to treat diseases associated with dysregulated cell death and inflammation.**

**Keywords** CD95; Necrosome; Mass Spectrometry; Death Domain; AlphaFold
**Subject Categories** Autophagy & Cell Death; Immunology; Structural Biology

## Introduction

Death receptor (DR) activation leads to the induction of cell death pathways. Apoptosis is one of the best-studied programs of cell death (Krammer et al, 2007). The apoptotic DR signaling cascade is triggered by one of the death ligands (DL), (TNF, CD95L/FasL, TRAIL) which binds to its cognate DR (Lemke et al, 2014). All members of the DR family possess the death domain (DD) (Wilson et al, 2009). The DD comprises six interfaces that can interact with each other via so-called type-I, -II and -III interactions formed by Ia/Ib, IIa/IIb, and IIIa/IIIb interfaces of DD.

CD95/Fas is one of the best-investigated DRs (Lavrik and Krammer, 2012; Nagata, 1999). Stimulation of CD95 by CD95L results in the formation of a death-inducing signaling complex (DISC), which comprises CD95 and Death Effector Domain (DED)—containing proteins (denoted hereafter as DED proteins): FADD, procaspase-8a/b, procaspase-10 as well as c-FLIP$_{L/S/R}$ (Lavrik and Krammer, 2012; Sprick et al, 2002). Procaspase-8a/b is activated via dimerization at the DISC in DED filaments formed via DED interactions (Dickens et al, 2012; Fu et al, 2016; Schleich et al, 2012). Activation of caspase-8 at the DED filaments leads to induction of the caspase cascade followed by the demolition of the cell.

Other platforms mediating procaspase-8 activation besides CD95 DISC have been reported (Lavrik et al, 2008). One of them is the complex IIa or ripoptosome, which is formed upon deubiquitinylation of the DD-containing kinase RIPK1, the latter is mediated via deubiquitinases or SMAC mimetics such as BV6. It comprises the RIPK1, FADD, procaspase-8/-10, and c-FLIP$_{L/S/R}$ proteins (Feoktistova et al, 2011; Geserick et al, 2009; Tenev et al, 2011). RIPK1, FADD, procaspase-8/-10 and c-FLIP$_{L/S/R}$ also associate into macromolecular complexes to trigger other responses than apoptosis such as induction of NF-κB (Hartwig et al, 2017; Henry and Martin, 2017) or sensing proliferation-associated DNA damage (Boege et al, 2017). Thus, it seems that the same proteins RIPK1, FADD, procaspase-8/-10 and c-FLIP form macromolecular complexes that, depending on their composition stoichiometry, are engaged in different functions. Therefore, understanding the stoichiometry of macromolecular complexes is key to uncovering their function.

Stimulation of CD95 upon inhibition of caspases and deubiquitinylation of RIPK1 might induce another program of cell death, necroptosis (Feoktistova et al, 2011; Geserick et al, 2009; Grootjans et al, 2017; Pasparakis and Vandenabeele, 2015). Necroptosis is mediated via the RIP kinase family members RIPK1 and RIPK3 that contain RIP homotypic interacting motifs (RHIMs) (Mompean et al, 2018; Nailwal and Chan, 2019; Silke et al, 2015; Wegner et al, 2017; Weinlich et al, 2017). The necrosome or complex IIb serves as a platform to induce the necroptotic signaling cascade (Vanden Berghe et al, 2016; Vandenabeele et al, 2010). The necrosome

[1]Translational Inflammation Research, Medical Faculty, Otto von Guericke University, Magdeburg, Germany. [2]Department of Dermatology and Allergology, University Hospital RWTH Aachen, Aachen, Germany. [3]Institute of Internal Experimental Medicine, Medical Faculty, Otto von Guericke University, Magdeburg, Germany. [4]These authors contributed equally: Nikita V Ivanisenko, Corinna König, Laura K Hillert-Richter. ✉E-mail: inna.lavrik@med.ovgu.de

comprises the kinases RIPK1 and RIPK3; DED proteins: FADD, procaspase-8/-10 as well as c-FLIP. The assembly of the necrosome leads to phosphorylation of RIPK1/RIPK3, RIPK3-mediated phosphorylation and activation of the pseudokinase MLKL driving necroptotic cell death via pore formation at the extracellular membranes (Alvarez-Diaz et al, 2016; Czabotar and Murphy, 2015; Davies et al, 2024; Li et al, 2012; Meng et al, 2021; Meng et al, 2023; Newton et al, 2019; Samson et al, 2020; Sun et al, 2012; Wang et al, 2014; Yuan and Ofengeim, 2024).

Activation of macromolecular complexes driving cell death and inflammation is the crucial step triggering these programs. Despite apparent progress in understanding necroptotic platforms, a detailed molecular mechanism of their activation remains elusive. One of the reasons is that information on the molecular architecture and stoichiometry of these complexes is still largely absent. In this regard, an enormous asset is presented by the novel technologies of quantitative mass spectrometry, which allow the determination of the composition stoichiometry of the macro-molecular complexes (Dickens et al, 2012; Schleich et al, 2012). The other key technologies for solving the structural composition of protein platforms are continuously derived from the ongoing Artificial Intelligence (AI) revolution in protein structure prediction and the development of the high accuracy models using AlphaFold, including its later versions AlphaFold2 and 3 (Abramson et al, 2024; Jumper et al, 2021; Varadi et al, 2022).

In this study, we aimed to investigate the stoichiometry and structural features of CD95L/BV6-induced necrosome. After establishing a biochemical approach to immunoprecipitate necrosome and performing quantitative mass spectrometry analysis, we used AI-based AlphaFold2 and AlphaFold3 modeling to validate the results and select the possible stoichiometries of the complex. Our results indicate that the major necrosome component is RIPK1, which forms the core of the complex, and reveal the other features of necrosome assembly. In particular, AlphaFold modeling in combination with experimental validation demonstrated that FADD DD/RIPK1 DD type-II interactions play a pivotal role in the assembly of this complex.

# Results

## CD95L/BV6/zVAD-fmk co-treatment induces necroptosis in sensitive lines

To analyze the necrosome assembly and composition, we have selected the following cell lines: colorectal carcinoma HT29, pancreatic carcinoma SUIT-020 and T-cell lymphoma Jurkat cells. To test necroptosis induction, these three cell lines were first treated with 5 μM SMAC mimetic BV6 and 50 μM pan-caspase inhibitor zVAD-fmk for one hour followed by 500 ng/ml CD95L stimulation, which is hereafter referred to as CD95L/BV6/zVAD-fmk co-treatment. These stimulation conditions were shown to trigger necroptosis in RIPK3-containing cells (Feoktistova et al, 2011; Geserick et al, 2009). Indeed, CD95L/BV6/zVAD-fmk co-treatment efficiently induced RIPK1, RIPK3 and MLKL phosphorylations (Figs. 1A–C and EV1A,B), which are the hallmarks of necroptosis induction (Liu et al, 2018). The phosphorylated form of RIPK1 (pRIPK1) was detected after two hours of CD95/BV6/zVAD-fmk stimulation in SUIT-020 cells and after three hours in HT29 and Jurkat 282 cells (Fig. 1A–C). The

phosphorylation of RIPK1 was blocked upon addition of Necrostatin-1s (Nec-1s) (Fig. EV1A,B), which is the well-established inhibitor of RIPK1 and necroptosis (Degterev et al, 2013). Along with pRIPK1, the phosphorylated forms of RIPK3 (pRIPK3) and MLKL (pMLKL) were detected upon CD95L/BV6/zVAD-fmk co-treatment in these cell lines. However, the signals for pRIPK3 were rather weak due to the reagents not yet being optimized (Figs. 1A and EV1A,B) (Samson et al, 2021), which is currently being improved (Chiou et al, 2024; Petrie et al, 2019). An increase in pMLKL was more clearly detected in these cell lines upon CD95L/BV6/zVAD-fmk co-treatment (Figs. 1A,C and EV1A,B).

In line with the phosphorylation of RIPK1, RIPK3 and MLKL, we have observed the loss of cell viability upon CD95L/BV6/zVAD-fmk co-stimulation, which was blocked by Nec-1s or the RIPK3 inhibitor GSK872 co-treatment in all three cell lines supporting necroptosis induction under these conditions (Figs. EV1C–H and EV2A–C). Caspase-8 is a negative regulator of necroptosis. Consistent with this, caspase-8-deficient Jurkat A3 cells (Jurkat C8 ko) were more efficiently protected by Nec-1s from CD95L-only or CD95L/BV6-only treatments than Jurkat 282 cells, indicating CD95-mediated necroptosis induction in Jurkat C8 ko cells in the absence of zVAD-fmk treatment (Fig. EV1E–H).

In line with the loss of cell viability, HT29 and Jurkat 282 cells underwent necroptotic cell death upon CD95L/BV6/zVAD-fmk co-treatment (Fig. 1D–G). The latter was measured by imaging flow cytometry, which allows for distinguishing between necroptotic and apoptotic cell death, as previously reported (Pietkiewicz et al, 2015b). Indeed, upon CD95L/BV6/zVAD-fmk stimulation, a high number of cells with the typical necroptotic morphology was observed in the bright field channel (Fig. 1H,I). These cells were swollen and had an enlarged nucleus, which are well-established features of necroptosis. At the same time, these cells were Annexin-V-FITC (AVF)/propidium iodide (PI) positive, as measured using AVF/PI staining. The amount of double-positive cells was increased in a time-dependent manner after CD95L/BV6/zVAD-fmk co-treatment in both HT29 and Jurkat 282 cells (Fig. 1D–G). CD95L/BV6/zVAD-fmk-induced cell death was blocked by the addition of Nec-1s, further supporting the induction of necroptosis in these cell lines (Fig. 1D–G). Moreover, dying cells were not detected in RIPK1-deficient Jurkat A3 cells (Jurkat RIPK1 ko) upon CD95L/BV6/zVAD-fmk co-treatment, which supports that this pathway is mediated via RIPK1 (Fig. EV2D,E). Taken together, these experiments demonstrated that CD95L/BV6/zVAD-fmk co-treatment induces necroptosis in the selected cell lines and that these cells can be used for the analysis of necrosome composition.

## CD95L/BV6/zVAD-fmk stimulation of HT29, SUIT-020 and Jurkat cells results in the necrosome assembly

To initiate the necrosome assembly, the investigated cell lines were treated with CD95L/BV6/zVAD-fmk (Figs. 2 and 3). For each cell line, we selected the time interval, corresponding to the initial detection of pRIPK1 in total cellular lysates, assuming that these conditions would correspond to the assembly of the active necrosome complex, e.g. shortly after activation of RIPK1. After co-treatment with CD95L/BV6/zVAD-fmk, the necrosome was immunoprecipitated using anti-pRIPK1 (Figs. 2A–C,E,F and 3D), anti-RIPK1 (Fig. 2D), anti-FADD (Fig. 3A,B), anti-FLIP (Fig. 3C) or anti-caspase-8 (Fig. 3D) antibodies. Indeed, pRIPK1 was

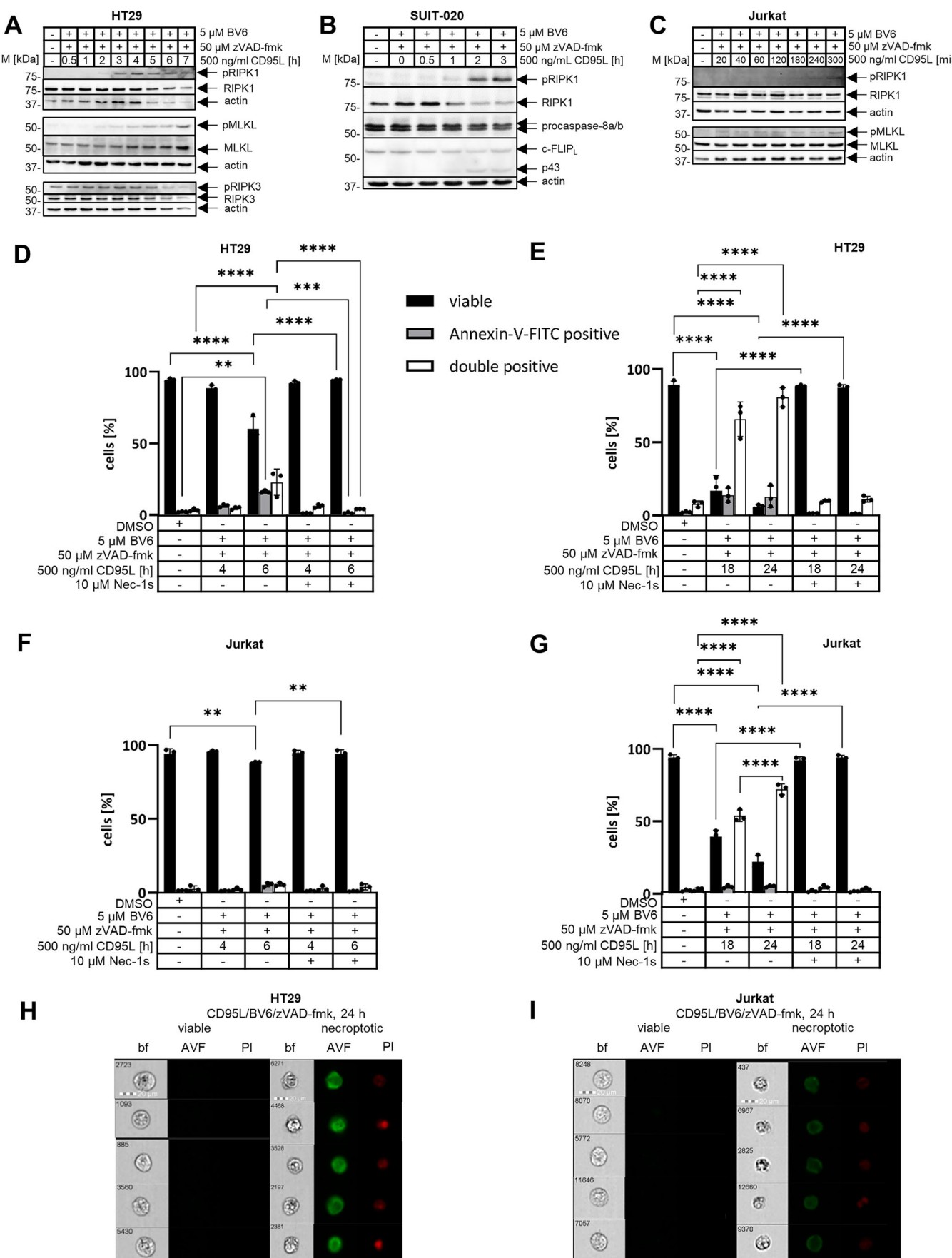

◄ **Figure 1.  CD95L/BV6/zVAD-fmk co-treatment induces necroptosis in sensitive cells.**

(A–C) HT29 (A), SUIT-020 (B) or Jurkat 282 (C) cells were pretreated for 1 h with 5 µM BV6 and 50 µM zVAD-fmk and subsequently treated with 500 ng/ml CD95L for the indicated timepoints. Total cellular lysates were analyzed using western blot with the indicated antibodies. Actin served as loading control. One representative experiment out of three is shown. (D–G) HT29 (D, E) and Jurkat 282 (F, G) cells were treated for 1 h with 5 µM BV6, 50 µM zVAD-fmk and 10 µM Nec-1s and subsequently treated with 500 ng/ml CD95L for indicated time intervals. Cells were analyzed via imaging flow cytometry. Populations were gated for viable (negative), Annexin-V-FITC positive and double-positive (Annexin-V-FITC and PI positive) cells. The mean and standard deviation from three independent experiments are shown. (H, I) Representative pictures from imaging flow cytometry for HT29 (H) and Jurkat 282 (I) cells with viable (negative) and double-positive (Annexin-V-FITC and PI positive) cells. Pictures are taken from treated cells after 24 h. Statistics were calculated with unpaired one-way ANOVA with Tukey post hoc test to compare two conditions. Significance values: ****$P < 0.0001$; ***$P < 0.001$; **$P < 0.01$; *$P < 0.05$; ns not significant. $P$ values from left to right for (D) $P < 0.0001$, $P = 0.0011$, $P < 0.0001$, $P < 0.0001$, $P = 0.0005$, $P < 0.0001$ (E) all $P < 0.0001$ (F) $P = 0.0056$, $P = 0.0060$ (G) all $P < 0.0001$. bf bright field, AVF Annexin-V-FITC, PI Propidium Iodide. Source data are available online for this figure.

detected in these immunoprecipitations (IPs), indicating the assembly of the active necrosome complex in these experiments. Besides pRIPK1, the other core components of the necrosome such as RIPK1, FADD, procaspases-8, -10 and c-FLIP were detected in these IPs (Figs. 2 and 3). Importantly, the addition of Nec-1, which blocks phosphorylation of pRIPK1, resulted in inhibition of necrosome co-IP with anti-pRIPK1 antibodies (Fig. 2B). However, Nec-1 treatment did not inhibit necrosome assembly as assessed by the increase in the amount of core complex components immunoprecipitated with anti-c-FLIP and anti-FADD antibodies (Fig. 3A,C). This further supports that phosphorylation of RIPK1 upon CD95/BV6/zVAD stimulation takes place directly in the necrosome upon assembly of the complex. Due to the aforementioned lack of optimized antibodies, RIPK3 was detected at very low levels, if at all, in these experiments. Eventually, anti-RIPK3 antibodies, which were used, showed the specificity for RIPK3 detection in the analysis of cellular lysates (Fig. EV2F). However, these antibodies showed a low sensitivity for RIPK3 detection in the IPs, which may be due to overlap of the anti-RIPK3 antibodies's epitope with the RIPK3 domains interacting with the other proteins in the necrosome as well as the overlap of the signal with the heavy chain of the immunoprecipitating antibodies. The latter was probably also a reason for the poor detection of MLKL in these IPs, as the signal for pMLKL and MLKL is likely to overlap with a signal from the antibody's heavy chain.

In Jurkat C8 ko cells slightly higher levels of the necrosome assembly were observed in comparison to Jurkat 282 cells as judged by higher levels of RIPK1 and FADD in pRIPK1-IP from Jurkat C8 ko cells (Fig. 2E). This suggests that the low levels of procaspase-8 activity that might be present in the necrosome of Jurkat 282 cells could lead to cleavage of RIPK1 and RIPK3 leading to down-regulation of the necrosome complex supporting previous observations (Oberst et al, 2011; Tummers et al, 2020). Consistent with the absence of cell death in CD95L/BV6/zVAD-fmk-treated Jurkat RIPK1 ko cells, no necrosome assembly was observed in this cell line (Fig. 2F).

The selected cell lines were previously shown to undergo both necroptosis and apoptosis (Hillert-Richter et al, 2024; Konig et al, 2025; Pietkiewicz et al, 2015a). Specifically, upon stimulation with CD95L alone, they formed a CD95 DISC containing all core components including CD95, FADD, procaspase-8a/b, c-FLIP$_{L/S}$ as well as their cleavage products as was shown for SUIT-020 cells (Fig. EV2G). Strikingly, in these experiments, RIPK1 was found to associate with CD95 in a stimulation-independent manner, albeit at low levels. This suggests the possibility of stimulation-independent weak interactions of FADD and CD95 DDs.

MLKL-, pRIPK3- and RIPK3-IPs also led to the pulldown of some core necrosome components from HT29, Jurkat 282 and SUIT-020 cells upon CD95L/BV6/zVAD-fmk stimulation (Appendix Figs. S1 and S2). However, these IPs were less efficient compared to pRIPK1-IP. The latter may be caused by overlapping interactions between necrosome core components and antibody binding sites, as well as the lack of optimized antibodies for RIPK3, as pointed out by others (Samson et al, 2021). In addition, MLKL-IP may largely immunoprecipitate oligomerised MLKL at the later stages of necroptosis, when it is already released from the necrosome complex and therefore this particular IP may not result in the efficient pulldown of the core necrosome components (Garnish et al, 2021; Meng et al, 2021; Meng et al, 2023).

Furthermore, we hypothesized that among the antibodies used for IPs, anti-pRIPK1 antibodies would immunoprecipitate the necrosome complex most efficiently and specifically, since phosphorylation of RIPK1 occurs in the necrosome and pRIPK1 is likely to remain in this complex. We also hypothesized that this would not be the case for FADD, RIPK1, and caspase-8, which would not necessarily be expected to be found only in the necrosome complex upon CD95L/BV6/zVAD-fmk stimulation, but could also be present in the cell without being involved in interactions within macromolecular complexes. To further support the hypothesis that pRIPK1 remains in the necrosome after phosphorylation, we performed immunoprecipitation after gel filtration. Necrosome has been reported to form a high molecular weight (HMW) platform (Feoktistova et al, 2011; Tenev et al, 2011). CD95/BV6/zVAD-fmk-induced macromolecular complexes were purified by gel filtration followed by IPs from the HMW fractions using the antibodies against the core components of the necrosome, procaspase-8, FADD, RIPK1, and pRIPK1 (Figs. 3E and EV3A–C). The core components of the necrosome were immunoprecipitated from HMW fractions (fractions 2–6), indicating the assembly of HMW-necrosome complexes (Fig. 3E and EV3A–C). In particular, procaspases-8/-10, c-FLIP$_{L/S}$, RIPK1 and FADD were detected in pRIPK1-IP from HMW fractions (Fig. 3E). Importantly, none of the core necrosome components were co-immunoprecipitated from the low molecular weight (LMW) fractions using pRIPK1 antibodies (Fig. 3E). Even pRIPK1 was not found in these IPs that were performed from LMW fractions. This further supports that pRIPK1 can be largely found in the HMW fractions, e.g. in the active necrosome complexes. The latter further supports the selection of the pRIPK1 antibody as the one to pull out active necrosome complexes.

Further evidence of the efficiency of the IPs used for necrosome co-immunoprecipitation was obtained by mass spectrometry

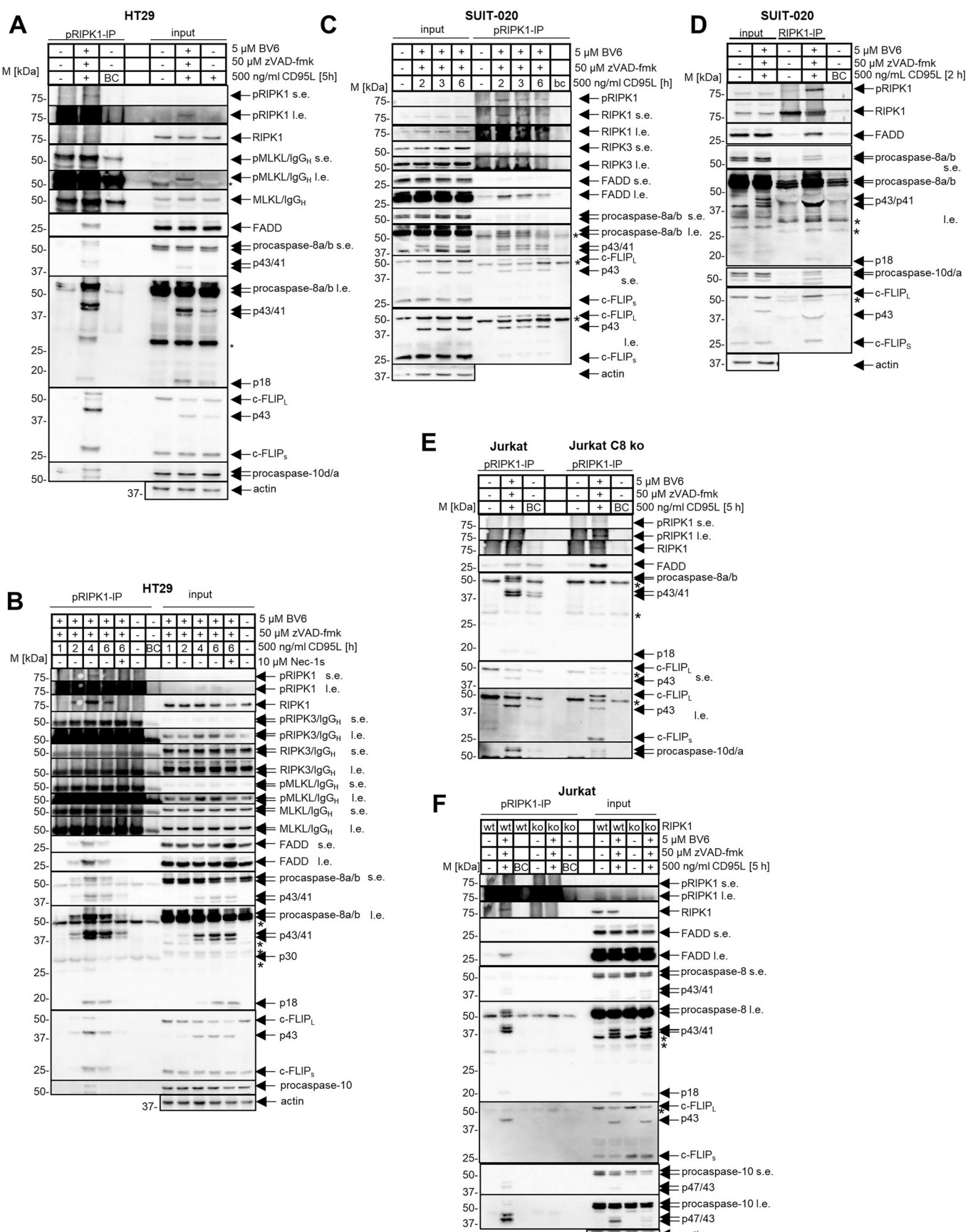

◀ **Figure 2. CD95L/BV6/zVAD-fmk stimulation of HT29, SUIT-020, and Jurkat cells leads to the necrosome assembly.**

(A–E) HT29 cells (**A, B**), SUIT-020 cells (**C, D**), Jurkat 282 and Jurkat C8 ko cells (**E**) and Jurkat A3 and Jurkat RIPK1 ko cells (**F**) were pretreated for 1 h with 5 μM BV6, 50 μM zVAD-fmk and 10 μM Nec-1s and subsequently treated with 500 ng/ml CD95L for indicated timepoints (**B, C**), 2 h (**C**) or 5 h (**A, E, F**). pRIPK1- and RIPK1-IPs were carried out using anti-pRIPK1 and anti-RIPK1 antibodies, respectively, and analyzed by western blot. Actin was used as a loading control for the total cellular lysates (input). "Beads-only" pulldown (BC) was used as a negative control for IPs. One representative experiment out of three (**A, B, D–F**) or two (**C**) is shown. s.e.-short exposure, l.e. long exposure, IP immunoprecipitation, BC beads-only control pulldown, * unspecific band, IgG$_H$ the heavy chain of antibody. Source data are available online for this figure.

analysis of the interactome in pRIPK1-, FADD- and c-FLIP-IPs (Fig. 4A). Importantly, all core components of the necrosome such as procaspase-8, c-FLIP, FADD, RIPK1 and RIPK3 were detected in these analyses upon CD95L/BV6/zVAD-fmk stimulation, further supporting the specificity of these approaches for necrosome co-immunoprecipitation (Fig. 4A). CD95 or CD95L were not detected in these IPs, suggesting that upon CD95L/BV6/zVAD-fmk co-treatment the complex II presents the most abundant complex among the CD95-induced cellular complexes (Fig. 4A). In addition to the core components of the necrosome, highly abundant and so-called "sticky" proteins were detected in this screen with the high scoring, such as heat shock proteins (HSP7C, HS90B, HS90A), chaperones (BiP), cytoskeletal proteins (ACTG, ACTB, TBB4B, TBB5, MYH9, MYH14), components of peroxiredoxin family (PRDX1), DNA repair machinery (PRKDC) and RNA binding proteins (NONO, DDX5, DDX6, DDX3X, YBOX1, MSI2H, TCOF) (Fig. 4A; Dataset EV1). Importantly, most of these proteins were found in the beads-only sample supporting their non-specific character of binding (Fig. 4A; Dataset EV1). There were also several proteins detected in the IPs upon CD95L/BV6/zVAD-fmk stimulation that were reported to play a role in cell death pathways such as TRAP1, VDAC2, HIPK2/3, BIRC6, JAK2 albeit with a very low scoring (Fig. 4A). Interestingly, some potential interactors such as ZBP1 were not detected (Imai et al, 2024). Bioinformatic GO analysis also confirmed the presence of a statistically significant group of proteins associated with cell death pathways, and specifically the necroptosis network had the high relevance in this analysis (Fig. EV3D; Dataset EV2). This further supports the specificity of this approach for immunoprecipitating the necrosome and confirms that these IPs can be used to pull down the necrosome complex.

## Quantitative mass spectrometry demonstrated that RIPK1 has a high abundance in the necrosome

To get more insights into the stoichiometry of the necrosome, we used quantitative mass spectrometry analysis. In particular, we used the absolute quantification (AQUA) peptide approach, which we have successfully applied before to unravel the stoichiometry of the CD95 DISC (Hillert et al, 2020; Schleich et al, 2016; Schleich et al, 2012; Warnken et al, 2013). The AQUA approach is based on the application of chemically synthesized "heavy" peptides containing stable isotopes ($^{13}$C, $^{15}$N) that correspond to parts of the proteins under investigation (Warnken et al, 2013). This allows determining the absolute amount of the target proteins and their stoichiometry in the investigated macromolecular complex.

We collected the evidence that CD95L/BV6/zVAD-fmk treatment leads to the assembly of the necrosome complex, which can be efficiently immunoprecipitated with anti-pRIPK1 and other

antibodies against core necrosome components. The quantitative mass spectrometry analysis of pRIPK1-IPs from SUIT-020 and HT29 cells revealed that RIPK3 is present in lower amounts compared to RIPK1 (Figs. 4B and EV3E). As discussed above, we assumed that pRIPK1 was only present in the necrosome complex, which was supported by HMW fractionation experiments. Therefore, we suggested that RIPK1 amounts obtained from pRIPK1-IPs can be considered for estimating its ratio to RIPK3. Furthermore, these results were supported by quantitative mass spectrometry analysis of FADD-, c-FLIP-, and MLKL-IPs, which also showed lower levels of RIPK3 compared to RIPK1 (Figs. 4C–E and EV3F). It should be noted that the absolute numbers of RIPK1 detected in these IPs differed between the four IPs, probably due to the unequal efficiency of the immunoprecipitating antibodies. The latter might be because some epitopes for the immunoprecipitating antibodies might be interfering with protein–protein interactions in the complex. However, in all these experiments, higher numbers of RIPK1 compared to RIPK3 were detected, strongly suggesting that RIPK1 is the major core component of the necrosome.

The other line of evidence for the low abundance of RIPK3 in the necrosome was obtained from the analysis of the interactome in pRIPK1-, FADD- and c-FLIP-IPs, in which RIPK3 was detected with a very low scoring. This indirectly supports its low abundance in the necrosome (Fig. 4A). In accordance with these observations, the analysis of the mRNA expression levels of RIPK1 and RIPK3 using The Cancer Genome Atlas (TCGA) database revealed that the mRNA of RIPK1 has significantly higher expression levels than the one of RIPK3 in most cancer tissues (Fig. EV3G). These results are consistent with the lower level of RIPK3 in most of the cancer cell lines and tissues according to the analysis of the quantitative proteomics database ProteomicsDB (Lautenbacher et al, 2022). In particular, the proteomics analysis derived from (Cao et al, 2021) shows higher levels of RIPK1 compared to RIPK3 within the Pancreatic Ductal Adenocarcinoma in both normal and tumor tissues, which is also in line with our data (Fig. EV4A).

The levels of FADD in the pRIPK1-IPs from HT29 and SUIT-020 cells were lower than those of RIPK1 (Figs. 5A, EV3E,F, and EV4B). Similar to pRIPK1-IPs the absolute numbers of FADD differed between the pRIPK1-, c-FLIP- and MLKL-IPs, probably due to the unequal efficiency of the immunoprecipitating antibodies and also due to the fact that some epitopes might be interfering with interactions of FADD with the other proteins in the complex (Fig. 5). The number of FADD molecules in FADD-IP was much higher than the amounts of other necrosome components, as anti-FADD-IP co-immunoprecipitates all cellular FADD. Hence, FADD numbers from FADD-IPs cannot be used to make quantitative assumptions. Similarly, we assumed that absolute numbers of c-FLIP and procaspase-8 from FADD- and c-FLIP-IPs could not be used for quantification (Figs. 5B,D and EV4B) because

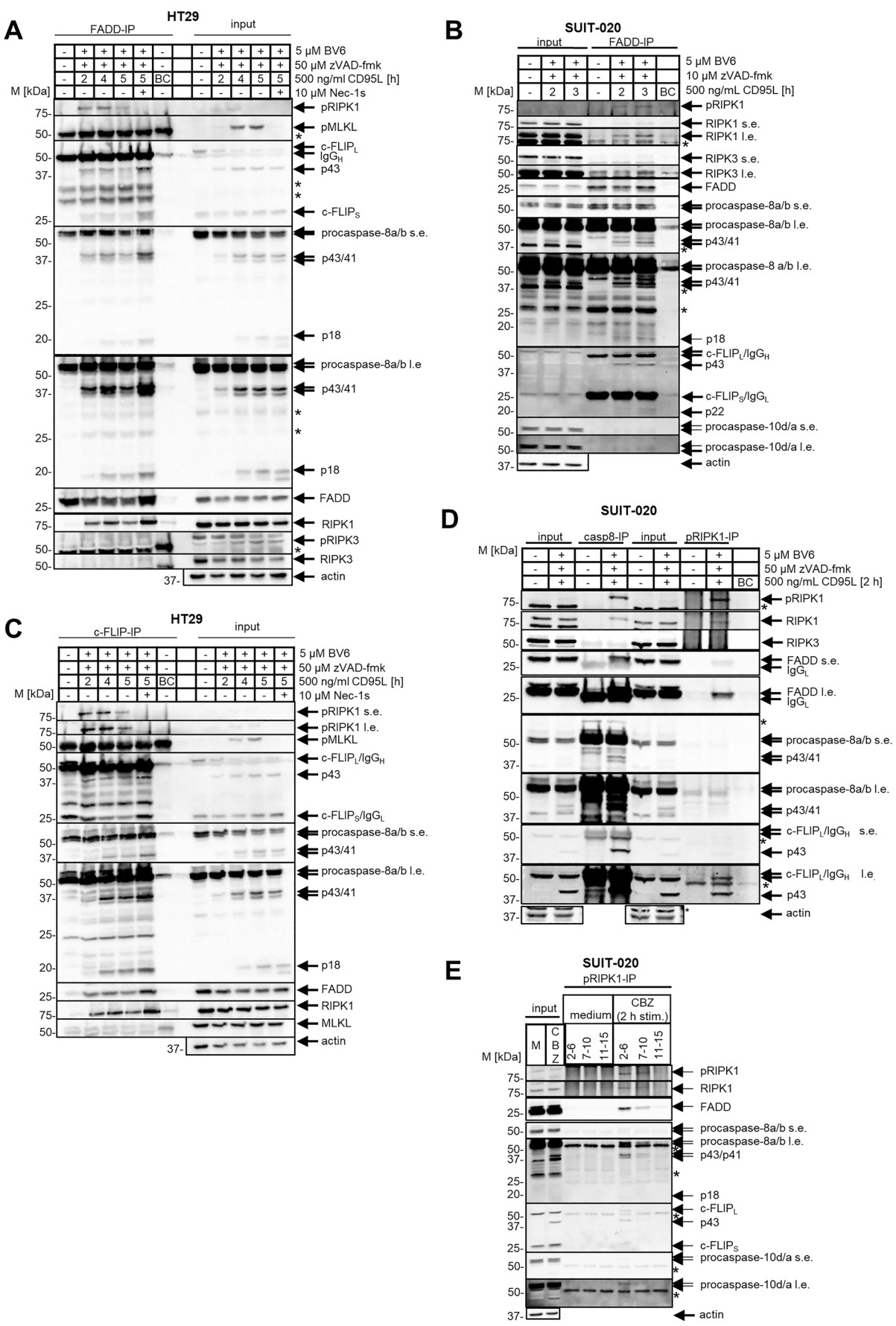

**Figure 3. CD95L/BV6/zVAD-fmk co-treatment leads to co-immunoprecipitation of the core necrosome components.**

(A–D) HT29 (A, C) or SUIT-020 cells (B, D) were pretreated for 1 h with 5 µM BV6, 50 µM zVAD-fmk and 10 µM Nec-1s and subsequently treated with 500 ng/ml CD95L for indicated timepoints (A–C) or 2 h (D). FADD-, c-FLIP-, caspase-8- and pRIPK1- IPs were carried out using anti-FADD, anti-c-FLIP, anti-caspase-8 or anti-pRIPK1 antibody, respectively, and analyzed by western blot. Actin was used as a loading control for the total cellular lysates (input). Beads-only pulldown (BC) was used as a negative control for IPs. One representative experiment out of three (A–C) or four (D) is shown. (E) SUIT-020 cells were pretreated with 5 µM BV6 and 50 µM zVAD-fmk for 1 h and afterwards stimulated with 500 ng/ml CD95L for 2 h. The total cellular lysates were fractionated by gel filtration. The different fractions were pooled together (2–6; 7–10, 11–15) followed by pRIPK1-IP. Total cell lysates and IPs were analyzed by western blot and probed for the indicated proteins. Actin was used as a loading control for total cellular lysate. One representative experiment out of three is shown. s.e. short exposure, l.e. long exposure, IP immunoprecipitation, CBZ CD95L/BV6/zVAD-fmk, BC beads-only control pulldown, * unspecific band, IgG_H the heavy chain of antibody, IgG_L the light chain of antibody. Source data are available online for this figure.

c-FLIP was used as a bait in c-FLIP-IP and these IPs may contain proteins from procaspase-8/c-FLIP or procaspase-8/c-FLIP/FADD complexes formed independently of DL stimulation (Golks et al, 2006; Yang et al, 2024).

The absolute numbers of core necrosome components in MLKL-IP were in general lower than in the other three IPs (Figs. 4D and 5C). This fits to the hypothesis that anti-MLKL antibodies may largely immunoprecipitate oligomerised MLKL at the later stages of necroptosis, when it is already released from the necrosome complex and therefore this particular IP may not result in efficient pulldown of the core necrosome complex (Garnish et al, 2021; Meng et al, 2021; Meng et al, 2023). Having obtained the information on the relative abundance of the necrosome components, the next step was to validate this information by constructing the structural model of the necrosome in silico.

## AlphaFold predicted the possible stoichiometries of DDs in the necrosome

To construct an in silico model of the necrosome, we employed advanced AI techniques, integrating high-precision AlphaFold3 and AlphaFold2-multimer modeling (Abramson et al, 2024; Jumper et al, 2021; Varadi et al, 2022). In this way, the structures of the core components of necrosome in different oligomeric states were predicted. Confidence metrics, including the ranking score and the interface predicted template modeling score (iPTM score), were implemented to evaluate and compare the accuracy of the predicted models. The ranking score is obtained as a combination of several AlphaFold-derived metrics to provide a comprehensive measure of model reliability (Abramson et al, 2024). The iPTM score specifically assesses the accuracy of protein–protein interaction binding interfaces and provides a robust measure of AlphaFold-Multimer's performance for macromolecular assembly.

The higher levels of RIPK1 compared to RIPK3 in the necrosome were detected by quantitative mass spectrometry (Figs. 4, 5, and EV4B), suggesting the formation of the RIPK1 oligomers as a major core for the necrosome complex. This prediction was validated through AlphaFold3 modeling, which indicated the potential assembly of RIPK1 homooligomers (Fig. EV4C). It was hypothesized that RIPK1 oligomers subsequently recruit RIPK3 via RHIM interactions. It should be noted that AlphaFold2 does not allow the modeling of amyloid RHIM structures and therefore only the recently released AlphaFold3, which has been implemented to model RIPK1/RIPK3 oligomers, was used for this purpose (Abramson et al, 2024).

In the necrosome besides RIPK3, RIPK1 interacts with FADD via DD interactions. Therefore, in order to determine how RIPK1 oligomerization matches the stoichiometry of the RIPK1 DD/

FADD DD part of the complex, the latter also had to be analyzed in detail. For FADD DD AlphaFold2-Multimer predicted the stoichiometry of 3× FADD as being one of the most optimal scaffolds for the necrosome assembly (Fig. 6A). Furthermore, different numbers of RIPK1 DDs forming the complex with 3× FADD DD were predicted by the AlphaFold2-Multimer ranging from 3 to 7 to 10 RIPK1 DD subunits (Fig. 6A,B). These stoichiometries corresponded to the highest iPTM scores, with values exceeding 0.75, indicating high model quality. Structural analysis suggests that these stoichiometries promote DD oligomer assembly by engaging a high number of DD interfaces. Furthermore, the high number of RIPK1 DD subunits in the necrosome, e.g. the ratio of FADD DD to RIPK1 DD, such as 3: 10 might indicate the propensity of RIPK1 to undergo homooligomerization (Fig. EV4C). Additional stoichiometries modeled with iPTM scores above 0.75 also included the 4: 9 and 2: 6 configurations of the FADD DD: RIPK1 DD subunits, suggesting even more possibilities for the assembly of the oligomeric necrosome complexes (Fig. 6A; Appendix Figs. S3–S5). This may also suggest that necrosome assembly is rather a dynamic process with several possible stoichiometries of RHIM and DD oligomers with the RIPK1 oligomer as the main scaffold. Importantly, quantitative mass spectrometry analysis supported the different ratios predicted by AlphaFold modeling, as a rather large variation in the RIPK1 DD: FADD DD ratio was observed in mass spectrometry analysis (Fig. EV4B).

However, as mentioned above, AlphaFold2-Multimer was limited in its ability to predict simultaneous oligomerization of DD and RHIM domains. To address this, AlphaFold3 was employed. It successfully reconstructed the structural models with distinct stoichiometries of DDs obtained by AlphaFold2-multimer (Fig. EV4D). Furthermore, according to AlphaFold3, RIPK1 DD/ RHIM subunits can form complexes with a maximum of approximately 10 RIPK1, enabling both DD complex formation and RHIM engagement (Fig. EV4D). This limit arises from the linker length between the RIPK1 DD C-terminus and the RIPK1-RHIM N-terminus, constraining RHIM interactions with distant RIPK1 subunits in larger oligomers (Fig. EV4D). This modeling analysis identified several necrosome stoichiometries consistent with mass spectrometry predictions. Both AlphaFold3 and AlphaFold2-Multimer screenings highlighted structural models containing 3 FADD DDs and 5–10 RIPK1 DDs as having the highest confidence metrics. These models featured extensive DD interface interactions (Figs. 6A and EV4D). We selected 7 RIPK1: 3 FADD DDs compositions, as a reference one, due to its high AlphaFold2 and AlphaFold3 metrics, as well as its correspondence to the experimentally evaluated range of ratios (Figs. 6A and EV4B,D).

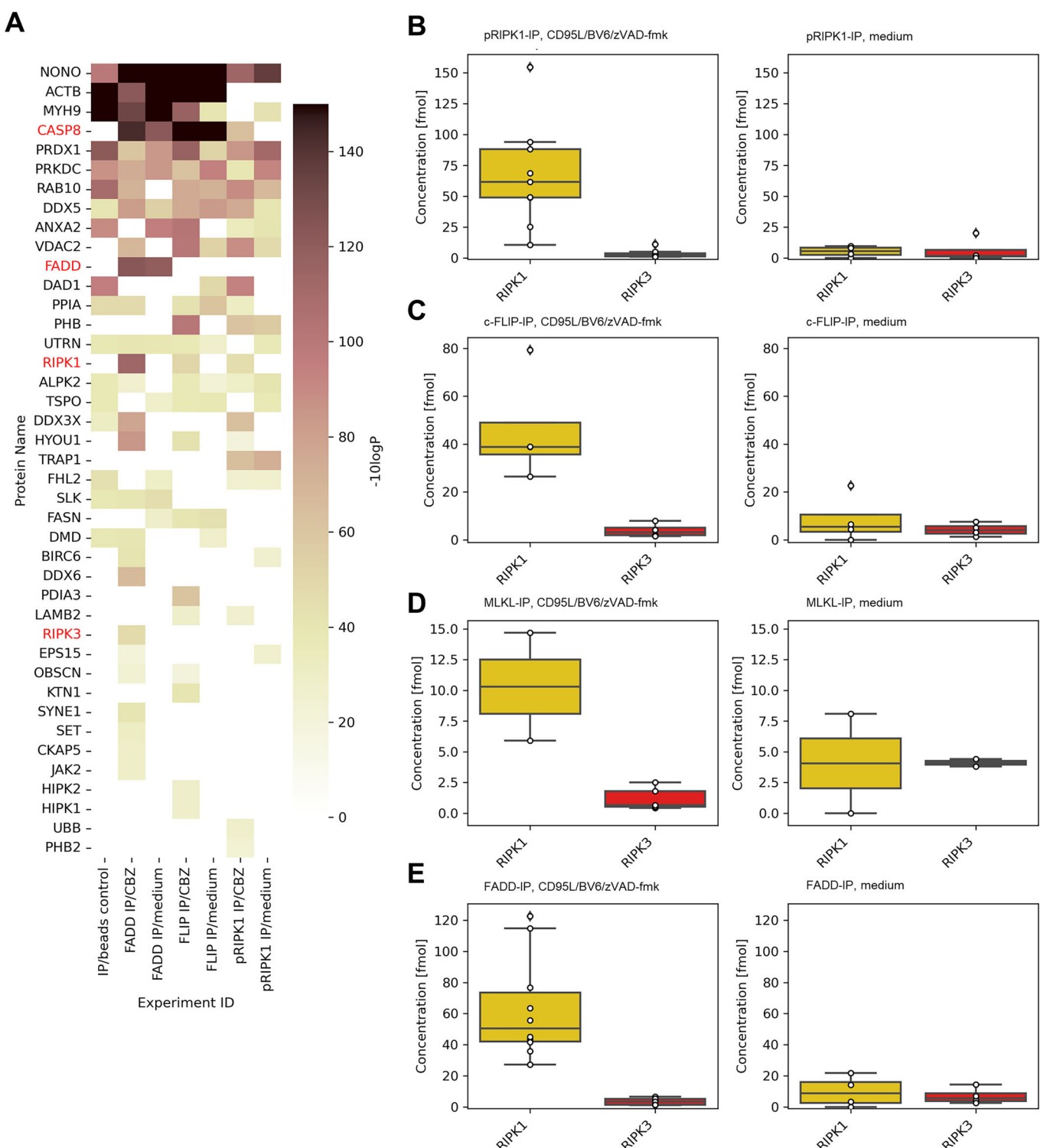

## The role of type-II DD interaction in the necrosome assembly

The predicted DD complex assembles into an oligomeric structure with helical symmetry, stabilized by three types of asymmetric interactions (type-I, type-II, and type-III) characteristic of the DD

superfamily, including the CD95 DD/FADD DD complex (Appendix Fig. S6). To get further structural insights into FADD DD/RIPK1 DD interactions in the necrosome, we analyzed in detail the models of FADD DD/RIPK1 DD with the highest AlphaFold3 and AlphaFold2-multimer confidence metrics. One of the features of this analysis was that FADD DD IIb interface formed a heterotypic

◄

**Figure 4.  Quantitative mass spectrometry demonstrated that RIPK1 has a high abundance in the necrosome.**

(A) Heatmap from FADD-IP, c-FLIP-IP and pRIPK1-IP after qualitative mass spectometry analysis. HT29 cells were pretreated for 1 h with 5 μM BV6 and 50 μM zVAD-fmk and subsequently treated with 500 ng/ml CD95L for 5 h, which was followed by IPs. CD95/BV6/zVAD-fmk (CBZ)-treated and untreated (medium) IP samples were analyzed by mass spectrometry. Representative proteins are shown for CBZ-treated cells (FADD-IP/CBZ, FLIP-IP/CBZ, pRIPK1/CBZ) and untreated cells (FADD-IP/ medium control, FLIP-IP/medium control, pRIPK-IP/medium control, and beads-only control). Necrosome components are highlighted in red color. The full list of detected proteins can be found in Dataset EV1. (B–E) HT29 cells were pretreated for 1 h with 5 μM BV6 and 50 μM zVAD-fmk and subsequently treated with 500 ng/ml CD95L for 5 h (B, C, E) or 6 h (D) or left untreated (medium). The pRIPK1-IP, c-FLIP-IP, MLKL-IP and FADD-IP were carried out using anti-pRIPK1, anti-c-FLIP, anti-MLKL and anti-FADD antibodies, respectively. The IPs were analyzed by AQUA mass spectrometry analysis. The amounts of the AQUA peptides (fmols) corresponding to each protein, that were detected in the IP, are shown mean and standard deviation from five (B, E) or two (C, D) experiments are shown. Box plots show the distribution of the data using the median (center line), interquartile range (IQR; box limits represent the first and third quartiles, Q1 and Q3) and whiskers extending from the lower and upper quartiles to 1.5× IQR. Points outside the whiskers are considered outliers and are shown as single points. IP immunoprecipitation, CBZ CD95L/BV6/zVAD-fmk.

interaction with the RIPK1 DD IIa interface within all predicted complexes (Fig. 6C; Appendix Fig. S6). To further evaluate the propensity for type-II interactions between DD interfaces within the necrosome we used the Rosetta molecular modeling software and the protocol developed by us in the previous work (Hillert et al, 2020). These predictions were in line with the AlphaFold-derived models and suggest that the heterotypic type-II interaction of the RIPK1 DD IIa and FADD DD IIb interfaces had the highest binding score among all analyzed homo- and heterotypic interactions (Appendix Figs. S6–12). According to the modeling analysis, this can be explained by the additional favorable interactions of the hydrophobic FADD DD residues with the extended H5-H6 loop of RIPK1 DD compared to the interactions of the corresponding regions of the FADD and CD95 DDs. In particular, RIPK1 DD forms a hydrophobic cluster by interactions of residues M637 and I641 with residues M170, L172 and L176 of FADD (Fig. 6C). In addition, according to the modeling, RIPK1 DD/FADD DD complex is stabilized by the hydrogen bond of the D175 side chain to the adjacent backbone nitrogen, which is similar to CD95 DD/ FADD DD complex (Fig. 6C).

We hypothesized that the high affinity between the DD interfaces of FADD IIb and RIPK1 IIa may play an important role in the assembly of RIPK1-mediated complexes. Specifically, this may provide a basis for competition between CD95 DD and RIPK1 DD for binding to FADD DD, which may be essential for DISC or necrosome assembly, respectively. Interestingly, this model suggested the possibility that RIPK1 could integrate into the oligomerized CD95/FADD complex and that RIPK1 DD could displace CD95 DD from the CD95 DD/FADD DD complex via competitive binding (Fig. 7A). This model gains support from experimental findings, particularly the observed recruitment of RIPK1 to the CD95 (EV2G).

To further validate the structural model we conducted literature and database mining to identify known mutations in RIPK1 DD associated with the cancer. Mining of the COSMIC database (Tate et al, 2019) revealed nine positions within the DD where non-synonymous mutation cases were observed (Fig. 7B). The largest set of amino acid mutations in the RIPK1 DD was located in the proximity of type-II interaction interface. It included Q663H/K, I641T, M637I, R638K/S non-synonymous substitutions (Fig. 8A). These substitutions may either directly affect the type-II DD protein–protein interaction, or affect the RIPK1 DD binding due to structural or conformational effects. Another high-frequency mutation residue position R603 (R603H, R603S, R603C) is directly involved in the formation of the salt bridge in the homotypic type-I DD interaction (Figs. 7B and 8A). Mutation R599K was reported to

block necroptosis by inhibiting RIPK1 dimerization and it is located within close proximity to R603 (Meng et al, 2018). Overall mapping of RIPK1 DD mutations in cancer further supports our structural model and the role of type-II interactions of RIPK1 DD in the necrosome assembly.

To validate the key role of type-II interactions in the necroptosis induction, several amino acid residues of RIPK1 and FADD involved in the interactions of IIa and IIb DD interfaces were mutated. In particular, RIPK1 ((M637A/I641D) termed RIPK1-AD), FADD ((L172D/D175R/L176S) termed FADD-DRS) and FADD ((M170A/L172D/D175R/L176N), termed FADD-ADRN) mutants were constructed (Fig. 6C). The mutations were designed to inhibit type-II DD protein–protein interactions as well as preserve the stability of the proteins, e.g., of FADD and RIPK1, according to estimation based on Rosetta scoring function.

RIPK1-AD was reconstituted into Jurkat RIPK1 ko (Figs. 8B,C and EV5A–C), while FADD-DRS and FADD-ADRN were introduced into FADD-deficient Jurkat (Jurkat FADD ko) (Fig. 8D). Furthermore, FADD- and RIPK1-deficient HeLa cells were also used for testing the effects of FADD-DRS, FADD-ADRN and RIPK1-AD (Figs. 9A–D and EV5D–G). Specifically, the constructed mutants were reconstituted into RIPK1-deficient RIPK3-overexpressing HeLa (HeLa RIPK1 ko RIPK3) and FADD-deficient-RIPK3-overexpressing HeLa (HeLa FADD ko RIPK3) cells (Figs. 9A–D and EV5D–G). WT-RIPK1 and WT-FADD were also reconstituted into these RIPK1- and FADD-deficient cells, respectively.

The loss of a cell viability upon CD95L/BV6/zVAD-fmk co-stimulation was observed in parental Jurkat A3 cells (Fig. 8B) as well as upon reconstitution of WT-RIPK1 and WT-FADD into RIPK1- and FADD-deficient cells (Figs. 8C,D and 9A–D). The addition of Nec-1s to these cells rescued the CD95L/BV6/zVAD-fmk-induced loss of their viability (Figs. 8B,C and 9A–D). Upon reconstitution of RIPK1-AD or FADD-DRS or FADD-ADRN into RIPK1- or FADD-deficient cells, CD95L/BV6/zVAD-fmk-induced loss of their viability was inhibited (Figs. 8C,D and 9A–D). This was observed upon both short-term and long-term CD95L/BV6/ zVAD-fmk stimulation (Fig. 9A–D). Importantly, CD95L alone stimulation led to a loss of cell viability in both Jurkat RIPK1 ko and HeLa RIPK1 ko RIPK3 cells, but not in FADD-deficient cells (Figs. 8, 9A–D, and EV5A–C). This is fully consistent with previous reports that CD95-mediated apoptotic cell death is FADD-dependent but not RIPK1-dependent (Abramson et al, 2024; Feoktistova et al, 2011; Geserick et al, 2009). Reconstitution of RIPK1-AD and FADD-DRS/FADD-ADRN into HeLa RIPK1 ko RIPK3 and HeLa FADD ko RIPK3 cells, respectively, was also

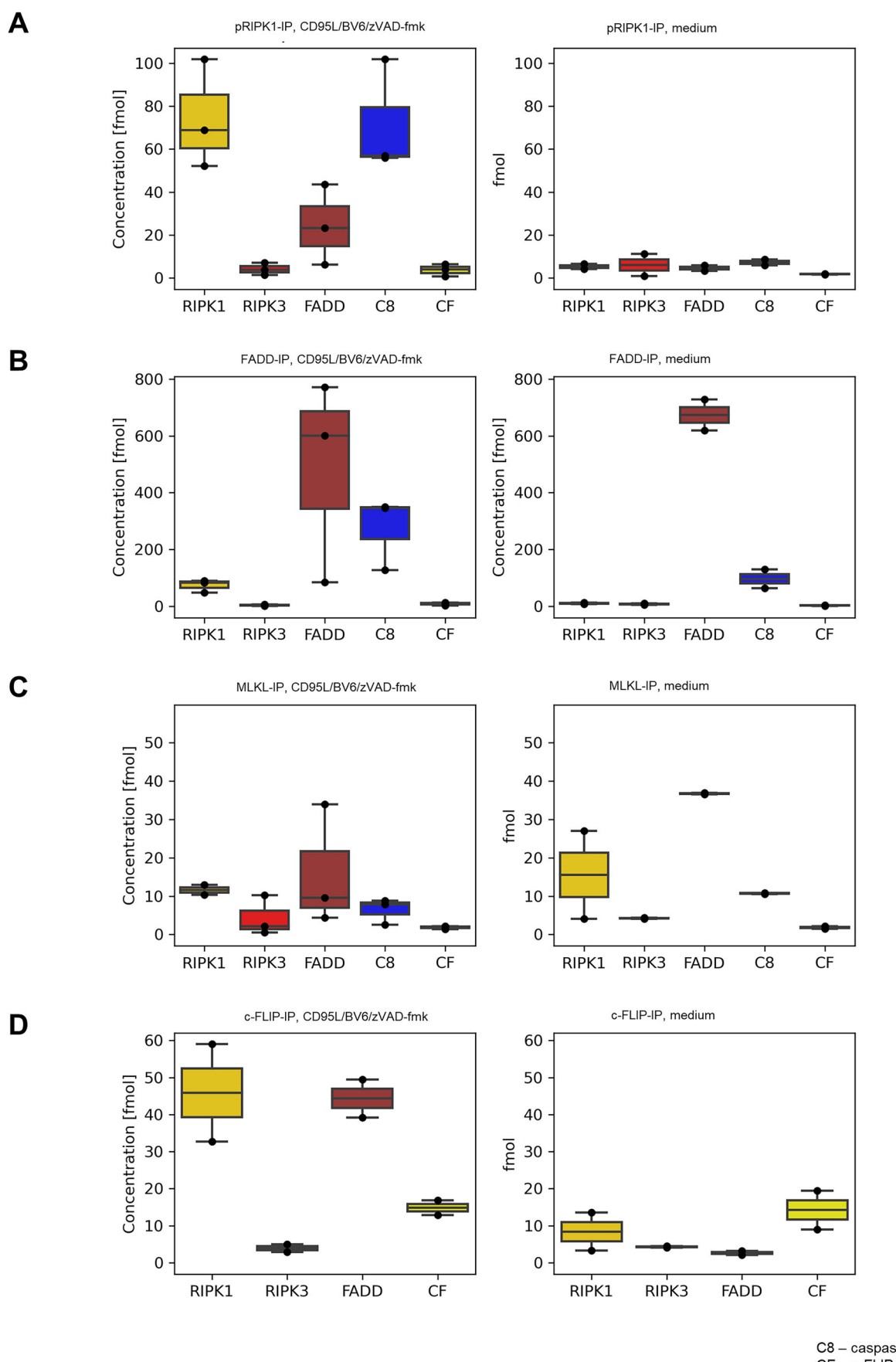

C8 – caspase-8
CF – c-FLIP

**Figure 5.** Quantitative mass spectrometry revealed the relative abundance of core proteins in the necrosome.

(A–D) HT29 cells were pretreated for 1 h with 5 µM BV6 and 50 µM zVAD-fmk and subsequently treated with 500 ng/ml CD95L for 5 h (A, B, D) or 6 h (C) or left untreated (medium). The pRIPK1-IP, c-FLIP-IP, MLKL-IP and FADD-IP were carried out using anti-pRIPK1, anti-c-FLIP, anti-MLKL, and anti-FADD antibody, respectively. The IPs were analyzed by AQUA mass spectrometry analysis. The amounts of the AQUA peptides (fmols) corresponding to each protein, that were detected in the IP, are shown. Mean and standard deviation from three (A, B) or two (C, D) experiments are shown. Box plots show the distribution of the data using the median (center line), interquartile range (IQR; box limits represent the first and third quartiles, Q1 and Q3) and whiskers extending from the lower and upper quartiles to 1.5× IQR. Points outside the whiskers are considered outliers and are shown as single points. The experiments in this Fig. are the same as those in Fig. 4, except that more of the components of the IPs are shown. IP immunoprecipitation, C8 caspase-8, CF c-FLIP.

accompanied by the reduction in phosphorylation of RIPK1 and MLKL (Fig. EV5D–G). These results strongly indicate that the introduction of mutations into RIPK1 and FADD, specifically disrupting type-II DD interactions, leads to the inhibitory effects on necrosome assembly and necroptosis induction. Taken together, the designed mutations provide support for the suggested structural model of the necrosome and support the role of type-II interactions of DDs in the necrosome assembly.

## Discussion

Necroptosis is an emerging type of cell death and its onset is determined by assembly of the complex IIb or necrosome (Geserick et al, 2009; Meng et al, 2018; Yuan and Ofengeim, 2024). In the course of this work, using biochemical analysis, quantitative mass spectrometry and state of the art computational modeling we got several insights into the necrosome assembly in CD95-mediated necroptosis. It should be noted that most of the studies so far were devoted to TNFα-induced but not to CD95-induced necroptosis, therefore, there is much less information available on CD95-mediated necroptotic pathway (Geserick et al, 2009; Grootjans et al, 2017; Silke et al, 2015).

We show that the major component of the CD95/BV6/zVAD-fmk-induced necrosome is RIPK1, which forms the core of the complex, and that RIPK1 aggregation drives necrosome activation. Importantly, AlphaFold modeling indicated that necrosome assembly is likely to be a dynamic process with multiple possible stoichiometries involving the formation of RHIM and DD oligomers. The particular stoichiometry may be determined by the strength of stimulation, the endogenous expression of key necrosome components and their post-translational modifications (de Almagro et al, 2015; Hoblos et al, 2024). This type of complex assembly will be consistent with other oligomeric structures such as DED filaments, for which mathematical models have shown that the length of the filaments is strongly dependent on the abundance of the core components of the DISC complex in particular cell type and the stimulation strength (Schleich et al, 2012).

We reconstructed the structure of the necrosome complex using AlphaFold3, incorporating key subunits, including RIPK1 (RHIM and DD domains), the RHIM domain of RIPK3, the full-length FADD, procaspase-8, and c-FLIP proteins (Fig. 9E,F). Based on AlphaFold3 predictions, we developed a representative model that fits to quantitative mass spectrometry data. We selected 7 RIPK1: 3 FADD DDs composition, as a reference one, due to its high AlphaFold2 and AlphaFold3 metrics, as well as its correspondence to the experimentally evaluated range of ratios (Figs. 6A and EV4B,D). When other core components were added to this basic unit, the final

model comprised a 7:3 RIPK1:FADD DD-containing oligomer, a short DED filament consisting of four procaspase-8 and two c-FLIP$_L$ subunits, and a single RIPK3 RHIM domain subunit (Fig. 9E,F). Notably, higher-order oligomerization of this so-called basic unit of necrosome complex (7:1:3:4:2 RIPK1:RIPK3:FADD:procaspase-8:c-FLIP$_L$) predicted by AlphaFold3 is likely possible. However, Alpha-Fold3 is limited in its capacity to predict these higher-order configurations. Hence, in light of aforementioned suggested dynamics in the composition stoichiometry of the necrosome, the next developments of AlphaFold might provide further insights into the dynamics of oligomer assembly.

The AlphaFold3-derived model provides further insights into the potential mechanisms of necrosome assembly. In particular, the model suggests that the size of the RIPK1 DD oligomer might be regulated by both RIPK1-RHIM domain nucleation and FADD binding to RIPK1. Specifically, RHIM oligomerization might prevent further recruitment of RIPK1 DD domains, while the DED filament, seeded by FADD-DED protein–protein interactions, might sterically inhibit the addition of additional RIPK1 DDs. The higher RIPK1-to-RIPK3 ratio likely reflects the requirement for RIPK1 oligomerization prior to RIPK3 recruitment. Notably, while AlphaFold3 successfully captured RHIM domain oligomerization, the structure of the RHIM oligomeric complex was reconstructed with low confidence (Fig. EV4C). This highlights a current limitation of AlphaFold3 in accurately modeling amyloid-like structures.

Overall our findings suggest that RIPK1 forms the major structural scaffold of the necrosome, just as caspase-8 forms the major core at the DED filaments (Dickens et al, 2012; Schleich et al, 2012), and all other components are recruited to RIPK1 homo-oligomers (Fig. 10). In this way, RIPK1 oligomer forms the structural platform for the assembly of this complex. It has to be noted that the possibility for RIPK3 homotypic interactions was suggested before by a number of biochemical studies (Mompean et al, 2018; Orozco et al, 2014). Importantly, our model does not exclude RIPK3 oligomerization within these RIPK1 amyloids, like making the small RIPK3 insertions, but the amount of RIPK1 at the necrosome appears to be higher than RIPK3 (Fig. 10). Furthermore, the hypothesis that RIPK1 serves as a major scaffold for necrosome assembly is consistent with recent reports on the assembly of this complex (Jacobsen et al, 2022; Pierotti et al, 2023).

Our results point out that the first stage of the necrosome assembly would require RIPK1 DD to outcompete the CD95 DD for binding to FADD, leading to the assembly of RIPK1 DD/FADD DD oligomer complexes (Fig. 10). Our structural model suggests that this process can be initiated by the formation of CD95/RIPK1/FADD intermediate or core oligomers. Afterward, due to competitive binding of RIPK1 DD to FADD DD and the

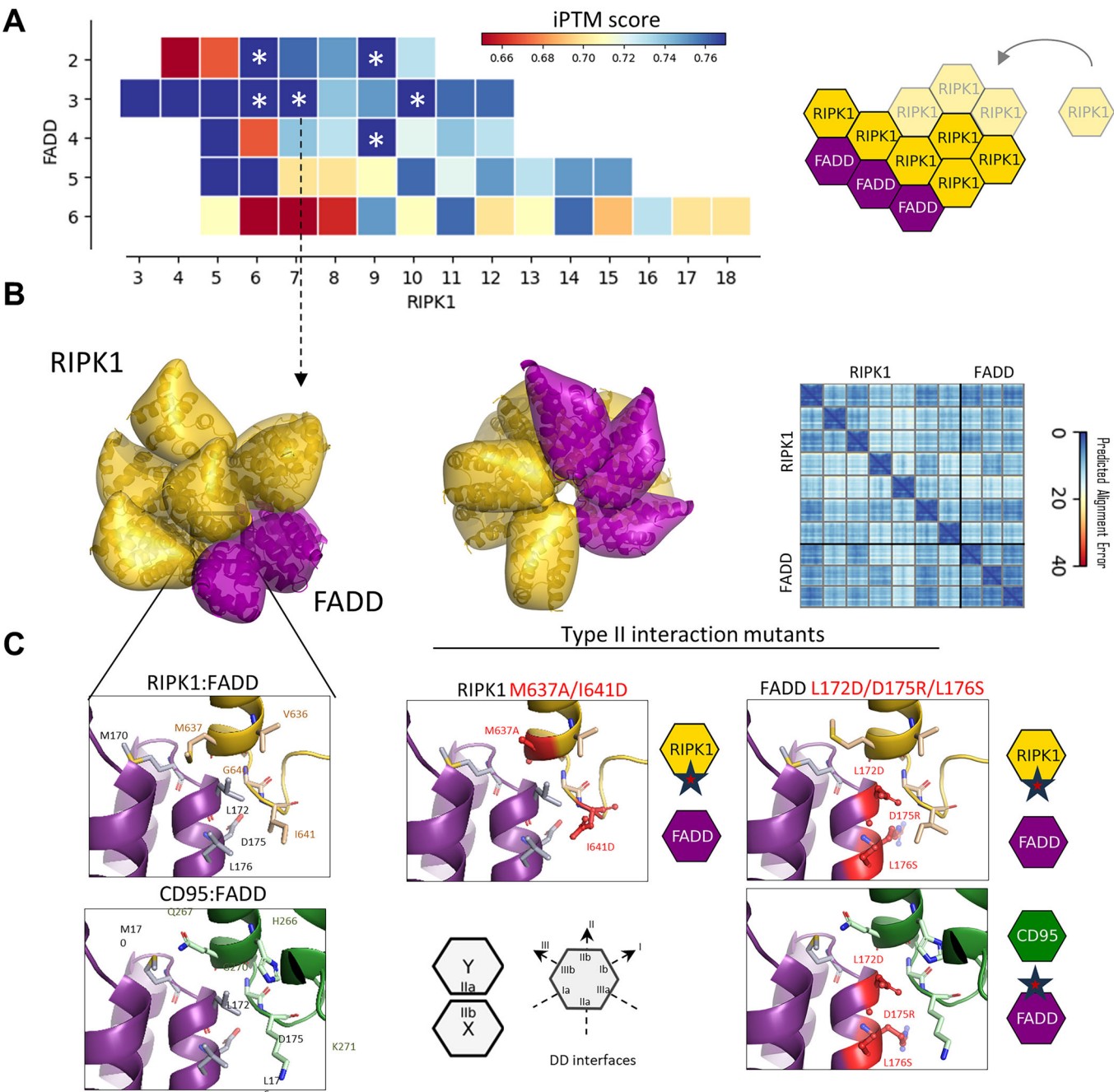

**Figure 6. AlphaFold-multimer prediction of the Structural Model of the necrosome complex.**

(A) The model confidence metrics, iPTM score for various models calculated for different ratios of RIPK1 and FADD proteins using AlphaFold2-multimer, are presented. The heatmap colors range from red (indicating low confidence) to blue (indicating high confidence). Models with the high iPTM score that fit to the experimentally determined stoichomteries obtained through AQUA peptides are indicated with *. (B) Schematic diagram of the RIPK1:FADD assembly is shown on the right generated by AlphaFold. Representative structural model of the RIPK1:FADD DD complex within the necrosome with high iPTM score is shown on the left. (C) Wild-type and mutant type-II DD interactions according to the necrosome structural model obtained by AlphaFold2-multimer are shown. The key residues are highlighted and their mutations are shown in red. Schematic representations of the DD interaction interfaces are displayed on the hexagon diagram. RIPK1 DD is shown in gold, FADD DD is shown in purple and CD95 DD is shown in green.

subsequent displacement of CD95 DD from the core complex, the formation of FADD-primed RIPK1 oligomer occurs, which provides the structural basis for the necrosome (Fig. 10). This is apparently followed by recruitment of more RIPK1 proteins leading

to the formation of RIPK1 oligomer, which serves as the core of the necrosome complex. The importance for the proper DD interactions for the necrosome assembly has been shown in previous studies (Meng et al, 2018).

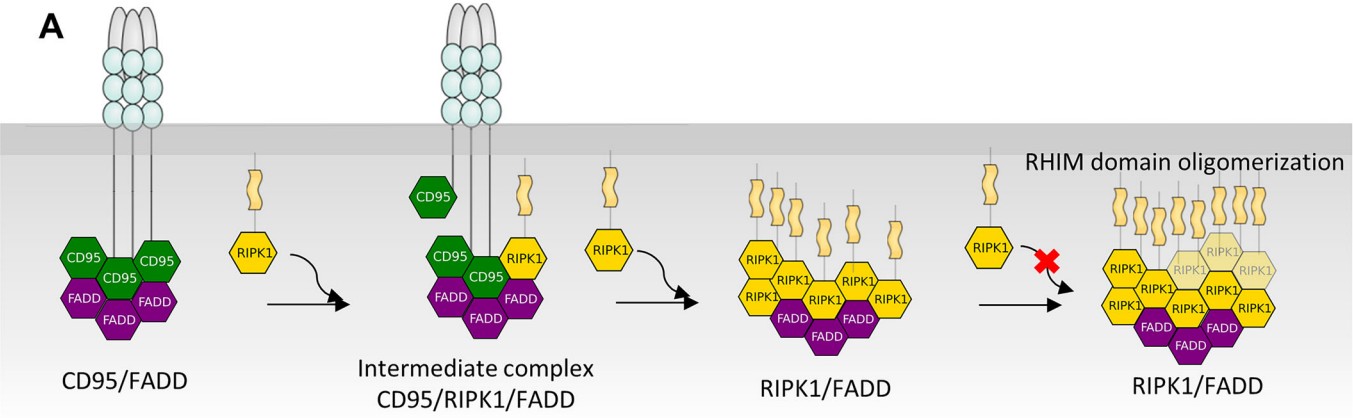

Quantitative mass spectrometry analysis in combination with functional genomics data revealed lower numbers of RIPK3 compared to RIPK1 in the majority of cancer cells. There are number of reports on the mechanisms of RIPK3 and RIPK1 activation (Meng et al, 2018; Orozco et al, 2014; Yuan and Ofengeim, 2024) as well as the structural analysis of the necrosomes (Li et al, 2012; Mompean et al, 2018). The ratios of RIPK1 and RIPK3 observed in these studies are different from the results obtained in our work. However, these studies were performed using overexpression of RIPK1 and RIPK3, in contrast

---

**Figure 7.   Validation of the model through the mining databases of mutations in RIPK1 DD.**

(A) Model depicting the formation of the necrosome in response to CD95 stimulation. RIPK1 DD/RHIM is shown in gold, FADD DD is shown in purple and CD95 DD is shown in green. (B) Somatic Mutations Observed in Cancer Tissues Involving the RIPK1 DD. Somatic mutations affecting the DD of RIPK1, including K599R and others, were extracted from the COSMIC database (Tate et al, 2019). Residues with somatic mutations observed located on type-II DD protein–protein interaction interfaces are indicated in cyan. RIPK1 DD is shown in gold, FADD DD in purple. Residues corresponding to the potential ubiquitinylation sites are denoted and colored in black. The scheme depicting the predicted functional significance of residues R599 and R603 in the homotypic type-I interaction of RIPK1 DD is shown on the right.

---

to our analysis, which was performed using immunoprecipitations of native complexes under endogenous conditions. Indeed, artificially changing the ratio of RIP kinases might lead to modulation of the composition of the necrosome, which might explain the results obtained by the others (Li et al, 2012; Mompean et al, 2018). Moreover, those studies were not done for the CD95-mediated necroptotic pathway in contrast to our study. Importantly, a higher concentration of RIPK1 compared to RIPK3 in the necrosome was also predicted by mathematical modeling using stochastic simulations (Matveeva et al, 2019), which would fit well with our study. Our findings are also consistent with RIPK1 dimerization as a key event driving necroptosis, as the RIPK1 dimer could provide the basis for RIPK1 oligomer assembly (Meng et al, 2018). Finally, the role of the balance between RIPK1 and RIPK3 in inducing cell death in cancer cells has recently been highlighted by the study using Protac targeting RIPK1, which was able to induce cell death in cancer cells (Mannion et al, 2024). This study provides important evidence for the correct stoichiometry of RIPK1 and RIPK3 in the necrosome to drive cell death induction.

Importantly, ubiquitinylation status of RIPK1 is a pivotal step, required for the formation of the necrosome (de Almagro et al, 2015; Feltham and Silke, 2017; Li et al, 2020). Therefore, the priming of necrosome formation by deubiquitylation of RIPK1 by BV6, leading to the assembly of RIPK1/FADD oligomers upon CD95L stimulation, also plays a key role in our model. However, the structural backround of how deubiquitinylation of RIPK1 promotes the formation of RIPK1/FADD oligomers in the course of the CD95-mediated necroptosis induction has to be delineated in the future studies. One of the molecular mechanisms might involve the steric hindrance of DD interaction interfaces due to polyubiquitinylation and/or subsequent conformational changes. Indeed, three residues within the RIPK1 DD have been reported to undergo ubiquitination in mouse RIPK1 (Hou et al, 2024), and these residues are conserved in the human RIPK1 sequences (K604, K627, and K648). Specifically, the ubiquitination of K648 and K627 may influence type-I and other DD interactions within the necrosome, according to our model (Fig. 7B).

The structure of DD oligomer of the necrosome complex within our model has been robustly validated through mutagenesis studies, providing a reliable foundation for assessing the impact of amino acid mutations on necroptosis induction. A key distinction from prior reports is the notably elevated oligomerization state of RIPK1 DD within the necrosome complex. Furthermore, our model proposes a plausible mechanism for the interaction of RIPK1 with CD95, which might explain the molecular mechanisms of the transition from the DISC to the necrosome. Our model was extensively validated by mapping the amino acid substitutions of RIPK1 DD observed in cancer tissue samples. We have observed a cluster of amino acid mutations within the type-II interaction

interface of RIPK1 and FADD, that plays the major role according to the structural modeling (Fig. 7B). Among these substitutions, I641 and M637 of RIPK1 were shown to have a functional effect on the DD oligomer assembly.

Taken together, we show that combining AlphaFold modeling with experimental stoichiometry constraints offers ample opportunities for predicting the structure and composition of macromolecular complexes. The structural model of the necrosome core predicted by AlphaFold offers unexpected insights into the potential mechanisms of its assembly. It also lays the groundwork for the rational design of small molecules and synthetic proteins aimed at regulating the necrosome complex structure (Jacobsen et al, 2022; Pierotti et al, 2023). This is of substantial interest for developing novel treatment strategies for autoimmune diseases and cancer.

## Methods

**Reagents and tools table**

| Reagent/resource | Reference or source | Identifier or catalog number |
|---|---|---|
| **Experimental models** | | |
| Human: SUIT-020 cell line | Konig et al, 2025 | N/A |
| Human: HT29 cell line | Konig et al, 2025 | N/A |
| Human: Jurkat 282 cell line | DSMZ collection, Braunschweig | ACC 282 |
| Human: Jurkat A3 cell line | Juo et al, 1998 | N/A |
| Human: Jurkat A3 C8 ko cell line | Juo et al, 1998 | N/A |
| Human: Jurkat A3 RIPK1 ko cell line | Juo et al, 1998 | N/A |
| Human: Jurkat A3 FADD ko cell line | Juo et al, 1998 | N/A |
| Human: HeLa RIPK1 ko RIPK3 cell line | Feoktistova et al, 2021; Jaco et al, 2017 | N/A |
| Human: HeLa FADD ko RIPK3 cell line | This manuscript | N/A |
| Human: HeLa-CD95 cell line | Neumann et al, 2010 | N/A |
| Human: MV4-11 cell line | DSMZ collection, Braunschweig | ACC 102 |
| **Recombinant DNA** | | |
| pRP-RIPK1 wt (WT-RIPK1) | Vector Builder | VB220223-1206adz |
| pRP-RIPK1-AD | Vector Builder | VB220317-1428gpu |
| pcDNA3 FADD (WT-FADD) | Addgene | # 78802 |

| Reagent/resource | Reference or source | Identifier or catalog number |
|---|---|---|
| pcDNA3 FADD-DRS | Genscript, this paper | N/A |
| pcDNA3 FADD-ADRN | Genscript, this paper | N/A |
| pSpCas9(BB)-2A-GFP (PX458) | Addgene | #48138 |
| **Antibodies** | | |
| Monoclonal rabbit phospho-RIPK1 (Ser166) | Cell Signaling | #65746 |
| Monoclonal rabbit anti-RIPK1 XP | Cell Signaling | #3493 |
| Monoclonal rabbit phospho-RIPK3 (Ser227), | Cell Signaling | #93654 |
| Monoclonal rabbit MLKL | Cell Signaling | #14933 |
| Polyclonal rabbit anti-actin | Sigma-Aldrich | A2103 |
| Polyclonal rabbit anti-RIPK3 | Abcam | ab226297 |
| Monoclonal rabbit anti-pMLKL (phospho S358) | Abcam | ab187091 |
| Monoclonal mouse anti-caspase-10 | MBL | M059-3 |
| Polyclonal mouse anti-CD95 | Santa Cruz | sc-8009 |
| Monoclonal mouse anti-FADD (clone 1C4) | Scaffidi et al, 2000 | N/A |
| Monoclonal mouse anti-caspase-8 (clone C15) | Scaffidi et al, 1997 | N/A |
| Monoclonal mouse anti-c-FLIP (clone NF6) | Scaffidi et al, 1999 | N/A |
| Monoclonal rabbit anti-APO-1 | Trauth et al, 1989 | N/A |
| Horseradish peroxidase-conjugated goat anti-mouse IgG1 | Southern Biotech | 1070-05 |
| Horseradish peroxidase-conjugated goat anti-mouse -2b | Southern Biotech | 1090-05 |
| Horseradish peroxidase-conjugated goat anti-rabbit | Southern Biotech | 4030-05 |
| **Oligonucleotides and other sequence-based reagents** | | |
| AQUA peptides procaspase-8 (NLYDIGEQLDSEDLASL[K $^{13}$C6,$^{15}$N2]; FLSLDYIPQ[R $^{13}$C6,$^{15}$N4]) | Schleich et al, 2012; Warnken et al, 2013 | N/A |
| AQUA peptides FADD (IDSIED[R $^{13}$C6, $^{15}$N4], ELLASL[R$^{13}$C6, $^{15}$N4)) | Schleich et al, 2012; Warnken et al, 2013 | N/A |
| AQUA peptide RIPK3 (LNLEEPPSSVP[K $^{13}$C6, $^{15}$N2], NDVMVSEWLN[K$^{13}$C6,$^{15}$N2]) | Boege et al, 2017 | N/A |
| AQUA peptide c-FLIP (LSVGDLAELLY[R $^{13}$C6,$^{15}$N4]; DVAIDVVPPNV[R $^{13}$C6,$^{15}$N4]) | Schleich et al, 2012; Warnken et al, 2013 | N/A |
| RIPK1 (NLLYELSEGIDSENL[K $^{13}$C6,$^{15}$N2], YQAIFDNTTSLTD[K $^{13}$C6,$^{15}$N2]) | Boege et al, 2017 | N/A |
| FADD gRNA ACACCGAGTGCAGCAGCACC | http://crispr.mit.edu/ | N/A |
| **Chemicals, enzymes, and other reagents** | | |
| CD95L | Fricker et al, 2010 | N/A |
| zVAD-fmk | BIOZOL | Bac-4026865.00025 |

| Reagent/resource | Reference or source | Identifier or catalog number |
|---|---|---|
| Nec-1s | Merck KGaA | 5.04297.0001 |
| GSK872 | Merck KGaA | 5.30389 |
| BV6 | Genentech | #OR-502922 |
| OPTI MEM | Thermo Fisher Scientific | 11058021 |
| DreamFect Gold | OZ Bioscience | DG81000 |
| Cell Line Nucleofector™ Kit V | Lonza | VCA-1003 |
| Cell Titer-Glo® 2.0 Assay | Promega | G9243 |
| Annexin Binding Buffer | BioLegend | 422201 |
| Annexin-V-FITC | ImmunoTools | 31490013×2 |
| Propidium Iodide (PI) | Sigma-Aldrich | 81845 |
| PROTEIN A SEPHAROSE CL4B | Th. Geyer | 17-0780-01 |
| DMEM Hams F-12 | Pan Biotech | P04-41150 |
| RPMI 1640 | Thermo Fisher Scientific | 21875034 |
| DMEM | Thermo Fisher Scientific | 41966052 |
| Fetal calf serum (FCS) | Pan Biotech | P30-3033 |
| Penicillin–Streptomycin | Biochrom | A2213 |
| Puromycin | Invitrogen | Ant-pr-1 |
| PBS | Pan Biotech | P04-36500 |
| Trypsin | Thermo Fisher Scientific | 25300-096 |
| Protease inhibitor | Merck KGaA | 04693132001 |
| Plasmocin™Prophylactic | Invivogen | ant-mpp |
| **Software** | | |
| GraphPad Prism 8 | GraphPad | https://www.graphpad.com/ |
| Image Lab | BioRad | https://www.bio-rad.com |
| IDEAS 2.0 | Cytek | https://cytekbio.com |
| AlphaFold2-Multimer model v3 | Varadi et al, 2022; Yin et al, 2022 | N/A |
| AlphaFold3 | Abramson et al, 2024 | N/A |
| Rosetta Modeling package | Leaver-Fay et al, 2011 | N/A |
| g:Profiler web server | Kolberg et al, 2023 | N/A |
| **Other** | | |

## Cell lines

The pancreatic cancer cell line SUIT-020 and the cervical cancer cell lines HeLa-CD95 (HeLa cells overexpressing CD95)(Neumann et al, 2010), were cultured in DMEM/Ham's F-12 (PAN Biotech, Germany). The colon cancer cell line HT29 and the cervical cancer cell lines HeLa RIPK1 ko, HeLa RIPK1 ko RIPK3 and HeLa FADD ko RIPK3 were cultured in DMEM (Thermo Fisher Scientific Inc.,

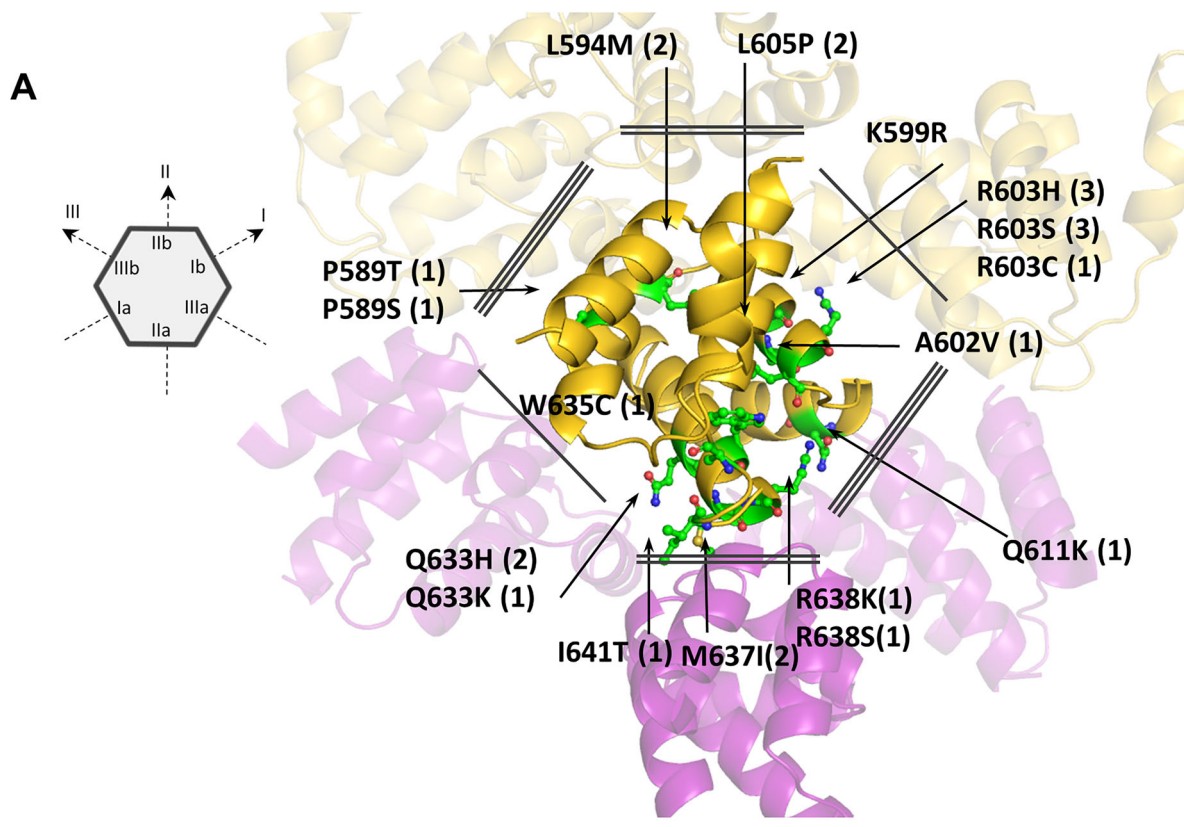

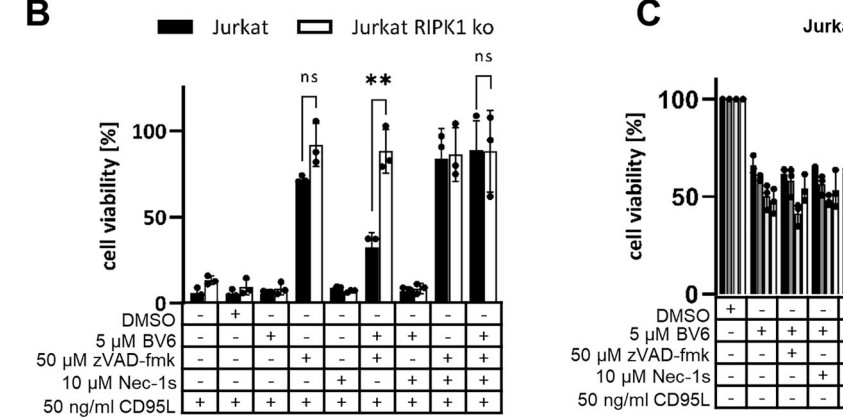

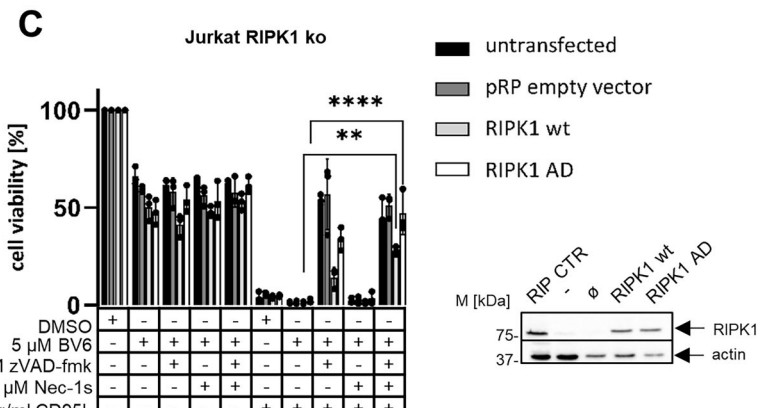

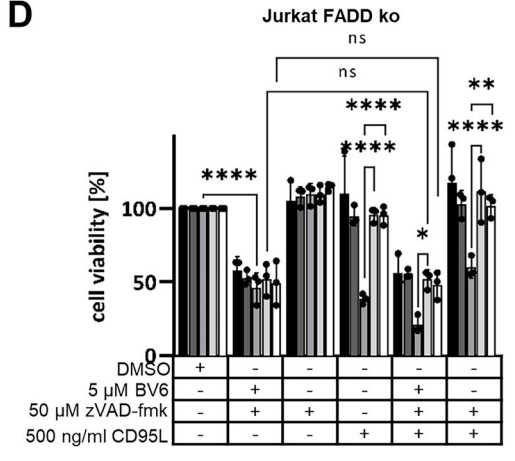

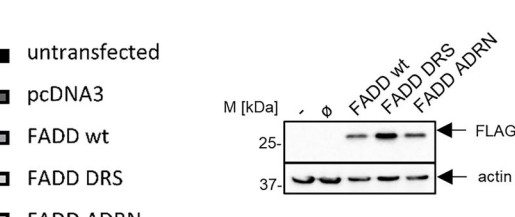

**Figure 8. Mutations of type-II DD interface in FADD and RIPK1 impair cell viability loss after CD95L/BV6/zVAD-fmk treatment.**

(A) Somatic Mutations Observed in Cancer Tissues within RIPK1 DD. DD interaction interfaces of RIPK1 are highlighted with 1 (type-I), 2 (type-II) or 3 (type-III) underlinings. Mutations were extracted from the COSMIC database. (B) Jurkat A3 and Jurkat RIPK1 ko cells were pretreated with 5 µM BV6, 50 µM zVAD-fmk and 10 µM Nec-1s for 1 h and afterwards stimulated with 50 ng/mL CD95L for 24 h. The complete experiment with all controls is shown in Fig. EV5B,C. (C) Jurkat RIPK1 ko cells were transfected with empty vector, WT-RIPK1 or RIPK1-AD (M637A, I641D). Transfected cells were pretreated with 5 µM BV6, 50 µM zVAD-fmk and 10 µM Nec-1s for 1 h and afterwards stimulated with 50 ng/mL CD95L for 24 h. Western blot transfection control is shown. Jurkat A3 cells were loaded on the same gel to have a positive control for RIPK1 expression (RIP CTR). Actin served as loading control. (D) Jurkat FADD ko cells were transfected with empty vector, WT-FADD, FADD-DRS (L172D, D175R, L176S) or FADD-ADRN (M170A, L172D, D175R, L176N). Transfected cells were pretreated with 5 µM BV6 and 50 µM zVAD-fmk for 1 h and afterward stimulated with 500 ng/mL CD95L for 24 h. Western blot transfection control is shown. Actin served as loading control. The mean and standard deviation from three experiments are shown. Cell viability was measured using the Cell Titer-Glo®-Luminescent Cell Viability Assay by Promega. Untreated cells were taken as 100%. Statistics were calculated with unpaired one-way ANOVA with Tukey post hoc test to compare two conditions. Significance values: ****$P < 0.0001$; ***$P < 0.001$; **$P < 0.01$; *$P < 0.05$; ns not significant. $P$ values from left to right for (B) $P = 0.5607$, $P = 0.0047$, $P > 0.9999$, (C) $P = 0.0018$, $P < 0.0001$, (D) $P < 0.0001$, $P > 0.9999$, $P > 0.9999$, $P < 0.0001$, $P < 0.0001$, $P = 0.0330$, $P < 0.0001$, $P = 0.0012$. wt wild type, - untransfected, ø empty vector. Source data are available online for this figure.

USA). T lymphoma cell lines Jurkat 282 and Jurkat A3, and its deficient variants: Jurkat RIPK1 ko, Jurkat FADD ko, Jurkat C8 ko were cultured in RPMI 1640 (Thermo Fisher Scientific Inc., USA) (Juo et al, 1998). The AML cell line MV4-11 was also cultured in RPMI 1640 (Thermo Fisher Scientific Inc., USA). All media except that used to grow MV4-11 cells were supplemented with 10% heat-inactivated fetal calf serum (FCS) and 1% penicillin–streptomycin. Cells were cultured at 37 °C and 5% $CO_2$. For HeLa-CD95 cells 0.0001% Puromycin was added. Media for MV4-11 cells was supplemented with 10% heat-inactivated FCS and 0.02% Plasmocin and 5.8% Additivum (β-mercaptoethanol, 1 M HEPES buffer pH 7.2, 100 mM Sodium pyruvate, 200 mM L-Glutamine, 10× non-essential amino acids, 10 mg/ml L-Aspartic acid). HeLa RIPK1 ko RIPK3 cells were previously generated (Feoktistova et al, 2021; Jaco et al, 2017). HeLa FADD ko RIPK3 cells were generated from HeLa RIPK3 cells using the pSpCas9(BB)-2A-GFP (PX458) plasmid (Addgene) gRNA sequences targeting the 5-end of the gene were designed using the open access software provided at http://crispr.mit.edu/. The gRNA sequence: ACACCGAGTGCAGCAG-CACC. Forty-eight hours after transfection, the cells were sorted with a BD FACSAria I (BD Biosciences), and single clones were isolated and analyzed to confirm the successful deletion of FADD by western blot. The cells were tested every 4 weeks for mycoplasma contamination.

## Antibodies and reagents

The following antibodies were used for western blot analysis: monoclonal phospho-RIPK1 (Ser166) (#65746), monoclonal anti-RIPK1 XP (#3493), monoclonal phospho-RIPK3 (Ser227) (#93654), monoclonal MLKL (#14933) antibodies from Cell Signaling Technology, USA; polyclonal anti-actin (A2103) from Sigma-Aldrich, Germany; polyclonal anti-RIPK3 (ab226297) and mono-clonal anti-pMLKL (phospho S358) (ab187091) antibodies from Abcam, Great Britain; monoclonal anti-caspase-10 (M059-3) antibody from MBL international, USA; polyclonal anti-CD95 (sc-715) antibodies from Santa Cruz, Germany; monoclonal anti-FADD (clone 1C4), monoclonal anti-caspase-8 (clone C15), and monoclonal anti-c-FLIP (clone NF6) antibodies were a kind gift of Prof. P. H. Krammer, (DKFZ, Heidelberg). Horseradish peroxidase-conjugated goat anti-mouse IgG1, -2b, goat anti-rabbit and rabbit anti-goat antibodies were from Southern Biotechnology, USA. Monoclonal anti-pRIPK3 (phospho S227) antibodies from Abcam, Great Britain, anti-phospho-MLKL (Ser358) antibody Set (#17-10400) from Merck, Germany, the monoclonal phospho-RIPK1

(Ser166) (#65746) and monoclonal anti-RIPK1 XP antibody (#3493) from Cell Signaling Technology, Massachusetts, USA were used for immunoprecipitations (IPs). All chemicals were of analytical grade and purchased from Merck (Germany) or Sigma (Germany). Recombinant LZ-CD95L was produced as described (Fricker et al, 2010). GSK872 (5.30389) was from Merck (Germany). BV6 was provided by Genentech, Inc. (# OR-502922) and the pan-caspase inhibitor zVAD-fmk (Z-Val-Ala-DL-Asp-fluoromethylketone) was ordered from Bachem Holding, Switzerland. The RIPK1 inhibitor 7-Cl-O-Nec-1 (Nec-1s; 5.04297.0001) and the RIPK3 inhibitor GSK872 (5.30389) were obtained from Merck, Germany.

## Western blot analysis and immunoprecipitations

The western blot analysis of total cellular lysates was implemented in accordance with our previous reports (Schmidt et al, 2015). The analysis of western blot images was performed by Image Lab 5.1 Software (BioRad). Immunoprecipitations (IPs) from $8 \times 10^6$ SUIT-020 or HT29 cells or $2 \times 10^7$ Jurkat 282, Jurkat A3, Jurkat C8 ko or Jurkat RIPK1 ko cells were carried out as previously reported (Hillert-Richter and Lavrik, 2021; Pietkiewicz et al, 2015a). For specific IPs, 2 µg of anti-FADD antibody (clone 1C4), anti-caspase-8 antibody (clone C15) or anti-APO-1 antibody and 2 µg of anti-phospho-RIPK1 or anti-RIPK3 or anti-phospho-RIPK3 or anti-RIPK1 or anti-MLKL antibodies were used, respectively. All IPs were rotated overnight at 4 °C, washed four times with PBS and prepared for western blot or mass spectrometry analyses.

## Size-exclusion chromatography (SEC)

SEC was done using a Superose 6 10/300 column connected to the Äkta pure 25 L chromatography system (GE Healthcare, Germany). After treatment of $8 \times 10^6$ SUIT-020 cells with the indicated stimuli, cells were lysed. 500 µL of cellular lysate (~10 mg/mL) was injected into a 250 µL sample loop with a syringe. 20 µL protease inhibitor (cOmplete™, Roche) was added to each collecting tubes. The fractions 2–6, 7–10, and 11–15 were pooled together and IPs were carried out from the pooled fractions as described before. All steps were performed at 4 °C and with a pump speed of 0.2 mL/min.

## Transfection

In total, $1 \times 10^6$ Jurkat RIPK1 ko or Jurkat FADD ko cells were used for transfection. For transfection, the Amaxa® Cell Line Nucleofector® Kit

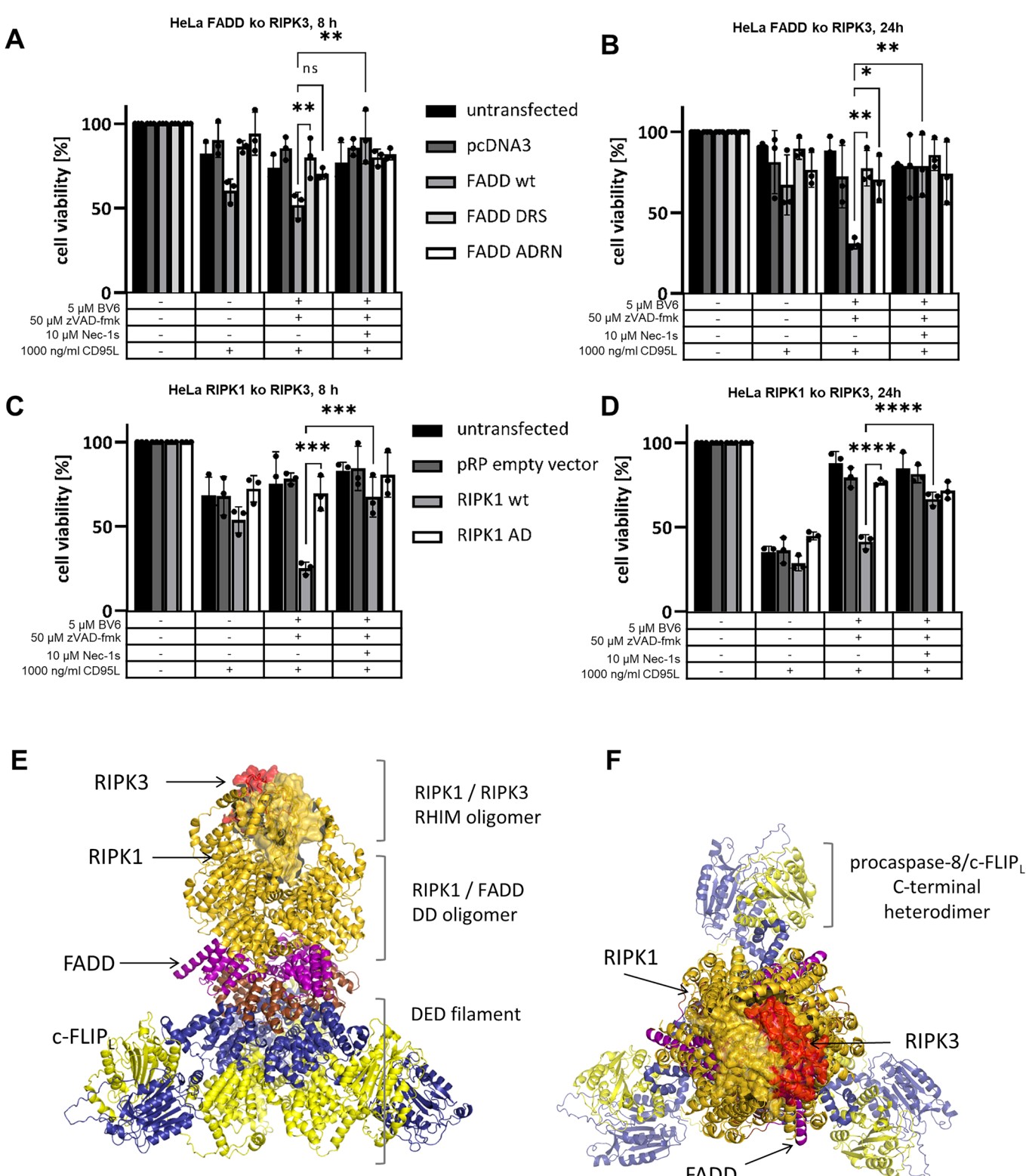

V from Lonza was used. Cells were transfected in accordance with the manufacturer's instructions and protocol for Jurkat cells. In all, $0.25 \times 10^6$ HeLa RIPK1 ko RIPK3 cells as well as HeLa FADD ko RIPK3 cells were seeded the day before transfection in six-well plates. Cells were transfected using DreamFect Gold (OZ Bioscience, France)

according to the manufacturer's instructions with 2 µg plasmid. Transfected cells were used after 24 h of transfection. WT-RIPK1 and RIPK1-AD (M637A, I641D) plasmids were produced by Vector Builder (USA). WT-FADD (#78802) was purchased from Addgene (USA). FADD-DRS (L172D, D175R, L176S) and FADD-ADRN

**Figure 9.  Mutations of type-II DD interface in FADD and RIPK1 impair cell viability loss after CD95L/BV6/zVAD-fmk treatment and AlphaFold3-generated model of the necrosome assembly.**

(A, B) HeLa FADD ko RIPK3 cells were transfected with empty vector, WT-FADD, FADD-DRS (L172D, D175R, L176S) or FADD-ADRN (M170A, L172D, D175R, L176N). Transfected cells were pretreated with 5 μM BV6, 50 μM zVAD-fmk or 10 μM Nec-1s for 1 h and afterwards stimulated with 500 ng/mL CD95L for 8 h (A) or 24 h (B). (C, D) HeLa RIPK1 ko RIPK3 cells were transfected with empty vector, WT-RIPK1 or RIPK1-AD (M637A, I641D). Transfected cells were pretreated with 5 μM BV6, 50 μM zVAD-fmk or 10 μM Nec-1s for 1 h and afterwards stimulated with 500 ng/mL CD95L for 8 h (C) or 24 h (D). Transfection controls are included in (EV9D-G). The mean and standard deviation from the three experiments are shown. Cell viability was measured using the Cell Titer-Glo®-Luminescent Cell Viability Assay by Promega. Untreated cells were taken as 100%. Statistics were calculated with unpaired one-way ANOVA with Tukey post hoc test to compare two conditions. Significance values: ****$P < 0.0001$; ***$P < 0.001$; **$P < 0.01$; *$P < 0.05$; ns not significant. P values from left to right for (A) $P = 0.0051$, $P = 0.2633$, $P = 0.0050$, (B) $P = 0.0037$, $P = 0.0272$, $P = 0.0024$, (C) $P = 0.0002$, $P = 0.0003$, (D) $P < 0.0001$, $P < 0.0001$. (E, F) The AlphaFold3-generated model includes a DD oligomer formed by the RIPK1 DD (gold) and FADD DD (purple), a DED filament composed of FADD-DED (brown), full-length procaspase-8 (blue), and c-FLIP$_L$ (yellow), as well as a RHIM oligomer consisting of RIPK1 and RIPK3 RHIM subunits. This complex comprises 2× c-FLIP$_L$, 4× procaspase-8, 3× FADD, 7× RIPK1, and 1 RIPK3 proteins. The RHIM oligomer is highlighted with the molecular surface. The side view is shown on the left (E) and top view is shown on the right (F). Source data are available online for this figure.

(M170A, L172D, D175R, L176N) plasmids were generated from FADD wt by Genscript (the Netherlands).

## Cell viability quantification by ATP assay

In all, $1.2 \times 10^4$ SUIT-020 or HT29 as well as HeLa FADD ko RIPK3 or HeLa RIPK1 ko RIPK3 cells transfected with the corresponding mutants were seeded in 96-well plates a day before experiments. Overall, $2 \times 10^4$ Jurkat A3, Jurkat C8 ko or Jurkat RIPK1 ko, as well as transfected Jurkat FADD ko or Jurkat RIPK1 ko cells were seeded in 96-well plates on the day of experiments. After described incubation times of indicated treatments, 50 μL of the Cell Titer-Glo solution was added to each sample. Measurements were accomplished according to the manufacturer's instructions (Cell Titer-Glo® 2.0 Luminescent Cell Viability Assay, Promega, Germany). The luminescence intensity was measured by a microplate reader Infinite M200pro (Tecan, Switzerland). The viability of untreated cells was normalized to 100%. Every condition was performed in duplicate.

## Mass spectrometry

Immunoprecipitations from $8 \times 10^6$ SUIT-020 or HT29 cells were performed as described above. Immunoprecipitates (protein A sepharose beads washed after IPs) were resuspended in 50 mM $NH_4HCO_3$ and incubated in 1 mM DTT at 56 °C for 45 min, which was followed by the subsequent S-Methyl methanethiosulphonate (MMTS) treatment (5 mM MMTS, 30 min). Tryptic digestion was performed by the addition of 0.5 μg Trypsin (Trypsin Gold, Promega) and incubation at 37 °C for 24 h. AQUA peptides (Boege et al, 2017; Schleich et al, 2012; Warnken et al, 2013) were spiked into the tryptic digestion solution in an absolute amount of 100 fmol of each peptide. After digestion, the supernatant was collected and dried in a vacuum centrifuge. The peptides were re-dissolved in 5 μl 0.1% trifluoroacetic acid (TFA) and purified on ZIP-TIP, C18-nanocolumns (Millipore, Billerica, USA). Peptides were eluted in 7 μl 70% (v/v) acetonitrile (ACN) and subsequently dried in a vacuum centrifuge. Samples were dissolved in 10 μl 2% ACN/0.1% TFA and separated on a 75 μm I.D., 25 cm PepMap C18-column (Dionex, Sunnyvale, USA) applying a gradient from 2% to 45% ACN in 0.1% formic acid over 120 min at 300 nl/min using an Ultimate 3000 Nano-HPLC (Thermo Scientific, San Jose, USA). Mass spectrometry was performed on a hybrid dual-pressure linear ion trap/orbitrap mass spectrometer (LTQ Orbitrap Velos Pro, Thermo Scientific, San Jose, USA) in exclusive orbitrap full MS

mode (FTMS; resolution 60,000; *m/z* range 400–2000). Absolute protein quantification was achieved using Skyline analysis platform (MacLean et al, 2010) for MS-peak integration on extracted ion chromatograms of selected peptide masses. The consideration of the monoisotopic precursor mass and at least two C13-isotopic variants ([M + 1] and [M + 2]) has been chosen for more accurate and confident quantification. The peak qualities of the quantified peptides were controlled by the "isotope dot product" (idotp) set to >0.95. Idotp provides a measure to assess precursor isotope distribution and its correlation between expected and observed pattern with optimal matching resulting in an idotp value of "1" (Schilling et al, 2012).

The qualitative analysis of interaction partners in pRIPK1-, FADD-, and c-FLIP-IPs by mass spectrometry was performed as described previously (Wohlfromm et al, 2024).

## Imaging flow cytometry

In all, $0.25 \times 10^6$ HT29 cells were seeded the day before the experiment in six-well plates. Overall, $1 \times 10^6$ Jurkat 282 or Jurkat RIPK1 ko cells were seeded on the day of experiment into six-well plates. Staining with Annexin-V-FITC and PI (Propidium Iodide), measurements, and analysis were done in accordance to our previous reports (Pietkiewicz et al, 2015b).

## Statistical analysis

Unpaired one-way ANOVA with Tukey post hoc tests were used to compare two different conditions statistically within one analysis using GraphPad Prism 8 software. The following values were used: ****$P < 0.0001$; ***$P < 0.001$; **$P < 0.01$; *$P < 0.05$; ns not significant.

## Databases mining

The RNA-seq data of RIPK1 and RIPK3 gene expression in the cancer tissues was derived from the TCGA database obtained from the TCGA data portal (Data ref: https://www.cancer.gov/tcga; release date after 1 May 2023). For the RNA-Seq-based expression normalization, Fragments Per Kilobase of transcript per Million mapped reads upper quartile (FPKM-UQ) method was used (Data ref: (Anders et al, 2015)).

The data on the RIPK1 mutations was extracted from the COSMIC database (Data ref: (Tate et al, 2019) (release v99, 28th November 2023)).

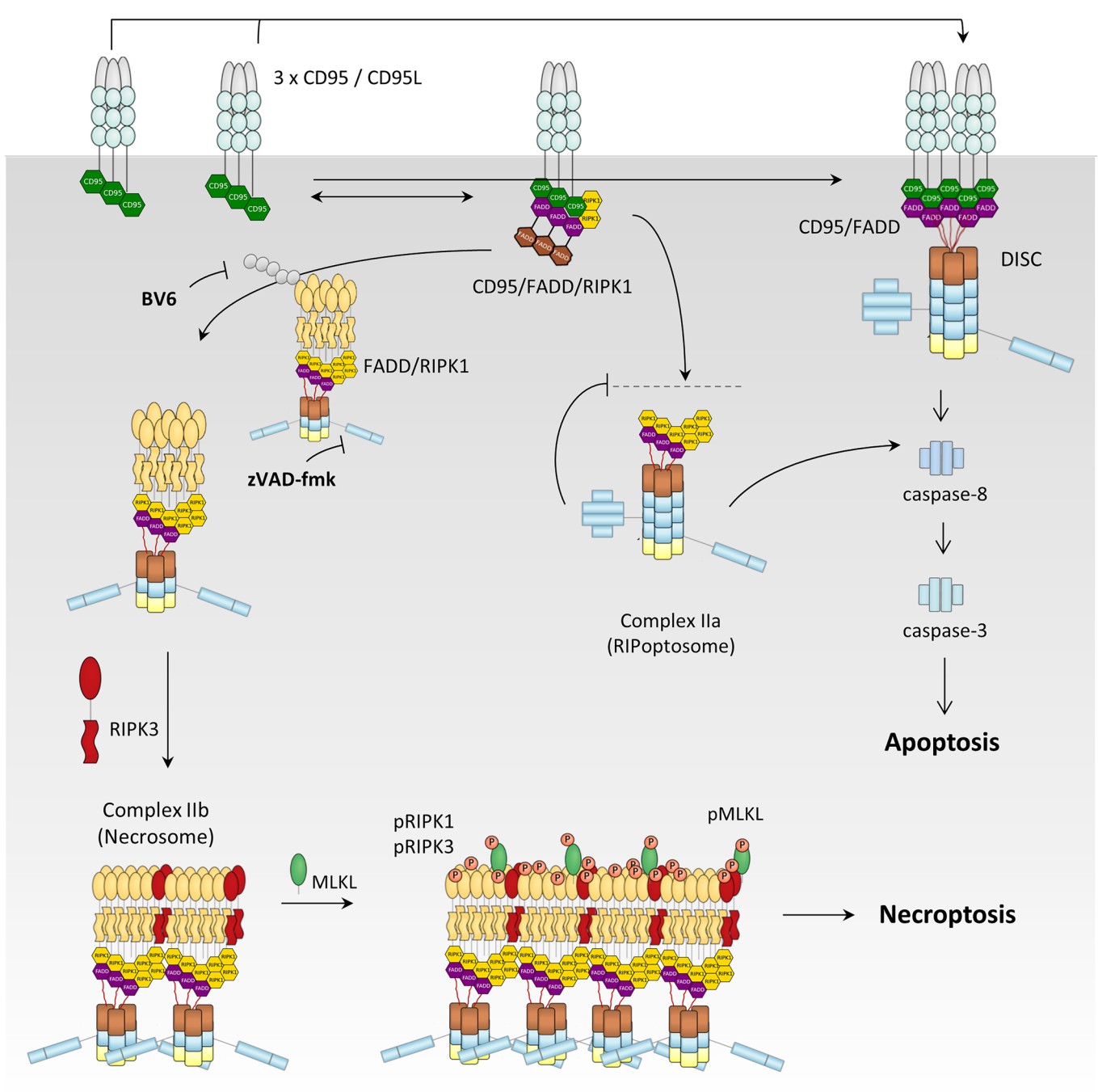

◄ **Figure 10.  Model of necrosome complex assembly in CD95 pathway.**

This model includes trimerization of the receptor upon CD95L binding, formation of the putative intermediate complex involving CD95/RIPK1/FADD DD interactions, leading to assembly of complex II or DISC, promoting necroptosis or apoptosis, respectively. The designations are shown at the bottom of the figure. According to the model, a high concentration of deubiquitinated RIPK1-forming oligomers is required to initiate necrosome assembly. In addition, FADD DD drives the initial assembly of complex II via interactions of FADD type-II DD interface and with type-II DD interface of RIPK1.

Mass spectrometry-based estimations of protein levels were obtained from ProteomicsDB (version on 1 February 2024) (Data ref: (Lautenbacher et al, 2022)). Levels of RIPK1 and RIPK3 in pancreatic ductal adenocarcinoma cases (Data ref: (Cao et al, 2021)) were derived from the Proteomic Data Commons (Data ref: (PDC: https://pdc.cancer.gov/pdc/) portal).

## Structural modeling

The molecular models of DD oligomers were predicted using AlphaFold3 and AlphaFold2-Multimer model v3 (Abramson et al, 2024; Jumper et al, 2021; Varadi et al, 2022). For the AlphaFold3 the web server (https://alphafoldserver.com) was used. For the AlphaFold-multimer the model inference and MSA alignment was carried out using ColabFold code and the number of recyclings was set to 20 (Mirdita et al, 2022). The AlphaFold-Multimer-v3 weights were employed. For the AlphaFold-multimer the DD sequences of CD95 (Uniprot ID P25445, 220–314), FADD (Uniprot ID Q13158, 93–188) and RIPK1 (Uniprot ID Q13546, 567–671) were used. For the AlphaFold3 the RHIM DD sequence of RIPK1 Uniprot ID Q13546, 521–671), RIPK3 (Uniprot ID Q9Y572, 446–518), full-length FADD (Uniprot ID Q13158), procaspase-8 (UniProt ID Q14790) and c-FLIP (Uniprot ID O15519) proteins were used. The ipTM (Interface Predicted Template Modeling) score was utilized to evaluate the quality of predicted molecular complexes using AlphaFold-multimer (Yin et al, 2022). The ranking score was utilized to evaluate the quality of predicted molecular complexes using AlphaFold3 (Abramson et al, 2024).

Molecular modeling of heterodimer interactions carried using Rosetta modeling package (version 2018.48.60516; (Leaver-Fay et al, 2011)) is decribed in detail in Appendix methods and Appendix Tables S1, S2 and S3. In brief, Relax, High-Resolution Protein–Protein Docking and InterfaceAnalyzer protocols were used for protein optimization, refinement and binding energy estimation. Protein domains for modeling were prepared using Relax module generating ensemble of 50 low-energy conformations. Complexes of DD subunits bound via type-I, -II, and -III interactions were predicted by superimposition and replacement of subunits in the reference structures of mouse CD95/human FADD DD (m-CD95/h-FADD) oligomer (3OQ9 (Wang et al, 2010)). Structural superimposition was carried out using using PyMOL 2.0 software (The PyMOL Molecular Graphics System, Version 2.0 Schrödinger, LLC). Each model was further refined using Relax and Rosetta High-Resolution Protein–Protein docking protocols. Complexes that had root mean square deviation (RMSD) > 1.5 Å from the reference structure were filtered out. Among ~3000 generated complexes top 5% with the lowest interaction energy were selected and the average interaction energy was calculated according to REF2015 scoring functions and Rosetta HBNet "buried" unsatisfied hydrogen bond penalties.

## Gene ontology analysis

The identification of Gene Ontology (GO) processes enriched related to CD95/BV6/zVAD-fmk (CBZ) treatment was performed using the g:Profiler web server ((Kolberg et al, 2023) accessed on 18 December 2024). Protein groups were divided based on those detected in the mass spectrometry analysis of CBZ-treated cells (FADD-IP/CBZ, FLIP-IP/CBZ, pRIPK1-IP/CBZ) and untreated cells (FADD-IP/medium control, FLIP-IP/medium control, pRIPK-IP/medium control, and beads-only control). The analysis of protein groups is presented in Dataset EV2. Gene ontology analysis for each group was conducted using the default settings of the g:Profiler web server.

## Data availability

The experimental data are accessible upon request. Mass spectrometry data are available via ProteomeXchange at https://www.ebi.ac.uk/pride/archive/projects/PXD059873. The models are available in ModelArchive at https://www.modelarchive.org/doi/10.5452/ma-uh8lz.

The source data of this paper are collected in the following database record: biostudies:S-SCDT-10_1038-S44318-025-00433-0.

## Peer review information

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

## Acknowledgements

The authors acknowledge the European Regional Development Fund (project ALBB), the Wilhelm Sander-Stiftung (2017.008.02), DFG (LA 2386), and START-Program, Faculty of Medicine of RWTH Aachen University (Az. 57/22). CK was supported by the fellowship of OvGU. The authors thank Prof. Natalia Giese (University of Heidelberg) for providing SUIT-020 cells, Prof. Thomas Brunner (University of Konstanz) for providing HT29 cells, and Prof. Irmela Jeremias (Helmholtz Munich) for providing Jurkat A3 and Jurkat FADD ko cells. The authors thank Lena Joppe for the technical assistance.

## Author contributions

**Nikita V Ivanisenko**: Software; Investigation; Writing—review and editing. **Corinna König**: Data curation; Validation; Investigation. **Laura K Hillert-Richter**: Investigation. **Maria A Feoktistova**: Investigation; Methodology. **Sabine Pietkiewicz**: Investigation. **Max Richter**: Investigation. **Diana Panayotova-Dimitrova**: Investigation. **Thilo Kaehne**: Investigation; Methodology. **Inna N Lavrik**: Conceptualization; Supervision; Writing—original draft; Project administration; Writing—review and editing.

Source data underlying figure panels in this paper may have individual authorship assigned. Where available, figure panel/source data authorship is listed in the following database record: biostudies:S-SCDT-10_1038-S44318-025-00433-0.

## Funding

## Disclosure and competing interests statement

The authors declare no competing interests.

# Expanded View Figures

**Figure EV1.   Combination of CD95L/BV6/zVAD-fmk leads to the appearance of necroptotic markers.**

(A, B) HT29 (A) or SUIT-020 (B) cells were pretreated for 1 h with 5 μM BV6, 50 μM zVAD-fmk and 10 μM Nec-1s and subsequently treated with 500 ng/ml CD95L for the indicated timepoints. Total cellular lysates were analyzed using western blot with the indicated antibodies. Actin served as loading control. One representative experiment out of three is shown. (C–H) HT29 (C, D) and Jurkat 282 as well as Jurkat C8 ko (E–H) cells were treated for 1 h with 5 μM BV6, 50 μM zVAD-fmk, 10 μM Nec-1s or with indicated concentrations of GSK872 and subsequently treated with 500 ng/ml (C, D, F, H) or 100 ng/ml (E, G) CD95L, or medium as a control for 24 h. Cell viability was measured using the Cell Titer-Glo®-Luminescent Cell Viability Assay by Promega. The cell viability of untreated cells was taken as 100%. Mean and standard deviation from three independent experiments are shown. Statistics were calculated with unpaired one-way ANOVA with Tukey post hoc test to compare two conditions. Significance values: ****$P < 0.0001$; ***$P < 0.001$; **$P < 0.01$; *$P < 0.05$; ns not significant. $P$ values from left to right for (C) $P < 0.0001$, $P > 0.9999$, $P < 0.0001$, $P < 0.0001$, $P < 0.0001$. (D) $P < 0.0001$, $P = 0.0499$ (E) $P = 0.0031$, $P = 0.0108$, $P < 0.0001$, $P < 0.0001$ (F) $P = 0.0018$, $P < 0.0001$, $P = 0.0004$, $P = 0.0002$ (G) $P = 0.0031$, $P = 0.0108$, $P < 0.0001$, $P < 0.0001$ (H) $P = 0.0018$, $P < 0.0001$, $P = 0.0004$, $P = 0.0002$. Source data are available online for this figure.

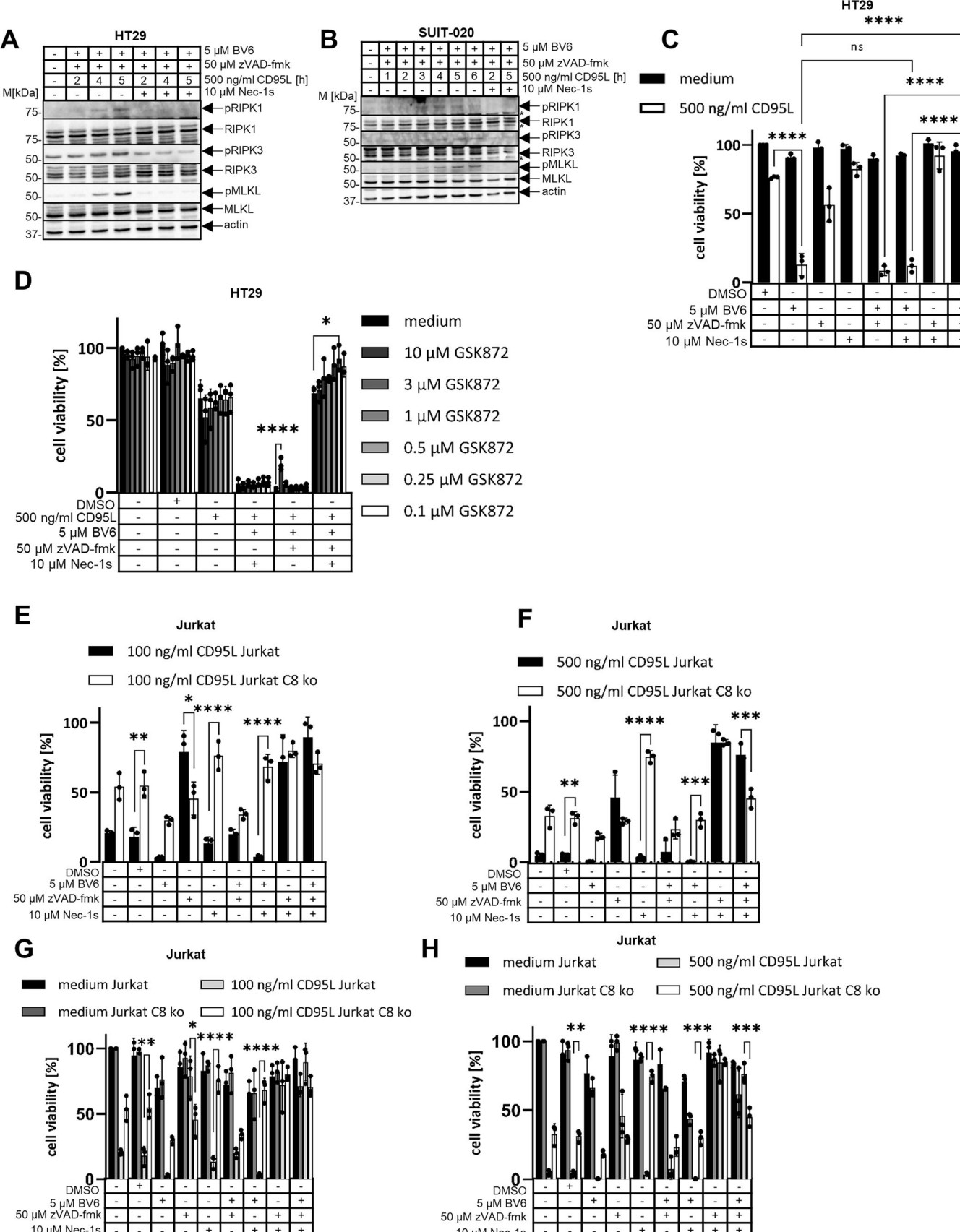

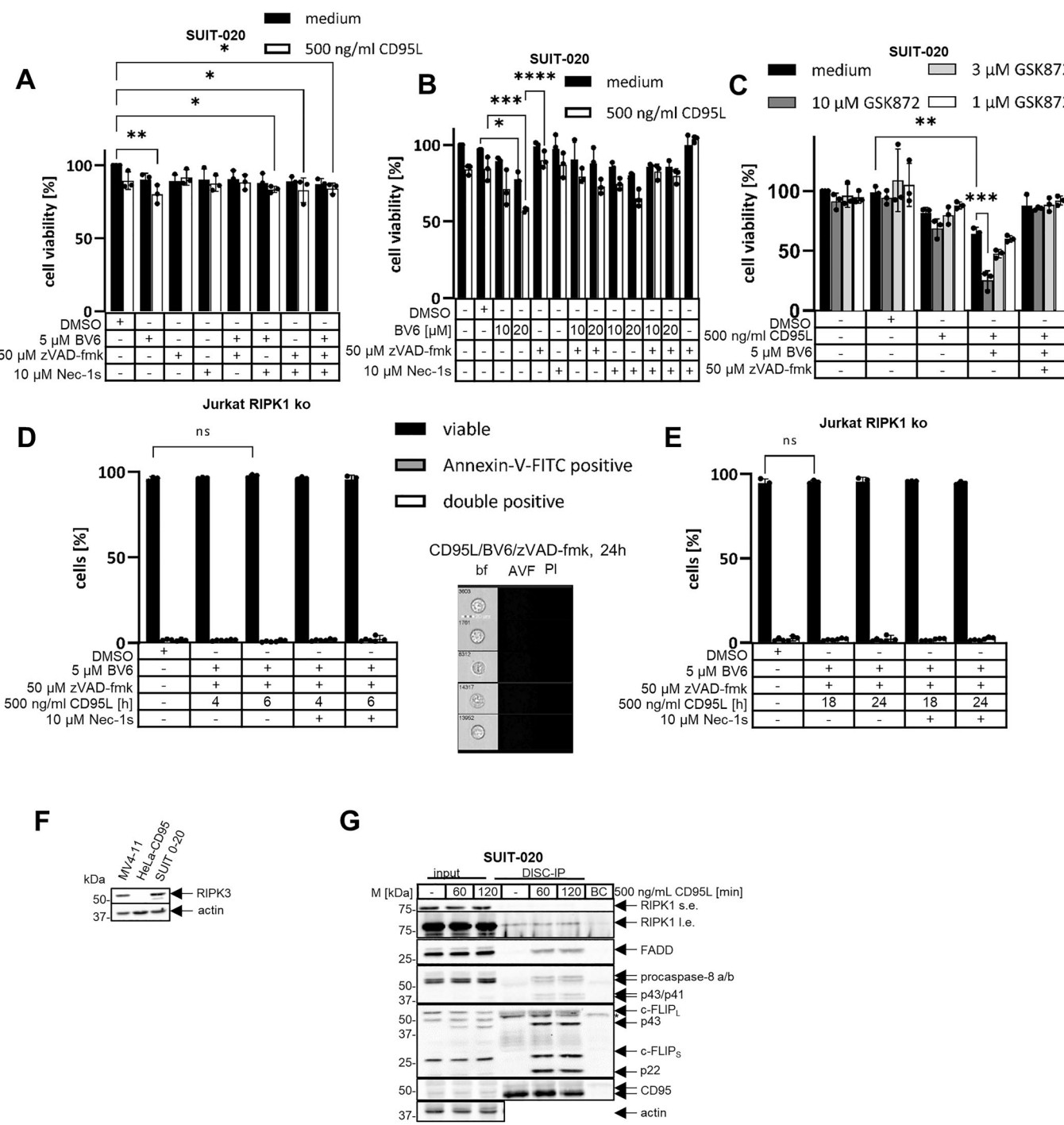

◀ **Figure EV2. CD95L/BV6/zVAD-fmk co-treatment induces necroptosis in sensitive cells.**

(A–C) SUIT-020 were pretreated with BV6, zVAD-fmk, Nec-1s and GSK872 with the indicated concentrations for 1 h. Afterwards, the cells were treated with 500 ng/ml CD95L for 24 h. Cell viability was measured using the Cell Titer-Glo®-Luminescent Cell Viability Assay by Promega. The cell viability of untreated cells was taken as 100%. (D, E) Jurkat RIPK1 ko cells were pretreated for 1 h with 5 µM BV6, 50 µM zVAD-fmk and 10 µM Nec-1s. Afterwards, the cells were treated with 500 ng/ml CD95L for the indicated time intervals. Cells were analyzed via imaging flow cytometry. Populations were gated for viable (negative), Annexin-V-FITC positive and double-positive (Annexin-V-FITC and PI positive) cells. Representative pictures from imaging flow cytometry with viable (negative) Annexin-V-FITC positive and double-positive (Annexin-V-FITC and PI positive) cells are shown in the middle. Mean and standard deviation from three independent experiments are shown. Statistics were calculated with unpaired one-way ANOVA with Tukey post hoc test to compare two conditions. Significance values: ****$P < 0.0001$; ***$P < 0.001$; **$P < 0.01$; *$P < 0.05$; ns not significant. $P$ values from left to right for (A) $P = 0.0030$, $P = 0.0281$, $P = 0.0198$, $P = 0.0403$ (B) $P = 0.0173$, $P = 0.0002$, $P < 0.0001$ (C) $P = 0.0033$, $P = 0.0006$ (D) $P > 0.9999$ (E) $p > 0.9999$. (F) Total cellular lysates of MV4-11, HeLa-CD95 and SUIT-020 cells were tested for RIPK3 expression using western blot. HeLa-CD95 cells do not express RIPK3 and were used as a negative control. One representative western blot out of two is shown. Actin served as loading control. (G) SUIT-020 cells were with 500 ng/ml CD95L for indicated timepoints and DISC IP was analyzed by western blot. Actin served as loading control. BC beads-only control, IP immunoprecipitation, bf bright field, AVF Annexin-V-FITC, PI Propidium Iodide, * unspecific band, IgG$_H$ the heavy chain of antibody. Source data are available online for this figure.

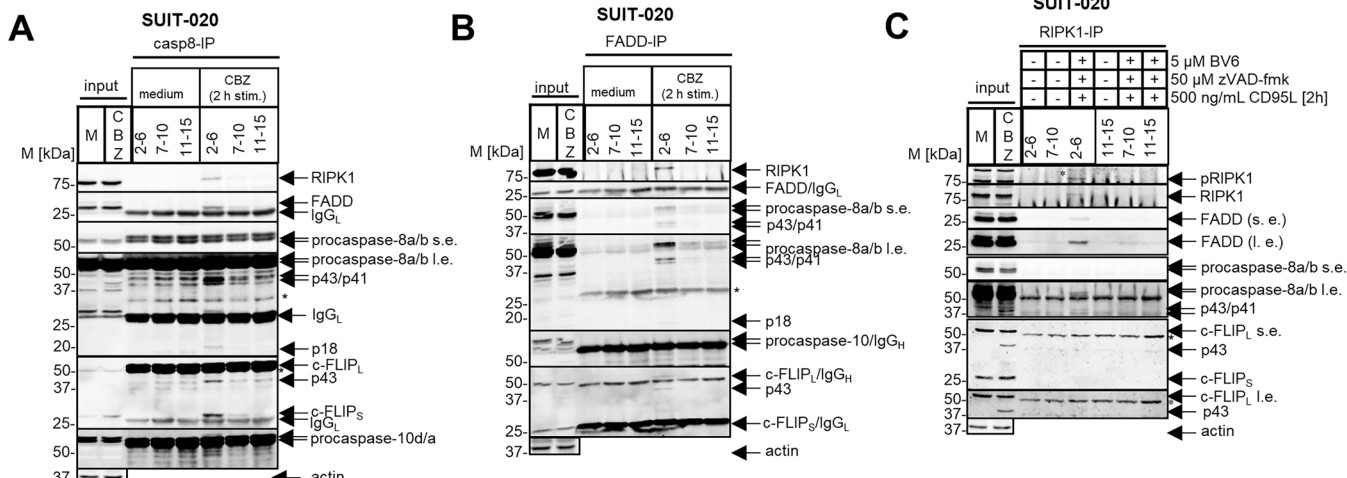

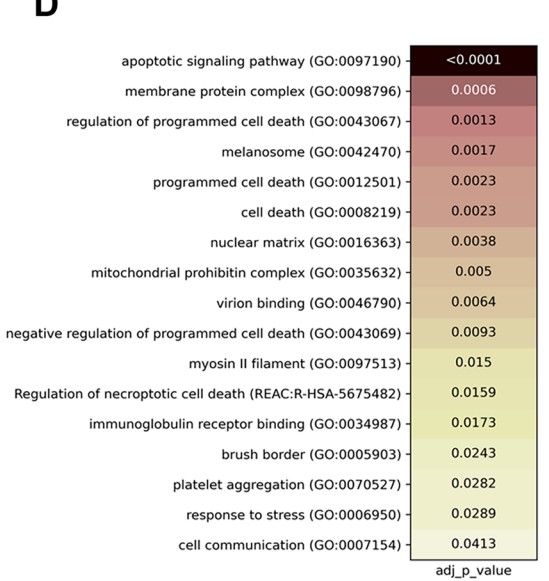

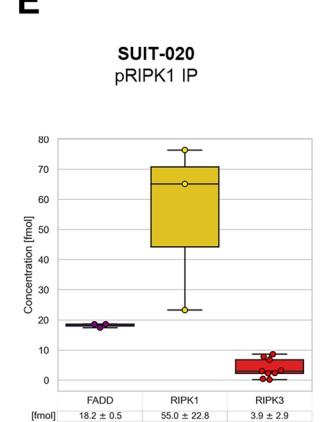

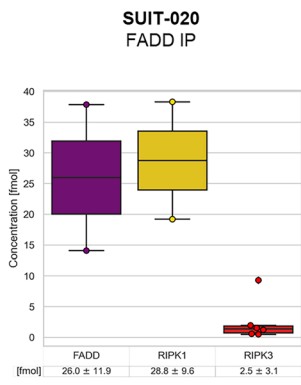

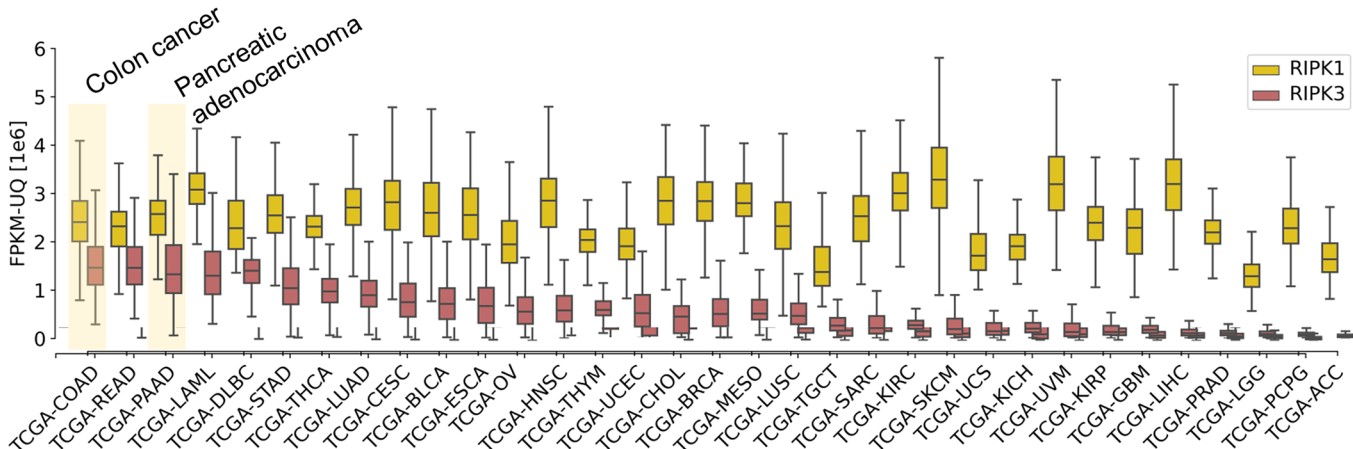

◀ **Figure EV3. Necrosome detection in HMW fractions and AQUA peptide mass spectrometry analysis of the necrosome.**

(A–C) SUIT-020 cells were prestimulated with 5 µM BV6 and 50 µM zVAD-fmk for 1 h and afterwards stimulated with 500 ng/mL CD95L for 2 h. Total cellular lysate was fractionated by gel filtration. The different fractions were pooled (2–6; 7–10, 11–15) followed by casp8-IP (A), FADD-IP (B) or RIPK1-IP (C). Total cellular lysates and IPs were analyzed by western blot and probed for the indicated proteins. Actin was used as a loading control for total cellular lysates (input). One representative experiment out of two is shown. (D) Groups of proteins identified by mass spectrometry analysis were analyzed by GO bioinformatics analysis. IPs from CBZ-treated cells (FADD-IP/CBZ, FLIP-IP/CBZ, pRIPK1-IP/CBZ) were analyzed against IPs from untreated cells (FADD-IP/control, FLIP-IP/control, pRIPK-IP/control and beads-only control). The major groups of proteins are shown. The full analysis is presented in Dataset EV2. The lowest significance is shown in black, the highest in light yellow. The proteins from the necroptotic cell death group have a similar significance to the proteins from the myosin group and immunoglobulin, which always have a high abundance in the IP experiments. (E, F) SUIT-020 cells were prestimulated with 5 µM BV6 and 50 µM zVAD-fmk for 1 h and afterwards stimulated with 500 ng/mL CD95L for 2 h. IP was done using anti-pRIPK1 (E) or anti-FADD (F) antibodies. The IPs were analyzed by AQUA peptide-based mass spectrometry analysis. The amounts of the AQUA peptides (fmols) corresponding to each protein, that were detected in the IP, are shown. Mean and standard deviations from three experiments are shown. Box plots show the distribution of the data using the median (center line), interquartile range (IQR; box limits represent the first and third quartiles, Q1 and Q3) and whiskers extending from the lower and upper quartiles to 1.5 × IQR. Points outside the whiskers are considered outliers and are shown as single points. (G) Gene expression levels in cancer tissues, as obtained from the TCGA data portal (https://www.cancer.gov/tcga). Gene expression levels were obtained using the RNA-Seq technique. IP immunoprecipitation, CBZ CD95L/BV6/zVAD-fmk, s.e. short exposure, l.e. long exposure, * unspecific band, IgG$_L$ the light chain of antibody. Source data are available online for this figure.

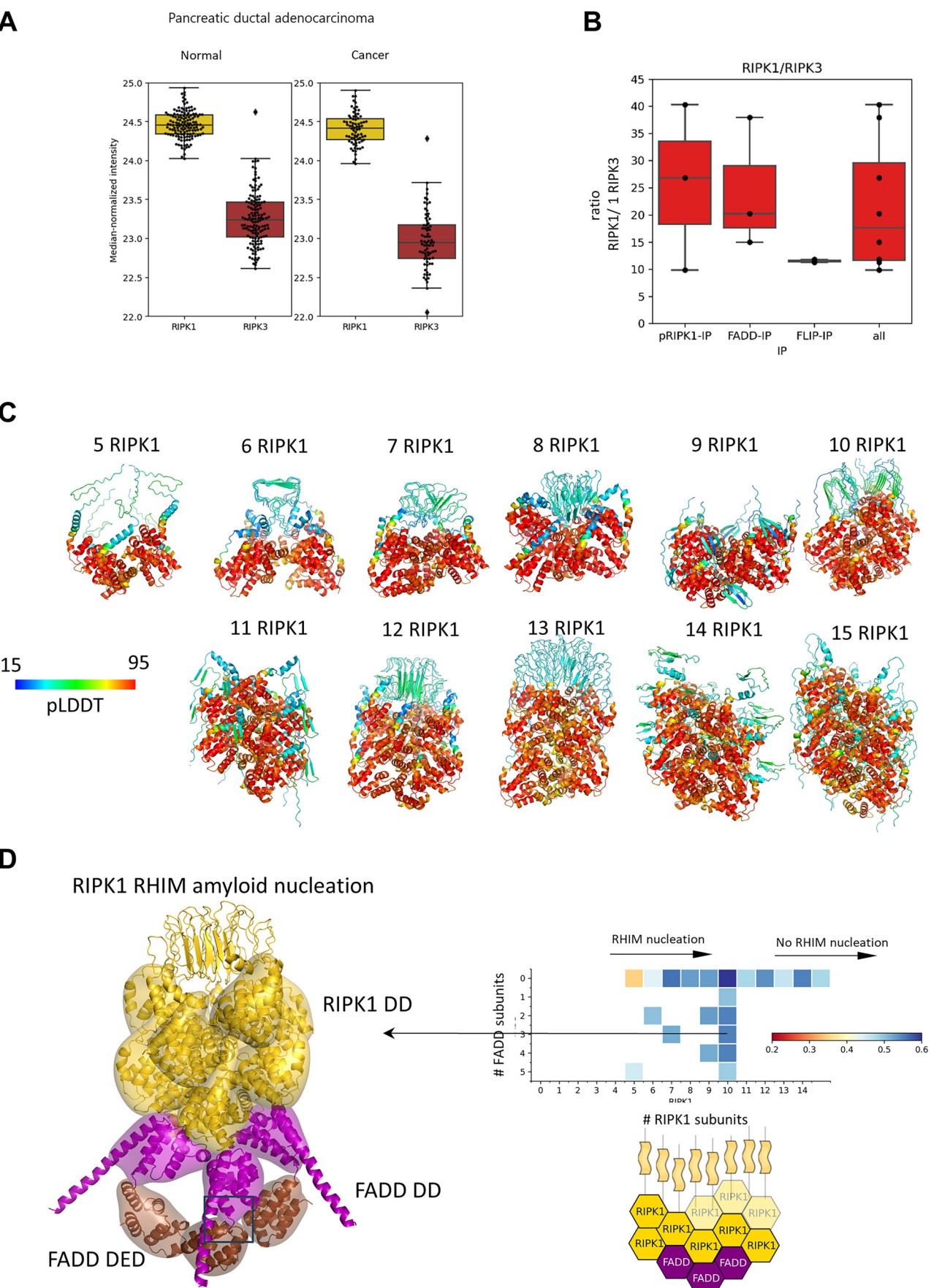

**Figure EV4. Necrosome stoichiometry analysis by mass spectrometry and AlphaFold3 modeling of necrosome.**

(A) Comparison of RIPK1 and RIPK3 expression levels from normal and cancer pancreatic ductal adenocarcinoma. (B) HT29 cells were pretreated with 5 μM BV6 and 50 μM zVAD-fmk for 1 h. Afterwards cells were stimulated with 500 ng/mL CD95L for 5 h, which was followed by pRIPK1-IP, FADD-IP or c-FLIP-IP. The IPs were analyzed by mass spectrometry analysis. The results are presented in the Fig. 5. Box plots show the distribution of the data using the median (center line), interquartile range (IQR; box limits represent the first and third quartiles, Q1 and Q3) and whiskers extending from the lower and upper quartiles to 1.5 × IQR. Points outside the whiskers are considered outliers and are shown as single points. The average ratios of proteins were calculated using interquartile range and presented in this panel. (C) Molecular models of RIPK1 (DD-RHIM) predicted by AlphaFold3 for various oligomerization states, with models colored according to the confidence metric pLDDT. (D) Molecular models of RIPK1 (DD-RHIM) and FADD predicted by AlphaFold3. The heatmap on the right shows AlphaFold3 ipTM scores for different stoichiometries of FADD and RIPK1 proteins, with colors ranging from red (low confidence) to blue (high confidence). The molecular model of a 10 RIPK1/3 FADD complex, as predicted by AlphaFold3, is shown on the left.

**A**

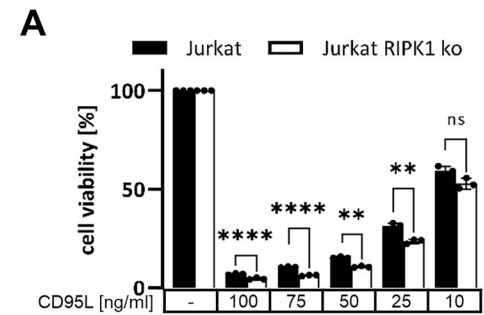

**B** Jurkat

**C** Jurkat RIPK1 ko

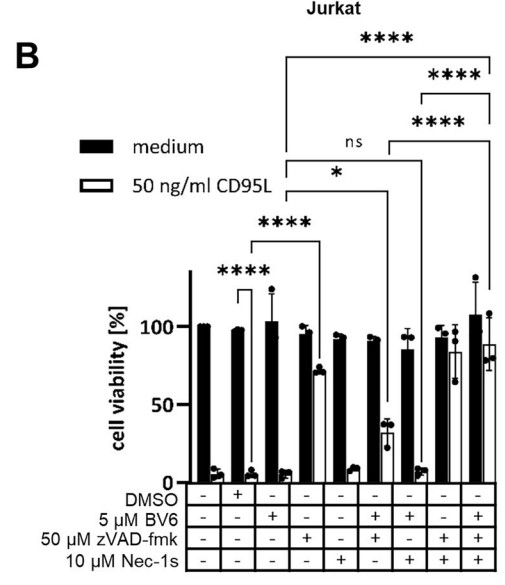

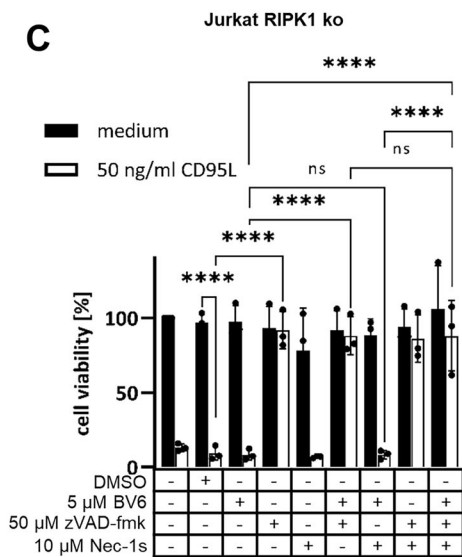

**D** HeLa FADD ko RIPK3

**E** HeLa FADD ko RIPK3

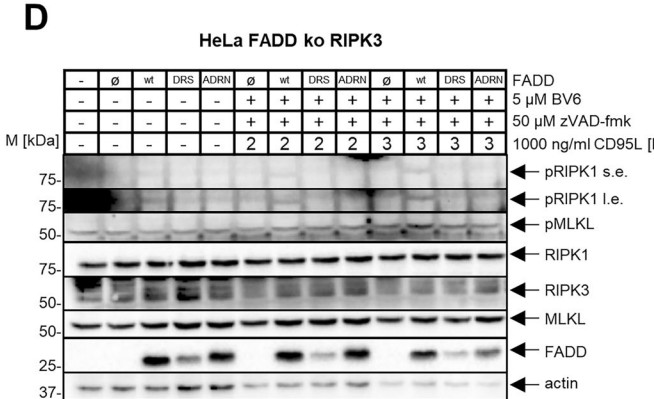

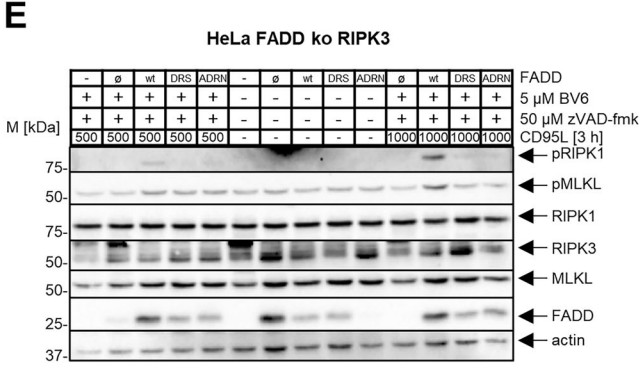

**F** HeLa RIPK1 ko RIPK3

**G** HeLa RIPK1 ko RIPK3

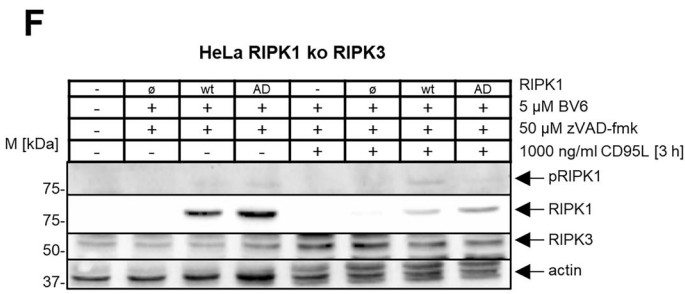

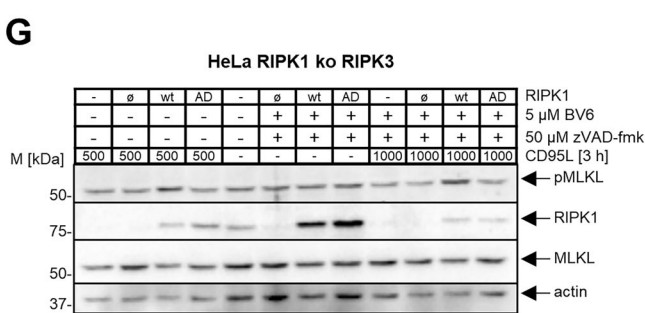

**Figure EV5.  The analysis of the role of the mutations in FADD and RIPK1 on necroptosis induction.**

(A) Jurkat A3 and Jurkat A3 RIPK1 ko cells were treated with indicated concentrations of CD95L for 24 h. (B, C) Jurkat A3 (B) or Jurkat A3 RIPK1 ko (C) cells were prestimulated with 5 µM BV6, 50 µM zVAD-fmk and 10 µM Nec-1s for 1 h and afterwards stimulated with 50 ng/mL CD95L for 24 h. Cell viability was measured using the Cell Titer-Glo®-Luminescent Cell Viability Assay by Promega. The cell viability measurements of untreated cells were taken as 100%. Mean and standard deviation from three independent experiments are shown. Statistics were calculated with unpaired one-way ANOVA with Tukey post hoc test to compare two conditions. Significance values: ****$P < 0.0001$; ***$P < 0.001$; **$P < 0.01$; *$P < 0.05$; ns not significant. (D, E) HeLa FADD ko RIPK3 cells were transfected with empty vector, WT-FADD, FADD-DRS (L172D, D175R, L176S) or FADD-ADRN (M170A, L172D, D175R, L176N). Transfected cells were pretreated with 5 µM BV6, 50 µM zVAD-fmk or 10 µM Nec-1s for 1 h and afterwards stimulated with 500 or 1000 ng/mL CD95L for 3 h or as indicated. (F, G) HeLa RIPK1 ko RIPK3 cells were transfected with empty vector, WT-RIPK1 or RIPK1-AD (M637A, I641D). Transfected cells were pretreated with 5 µM BV6 and 50 µM zVAD-fmk for 1 h and afterwards stimulated with 500 or 1000 ng/mL CD95L for 3 h. Total cellular lysates were analyzed using western blot with the indicated antibodies. Actin served as loading control. One representative western blot out of three is shown. WT wild type, - untransfected, ø empty vector, s.e. short exposure, l.e. long exposure. Source data are available online for this figure.

