## [Peer Review File · The EMBO Journal]

Oligomerised RIPK1 is the main core component of the CD95 necrosome

Nikita Ivanisenko, Corinna Koenig, Laura Hillert-Richter, Maria Feoktistova, Sabine Pietkiewicz, Max Richter, Diana Panayotova-Dimitrova, Thilo Kaehne, and Inna Lavrik

Corresponding author(s): Inna Lavrik (inna.lavrik@med.ovgu.de)

Review Timeline:

Submission Date:	9th Jun 24
Editorial Decision:	20th Aug 24
Revision Received:	9th Jan 25
Editorial Decision:	18th Feb 25
Revision Received:	13th Mar 25
Accepted:	26th Mar 25

Editor: Ioannis Papaioannou

Transaction Report:

Dear Inna,

Thank you again for submitting your manuscript EMBOJ-2024-118145 for consideration by The EMBO Journal and for your patience during peer review. Your manuscript has been seen by three experts in the field, and we have received the full set of their comments, which I have already shared with you for your information (they are included again below). I would also like to thank you for your detailed and informative response to the referee concerns (and tentative revision plan), which was very helpful for us to reach a balanced and fair decision on your manuscript.

Referees #1 and #3 are generally supportive of the work and the manuscript, find the topic important and relevant, and the amount of presented data commendable. On the other hand, referee #2 raises a major novelty concern, pointing out that a significant part of the manuscript recapitulates what was already reported in an earlier study. I further discussed this point with the referees, who expressed divergent opinions on the extent of conceptual novelty provided by the study taking the earlier report mentioned by referee #2 into consideration. Your detailed explanation of the novelty of this study compared to the previous literature, as well as your willingness to extend your study by addressing the remaining concerns raised by all three referees, were very helpful for us to reach a decision on the manuscript.

On balance, and upon discussion of the referees' input and your feedback in our editorial team, we are open to considering a substantially revised version of your manuscript for publication in The EMBO Journal should you be able to successfully address the referees' concerns along the lines you described in your tentative revision plan. In particular, I would like to recommend taking on board referee #1's major point on the need for additional validation of the proposed necrosome stoichiometry, as well as the first major comment/shortcoming mentioned by referee #3 regarding new possible necrosome components, which we think would be important for the field and increase the overall novelty of the study. I would also like to emphasize the need for citing all relevant literature in the revised version of your manuscript, and for clearly discussing the novelty provided by this study over the previous papers, both in your manuscript and in your responses to the referees' comments.

Please include in your resubmission a detailed point-by-point response addressing all referees' comments. I should add that it is EMBO Journal policy to allow only a single round of major revision, and acceptance of your manuscript will therefore depend on the completeness of your responses in this revised version. Please let me know if you have any questions or comments that you would like to discuss further with me.

We generally allow three months as standard revision time (November 19, 2024). As a matter of policy, competing manuscripts published during this period will not negatively impact our assessment of the conceptual advance presented by your study. However, we request that you contact us as soon as possible upon publication of any related work, to discuss how to proceed. Should you foresee a problem in meeting this three-month deadline, please let us know in advance and we may be able to grant an extension.

Thank you for the opportunity to consider your work for publication in The EMBO Journal. I look forward to your revision.

Best regards,

Ioannis

Instructions for preparing your revised manuscript

1. When you are ready to submit the revision, please upload:

- A Word file of the manuscript text (including legends of main Figures, EV Figures and Tables). Please make sure that changes are highlighted (or "tracked") to be clearly visible.

- Individual production-quality figure files (one file per figure). When assembling your figures, please refer to our figure preparation guidelines in order to ensure proper formatting and readability in print as well as on screen:

If the data shown in a figure are obtained from n {less than or equal to} 2, please use scatter plots showing the individual data

points.

- i. the name of the statistical test used to generate error bars and P values
- ii. the number (n) of independent experiments (please specify technical or biological replicates) underlying each data point (discussion of statistical methodology can be reported in the Materials and Methods section, but figure legends should contain a basic description of n, P, and the test applied)
- iii. the nature of the bars and error bars (s.d., s.e.m.).

- A point-by-point response to the referees' comments, with a detailed description of the changes made (as a word file). All referees' concerns must be fully addressed and their suggestions taken on board. When preparing your letter of response to the referees' comments, please bear in mind that this will form part of the Review Process File and will therefore be available online to the community. Please note that you have the possibility to opt out of the transparent process at any stage prior to publication by letting the editorial office know (contact@embojournal.org); if you do opt out, the Review Process File link will point to the following statement: "No Peer Review File is available with this article, as the authors have chosen not to make the review process public in this case.". For more details on our Transparent Editorial Process, please visit our website: <https://www.embopress.org/page/journal/14602075/authorguide#transparentprocess>

- Expanded View (EV) files (replacing Supplementary Information) that are collapsible/expandable online. A maximum of 5 EV Figures can be typeset. EV Figures should be cited as "Figure EV1, Figure EV2" etc. in the text, and their respective legends should be included in the manuscript file after the legends of regular figures. See detailed instructions regarding Expanded View files here:

- For the figures that you do NOT wish to display as Expanded View figures, they should be bundled together with their legends in a single PDF file called "Appendix", which should start with a short Table of Contents (including page numbers). Appendix figures should be referred to in the main text as: "Appendix Figure S1, Appendix Figure S2" etc. Please see detailed instructions here: <https://www.embopress.org/page/journal/14602075/authorguide#expandedview>

- A complete author checklist, which you can download from our author guidelines (<https://www.embopress.org/page/journal/14602075/authorguide>). Please note that the checklist will also be part of the Review Process File.

2. Please note that no statistics should be calculated and shown in Figures if n=2. Please also note that each p value should be reported as an exact value.

3. Before submitting your revision, primary datasets (and computer code, where appropriate) produced in this study need to be deposited in appropriate public databases (see <https://www.embopress.org/page/journal/14602075/authorguide#dataavailability>).

In particular, you are kindly requested to deposit the mass spectrometry data and the in silico models that were generated in your study in appropriate public repositories. The accession numbers, databases, and the specific URLs (links) should be listed in a formal "Data availability" section (placed after Materials and Methods) that follows the model below (see also <https://www.embopress.org/page/journal/14602075/authorguide#dataavailability>):

Data availability

- RNA-seq data: Gene Expression Omnibus GSE46843 (<https://www.ncbi.nlm.nih.gov/geo/query/acc.cgi?acc=GSE46843>)
- [data type]: [name of the resource] [accession number/identifier/doi] ([URL or identifiers.org/DATABASE:ACCESSION])

*** All links should resolve to a page where the data can be accessed. ***

*** Please remember to provide in the Data availability section of your revised manuscript reviewer passwords if the datasets are not yet public. ***

*** The Data Availability Section is restricted to new primary data that are part of this study. In case you have no data that require deposition in a public database, please state so instead of referring to the database: "Our study includes no data deposited in public repositories." under the heading "Data availability". ***

*** Please use detailed data citations for already available datasets that were re-analyzed in your study - for more information on the format, see point #9 below. ***

4. Please check that the title and the abstract of the manuscript are brief, yet explicit, even to non-specialists. The length of the title should not exceed 100 characters, and the abstract should be a single paragraph not exceeding 175 words.
5. The Materials and Methods need to be described in the manuscript using our "Structured Methods" format, which is now required for all research articles. According to this format, the Materials and Methods section includes a single "Reagents and Tools Table" -listing key reagents, experimental models, software and relevant equipment and including their sources and relevant identifiers- followed by a "Methods and Protocols" section describing the methods. More information on this format as well as detailed instructions, examples, and a template (.docx) for the "Reagents and Tools Table" can be found in our author guide: <https://www.embopress.org/page/journal/14602075/authorguide#structuredmethods>.
6. Please also note our reference format: <https://www.embopress.org/page/journal/14602075/authorguide#referencesformat>.
7. At EMBO Press we ask authors to provide source data for the main manuscript figures. Our source data coordinator will contact you to discuss which figure panels we would need source data for and will also provide you with helpful tips on how to upload and organize the files.
8. Please remember: digital image enhancement is acceptable practice, as long as it accurately represents the original data and conforms to community standards. If a figure has been subjected to significant electronic manipulation, this must be noted in the figure legend or in the "Materials and Methods" section. The editors reserve the right to request original versions of figures and the original images that were used to assemble the figure.
9. Our journal encourages inclusion of data citations in the reference list to directly cite datasets that were obtained from public databases. Data citations in the article text are distinct from normal bibliographical citations and should directly link to the database records from which the data can be accessed. In the main text, data citations are formatted as follows: "Data ref: Smith et al, 2001" or "Data ref: NCBI Sequence Read Archive PRJNA342805, 2017". In the Reference list, data citations must be labeled with "[DATASET]". A data reference must provide the database name, accession number/identifiers, and a resolvable link to the landing page from which the data can be accessed at the end of the reference. Further instructions are available at: <https://www.embopress.org/page/journal/14602075/authorguide#referencesformat>.
10. We request authors to consider both actual and perceived competing interests. Please review our policy (<https://www.embopress.org/page/journal/14602075/authorguide#conflictsofinterest>) and update your competing interests statement if necessary. Please name this section 'Disclosure and competing interests statement' and place it after the Acknowledgements section.
11. Please note that all corresponding authors are required to provide an ORCID ID upon submission of a revised manuscript (<https://orcid.org/>). Please find instructions on how to link your ORCID ID to your account in our manuscript tracking system in our Author guidelines (<https://www.embopress.org/page/journal/14602075/authorguide#authorshipguidelines>).
12. We use CRediT to specify the contributions of each author in the journal submission system. CRediT replaces the author contribution section, which should be removed from the manuscript. Please use the free text box to provide more detailed descriptions. See also guide to authors: <https://www.embopress.org/page/journal/14602075/authorguide#authorshipguidelines>.
13. Further information is available in our Guide For Authors: <https://www.embopress.org/page/journal/14602075/authorguide>
14. We would also welcome the submission of cover suggestions or motifs to be used by our Graphics Illustrator in designing a cover.
15. Please use the link below to submit your revision:
<https://emboj.msubmit.net/cgi-bin/main.plex>

Referee #1:

In this study Ivanisenko and colleagues apply in silico alphafold approaches alongside quantitative proteomics to build a model of necrosome constitution, enabling implication of an assembly model. This, in my view, a key biological question that has not been adequately addressed to date. The authors data is substantial and robust and, to a large extent, I think the data supports the authors model. In my view some points require further addressing.

- I appreciate the ip of RIPK1 to pull down the necrosome as a valid approach, however ideally one would like to validate this data (for instance stoichiometry) using additional approaches, potentially the pRIPK1 antibody affect necrosome stability and will of course detect any free pRIPK1 as well, potentially affecting the interpretation of the data. Some additional validation (hopefully giving similar data) with antibodies versus FLIP and FADD would give added support to the necrosome stoichiometry proposed by the authors.

- there are various unexplained observations that require clarification, for instance MLKL ip in 5D (HT29s) pulls down both phosphor RIPK1 and 3 in the absence of any stimulus. Figure 4B there is pRIPK3 associating with pRIPK1 in the absence of any stimulus, though potentially this may be a non-specific band ?

- Overall, the amount of data presented here is heroic, however as presented it will require the reader to be equally heroic to read it in depth. I would suggest moving a significant amount of the main figure panels to supplemental data - for instance, the first two figures while important to establish the model (i.e. these cells die via necroptosis) could be moved to supplemental.

Referee #2:

Manuscript #EMBOJ-2024-118145

General Remarks

This study looks at the necrosome the key complex driving necroptosis. While some of the data with regard to the RIPK1 mutants could be of value to the field, far too much of the manuscript is devoted to basic experiments that recapitulate much of what is already known and doesn't thoroughly analyse the RIPK1 mutants themselves. I'd advise the authors to review Geserick et al, 10.1083/jcb.200904158, their detailed analysis of caspase-8 IPs in the presence of CD95L, smac-mimetic and Z-VAD-FMK are directly relevant.

Summary: Fig. 1 performs a Western blot analysis of necroptosis markers in 3 cell lines as well as a cell death analysis Fig. 1F - H. Fig. 2 looks at varying some of the parameters to find the optimal cell death conditions, for example time of CD95L or Nec-1. Fig. 3 performs FADD, (p)RIPK1 and caspase-8 IPs in the SUIT-020 line that overall confirms that these proteins are in a complex, possibly together with RIPK3 although the reagents for RIPK3 are not great. Fig. 4 is similar to Fig. 3 but looks at the other cell lines (HT29s, Jurkat) and also extends to MLKL. Since MLKL appears to be immunoprecipitated by the phospho-RIPK1 antibody even in the absence of a necroptotic signal (Fig. 4A lane 1, Fig. 4B lane 6) when the levels of phospho-RIPK1 should be negligible, this does raise concerns about the specificity of the IP in general and the data certainly can't be used to conclude that MLKL is in the complex (as in page 9 of the text). Fig. 5 is similar to Fig. 4 using (p)RIPK3 and MLKL IPs. While there is a suggestion of specificity, mostly with caspase-8, on the whole there is a lack of difference between IPs {plus minus} BCZ treatment . Therefore the main conclusion possible from this figure is that RIPK3 and phospho-RIPK3 antibodies are not very good tools to explore the necrosome (and might therefore be better as a supplementary figure). Fig. 6 uses quantitative Mass Spectrometry to look at the levels of RIPK1, FADD and RIPK3. Basically it shows that there is more FADD and RIPK1 in the complex than RIPK3. Since there are different ratios of FADD and RIPK1 depending on whether FADD or RIPK1 is pulled down whereas if there was a single complex one would expect ratios to stay the same, it's very unclear what solid quantitative conclusions we can draw from this data. The RNA levels (Fig. 6E,F), while no doubt of interest to some-one, are peripheral to this figure since level of mRNA doesn't reliably correlate with protein level. Fig. 7 uses Alphafold to predict the structure of a "minimal" necrosome containing a FADD and RIPK1 DD. Since it contains only the DDs of these proteins and relies to some extent on the stoichiometry data generated in Fig. 6 to choose which structure is most accurate, it fails to convince. Fig. 9 is where the manuscript becomes more interesting as it tests the predicted mutants, however the cell viability assays do not sufficiently investigate the effect of the mutations on complex formation etc to prove that signalling has been affected in the predicted manner. i.e. earlier stoichiometry and IP experiments are needed to validate.

Specific Remarks

Fig. 8 legend states, "Residues with somatic mutations observed located on type II protein-protein interaction interfaces are indicated in cyan, while other residues with somatic mutations observed are shown in cyan." I think one of these should be green.

Fig. 9D & E Were RIPK1 mutants used in these panels? The legend doesn't say although the text on page 14 says "the introduction of mutations into RIPK1 inhibited CD95L/BV6/zVAD-fmk-induced cell death". If mutants were not used I fail to see the logical connection to the rest of this figure which is about the point mutants. If they were used I do not see the positive control of death with wild type RIPK1.

Referee #3:

Ivanisenko, König, Hillert-Richter and colleagues present a really interesting and well formulated dissection of the chronology of FasL signaling and the componentry of the necrosomes underpinning induction of necroptotic death. FasL is poorly characterized as a necroptotic ligand and this study fills a massive void in knowledge. The authors take quantitative data, validating blots, and apply contemporary protein modelling approaches to understand complex assembly. The strength is in the use of multiple cell lines, multiple time points, and validating the models with mutational studies.

The work is extensive and is far more substantive than their prior submission, which I feel merits publication. The only major shortcoming for me is that the authors have not used their IP/MS approach to report new possible necrosome components (or MLKL) even though they are likely present in their existing datasets. On upload to an appropriate database for inspection (i.e. PRIDE), these will be revealed, so why not discuss here? There's an opportunity here for them to elaborate on the identified interactors in the AQUA MS, their stoichiometries, and contextualize how they might contribute to necroptotic signaling/necrosome stability or assembly. I do not expect them to validate new interactions using an orthogonal method as this is already a massive study, but I would appreciate some commentary about what other proteins they identified. Here I make a number of suggestions for additional analyses and data presentation, but do not expect further experiments.

Major comments

1. Please ensure the MS dataset is uploaded to the PRIDE database. Please comment on other components beyond those blotted that appear in the IPs. Whether they are novel or not is immaterial, they'd be important data for the field. Especially interesting if there are FasL specific interactors, although this may be fantasy. If there are, however, this would add enormously to the citability of this work. I'm especially interested to know if ZBP1 popped up in these interactomes. This is a highly topical area and would benefit the authors to discuss regardless or not of occurrence in their necrosome IPs.
 2. Citations. The key references are listed for the necroptosis pathway in the second last paragraph of page 3, although over the past 4 years, knowledge has moved enormously on the underlying mechanism of MLKL activation. I encourage the authors to contemporize their reference list in the intro. In Discussion on page 16, Orozco et al. is one of many that have contributed to knowledge of the apoptosis-necroptosis switch. This was published in concert with other papers and there are others from, for example the Vucic lab, that add to our understanding on the chronology of PTMs and signaling events.
 3. The idea of RIPK1 as the nucleating force in human necrosomes has been proposed in the period between the last submission and this one - e.g. Jacobsen CDDis and Pierotti Biochem J. This could be suitably acknowledged to reflect the idea that pRIPK1 is a scaffold for these complexes in human cells. I feel it is also worth noting this is probably peculiar to human over mouse cells, which is the focus of the submission in question.
 4. The shortcomings of RIPK3 reagents are duly noted. This has advanced somewhat for monoclonals in the years since first submission, however. I do not feel the authors need to revisit this but I wanted to note that hRIPK3 antibodies from Petrie Cell Rep 2019 and Chiou EMBO Mol Med 2024 have now been vetted for WB and IF/IHC, respectively. Hopefully the availability of vetted and ever-improving reagents will overcome the difficulties with detection of hRIPK3 in future studies.
 5. Please plot all replicate data points on bar charts.
 6. In Figure 6, if the kinetics are anything like TNF on HT29 cells, the authors should see MLKL in the necrosome at 3+ hours. It would be of broad interest to compare the abundance of RIPK3 and MLKL in the IPs. Meng Nat Comm 2021 proposed that RIPK3 and MLKL reside in a 1:1 complex in the cytosol under basal conditions. It would be of enormous interest to know how this plays out in the real world inside cells, and quantitative mass spec seems like the ideal opportunity to define this.
- #### Minor
7. The authors have now too few articles in the text, although this not a concern since I am sure the copyeditors/typesetters will fix this
 8. Second last line, page 10. "the ones" could be replaced with "those"
 9. Discussion. I'm not sure what the statute of limitations is but I feel that saying necroptosis is recently discovered might be an oversell. It is certainly more recent than apoptosis, but I think it'd be more useful to put a time stamp on necroptosis, such as by saying only the past 24 (Holler) or 19 (Degterev) years.
 10. I might have missed it, but I'm unclear why antibodies towards mouse RIPK3 and MLKL are included in the methods. My reading (apologies if I've missed it) is that the study is on a bunch of human lines.
 11. In Figure 2, there are 2 fonts used for the key and the other labels. I suggest harmonizing.

Point to Point Response

First of all, we would like to cordially thank all the reviewers for taking the time to read this manuscript and important critical remarks, which allowed us to improve the quality of our work. Please, find our point to point response below.

Referee #1 (Report for Author)

General comment

In this study, Ivanisenko and colleagues apply *in silico* alphafold approaches alongside quantitative proteomics to build a model of necrosome constitution, enabling implication of an assembly model. This, in my view, a key biological question that has not been adequately addressed to date. The authors' data is substantial and robust and, to a large extent, I think the data supports the authors model. In my view, some points require further addressing.

General answer

We would like to thank the reviewer for the kind words and appreciation of our efforts.

Comment 1

I appreciate the ip of RIPK1 to pull down the necrosome as a valid approach, however ideally one would like to validate this data (for instance stoichiometry) using additional approaches, potentially the pRIPK1 antibody affect necrosome stability and will of course detect any free pRIPK1 as well, potentially affecting the interpretation of the data. Some additional validation (hopefully giving similar data) with antibodies versus FLIP and FADD would give added support to the necrosome stoichiometry proposed by the authors.

Answer 1

We fully agree with this comment of the reviewer. We performed immunoprecipitations with antibodies against **c-FLIP and FADD** using well-established antibodies (Golks et al, 2006, J.E.M.; Lavrik et al, 2008, JBC; and data not published). This was followed by quantitative mass spectrometry analysis, which provided additional support for the previously obtained necrosome stoichiometry. In these IPs, we also observed more RIPK1 than RIPK3, fully supporting our hypothesis. We present the Western Blot analysis of the FADD- and c-FLIP IPs in Figure 3 and the mass spectrometry analysis in Figures 4 and 5. Please also see the new Figures 3, 4 and 5 in the text of the manuscript and parts of Figure 4 below.

Part of Figure 4 for the reviewer. (B-E) HT29 cells were pretreated for 1 h with 5 μ M BV6 and 50 μ M zVAD-fmk and subsequently treated with 500 ng/ml CD95L for 5 h (**B, C, E**) or 6 h (**D**) or left untreated (medium). The pRIPK1-IP, c-FLIP IP, MLKL-IP and FADD-IP were carried out using anti-pRIPK1, anti-c-FLIP, anti-MLKL and anti-FADD antibodies, respectively. The IPs were analyzed by AQUA mass spectrometry analysis. Mean and standard deviation from five (**B, E**) or two (**C, D**) experiments are shown. Abbreviations: IP immunoprecipitation

Comment 2

There are various unexplained observations that require clarification, for instance MLKL ip in 5D (HT29s) pulls down both phosphor RIPK1 and 3 in the absence of any stimulus. Figure 4B there is pRIPK3 associating with pRIPK1 in the absence of any stimulus, though potentially this may be a non-specific band?

Answer 2

We fully support this criticism of the reviewer. We have **many unspecific bands** in our IPs, which especially includes **MLKL -and pRIPK3-IPs** highlighted by the reviewer. We were more accurate in indicating all the non-specific bands in our IPs as well as the specific proteins. There are many bands around 50 kDa, which also correspond to the heavy chains of the immunoprecipitating antibodies

and at the same time could indicate the presence of the specific proteins if they have a molecular mass around 50 kDa, which is the case for both RIPK3 and MLKL. We have paid a particular attention to the proper indication of signals corresponding to the heavy chains of the immunoprecipitating antibodies, non-specific signals and specific signals. We fully agree that we were not precise enough in this aspect in the previous version of the manuscript.

To specific remarks on the IPs in former figures 5D and 4B: It is important to note that these particular IPs have been moved to Appendix Figures following Reviewer 2's comments. The former MLKL IP (5D) is currently **Appendix Figure S1D**. We believe that this was a very poor IP, as indeed hardly any components (except for p43-FLIP) were co-immunoprecipitated in this IP. For the **pRIPK3** signal, we believe that this is a signal from **the heavy chain** of the immunoprecipitating antibody, which we have indicated in the current version. For the pRIPK1 signal in the former **Figure 4B**, now **2B** (shown below), in the absence of any stimulus, it could be noted that this signal is slightly increased upon time of stimulation. In this experiment, we did not show unstimulated conditions in the IP, we start with the 1 hour time point. Unstimulated For the **pRIPK3** signal in this IP, we believe that this is also a signal from **the heavy chain** of the immunoprecipitating antibody, which we have indicated in the current version. As highlighted above, we indicated the signal for the antibody chains in all **IPs (IgG)** as well as **unspecific bands (*)** more carefully in this and other figures and we hope that the reviewer will be satisfied with our changes to **Figures 2, 3** and **Appendix Figures S1 and S2**. We also write about IgG issue in the text. Please, see below: (i) representative Fig. 2B with highlighted IgGs and unspecific bands(*)-all the other IP figures are now presented in the similar fashion, (ii) paragraph discussing IgG issue and (iii) Figure S1D.

Part of Figure 2 (2B) for the reviewer: HT29 cells were pretreated for 1 h with 5 μ M BV6, 50 μ M zVAD-fmk and 10 μ M Nec-1s and subsequently treated with 500 ng/ml CD95L for indicated timepoints. pRIPK1- IPs were carried out using anti-pRIPK1 antibodies and analyzed by Western Blot. Abbreviations: s.e.-short exposure, l.e.-long exposure, IP-immunoprecipitation, BC-beads only control pulldown, *-unspecific band, IgG_H-the heavy chain of antibody

Text, discussing IPs and issues with detection of MLKL and RIPK3 in pRIPK1, Caspase-8, FADD and c-FLIP-IPs, p.6:

To initiate the necrosome assembly, the investigated cell lines were treated with CD95L/BV6/zVAD-fmk (Figs. 2, 3). For each cell line, we selected the time interval, corresponding to the initial detection of pRIPK1 in total cellular lysates, assuming that these conditions would correspond to the assembly of the active necrosome complex, *e.g.* shortly after activation of RIPK1. After co-treatment with CD95L/BV6/zVAD-fmk, the necrosome was immunoprecipitated using anti-pRIPK1, anti-RIPK1, anti-FADD, anti-FLIP or anti-caspase-8 antibodies (Figs. 2, 3). Indeed, pRIPK1 was detected in these immunoprecipitations (IPs), indicating the assembly of the active necrosome complex in these experiments. Besides pRIPK1, the other core components of the necrosome such as RIPK1, FADD, procaspases-8, -10 and c-FLIP were detected in these IPs (Figs. 2, 3). Due to the aforementioned lack of optimized antibodies, RIPK3 was detected at very low levels, if at all, in these experiments. Eventually, the used anti-RIPK3 antibodies showed the specificity against RIPK3 in the analysis of cellular lysates (Fig. EV2F). However, these antibodies showed a low sensitivity for RIPK3 detection in the IPs, which may be due to overlap of the anti-RIPK3 antibodies's epitope with the RIPK3 domains interacting with the other proteins in the necrosome as well as the overlap of the signal with the heavy chain of the immunoprecipitating antibodies. The latter was probably also a reason for the poor detection of MLKL in these IPs, as the signal for pMLKL and MLKL is likely to overlap with a signal from the antibody's heavy chain.

Part of Figure EV3 for the reviewer (Former 5D): HT29 cells were pretreated for 1 h with 5 μ M BV6, 50 μ M zVAD-fmk and 10 μ M Nec-1s and subsequently treated with 500 ng/ml CD95L. MLKL-IPs were carried out using anti-MLKL antibodies and analyzed by Western Blot. Abbreviations: s.e.-short exposure, l.e.-long exposure, IP-immunoprecipitation, BC-beads only control pulldown, *-unspecific band, IgG_H-the heavy chain of antibody. The changes are highlighted in red.

With regard to **MLKL-IP** as a way to immunoprecipitate necrosome, we suggest that the IP *via* MLKL has a **low efficiency**, as it may also co-immunoprecipitate the oligomerized MLKL from the cellular membrane, which lowers the numbers of the core component of the complex in this IP. This also might explain the lower number of the core necrosome components detected in the MLKL IP. Please, see the corresponding two fragments of text:

p.8:

MLKL-, pRIPK3- and RIPK3-IPs also led to the pulldown of some core necrosome components from HT29, Jurkat 282 and SUIIT-020 cells upon CD95L/BV6/zVAD-fmk stimulation (Appendix Figs. S1 and S2). However, these IPs were less efficient compared to pRIPK1-IP. The latter may be related to the overlapping binding sites of the antibodies and the necrosome core components, as well as the lack of optimized antibodies for RIPK3, as pointed out by others (Samson *et al.*, 2021). In addition, MLKL-IP may largely immunoprecipitate oligomerised MLKL at the later stages of necroptosis, when it is already released from the necrosome complex and therefore this particular IP may not result in

efficient pulldown of the core necrosome components (Garnish *et al.*, 2021; Meng *et al.*, 2021; Meng *et al.*, 2023) .

Text for the description of Mass-spectrometry experiments, p. 12:

The absolute numbers of core necrosome components in MLKL-IP were in general lower than in the other three IPs (Figs. 4D, 5C). This fits to the hypothesis that anti-MLKL antibodies may largely immunoprecipitate oligomerised MLKL at the later stages of necroptosis, when it is already released from the necrosome complex and therefore this particular IP may not result in efficient pulldown of the core necrosome complex (Garnish *et al.*, 2021; Meng *et al.*, 2021; Meng *et al.*, 2023).

Comment 3

Overall, the amount of data presented here is heroic, however as presented it will require the reader to be equally heroic to read it in depth. I would suggest moving a significant amount of the main figure panels to supplemental data - for instance, the first two figures while important to establish the model (i.e. these cells die via necroptosis) could be moved to supplemental.

Answer 3

We fully agree with the reviewer and have moved most of the main Figures 1 and 2 to the **Supplementary Figures (EV Figures)** to make the overall reading easier. We have kept only the key experiments, *e.g.* showing that these cells die by necroptosis in Figure 1. We have also moved some of the IPs experiments from the former Figures 3-5 to the **Appendix Supplementary Figures**. In this way, Figures 1-5, which preceded the mass spectrometry analysis, have been shortened to three figures, currently Figures 1-3.

We would like to thank reviewer 1 for taking the time to read this manuscript and important critical remarks, which allowed us to improve the quality of our work.

Referee #2 (Report for Author)

General Remarks

General Remark 1

This study looks at the necrosome the key complex driving necroptosis. While some of the data with regard to the RIPK1 mutants could be of value to the field, far too much of the manuscript is devoted to basic experiments that recapitulate much of what is already known and doesn't thoroughly analyse the RIPK1 mutants themselves. I'd advise the authors to review Geserick et al, 10.1083/jcb.200904158, their detailed analysis of caspase-8 IPs in the presence of CD95L, smac-mimetic and Z-VAD-FMK are directly relevant.

Answer 1

We fully agree with this criticism. We have moved a number of panels from the former Figures 1-5, to the **Supplementary Figures (Expanded View (EVs) and Appendix Supplementary Figures**, e.g. establishment of the model for CD95L/BV6/zVAD-fmk-mediated necroptosis and necrosome assembly). We had these experiments presented in the former Figures 1-5 because of the previous revision, where the reviewers explicitly asked us to show the time-dependent assembly of the necrosome and induction of necroptosis in different cell lines. However, we also agree this was not the optimal way of starting the manuscript. We hope that the reviewer is satisfied with our changes and shortening the introductory figures.

We expanded **the part devoted to type II DD interactions** of FADD and RIPK1 in the necrosome assembly. To this point, we have implemented RIPK1- and FADD- deficient HeLa cells overexpressing RIPK3, which allowed us to further expand this part. Please, see more details on this below.

We are also **delighted to cite** the paper **by Geserick et al.** in the current version. We apologise for citing only Feoktistova et al, 2011, the publication from the same group in the previous version, as the term 'ripiptosome' was introduced in the Feoktistova et al, 2011 paper, but of course we should have cited Geserick et al, as complex II formed upon CD95L/IAP antagonist stimulation was first described in this paper.

Indeed, in 2009, Geserick *et al.* showed that stimulation with the CD95L/IAP antagonist leads to the formation of complex II, which consists of DED proteins and RIPK1 and does not contain CD95. The authors showed that this complex can induce both apoptosis and necroptosis pathways, the latter following the addition of zVAD-fmk and the presence of RIPK3 in the particular cell line. They also characterised the role of c-FLIP proteins in controlling the function of this complex. This was a truly pioneering study. Later, there was another manuscript from the same group, Feoktistova et al, 2011, which we cite extensively in our previous version of the manuscript. This manuscript further investigated the role of IAP antagonist-induced complex II in apoptosis/necroptosis and gave this complex the name: ripiptosome. Hence, we cite this 2011 manuscript a lot because of the introduction of the name 'ripiptosome', as mentioned above, although we fully agree that we should have cited 2009 manuscript by Geserick *et al.* because this was the first description of CD95L/IAP antagonist-induced complex II. But, of course, we cite Geserick *et al.*, 2009 in the revised version at many key positions giving its well-deserved pivotal role.

General Remark 2

Summary: Fig. 1 performs a Western blot analysis of necroptosis markers in 3 cell lines as well as a cell death analysis Fig. 1F -H. Fig. 2 looks at varying some of the parameters to find the optimal cell death conditions, for example time of CD95L or Nec-1. Fig. 3 performs FADD, (p)RIPK1 and caspase-8 IPs in the SUIT-020 line that overall confirms that these proteins are in a complex, possibly together with RIPK3 although the reagents for RIPK3 are not great. Fig. 4 is similar to Fig. 3 but looks at the other cell lines (HT29s, Jurkat) and also extends to MLKL. Since MLKL appears to be immunoprecipitated by the phospho-RIPK1 antibody even in the absence of a necroptotic signal (Fig. 4A lane 1, Fig. 4B lane 6) when the levels of phospho-RIPK1 should be negligible, this does raise concerns about the specificity of the IP in general and the data certainly can't be used to conclude that MLKL is in the complex (as in page 9 of the text). Fig. 5 is similar to Fig. 4 using (p)RIPK3 and MLKL IPs. While there is a suggestion of specificity, mostly with caspase-8, on the whole there is a lack of difference between IPs {plus minus} BCZ treatment. Therefore the main conclusion possible from this figure is that RIPK3 and phospho-RIPK3 antibodies are not very good tools to explore the necrosome (and might therefore be better as a supplementary figure).

Answer 2

We thank the reviewer for pointing out important issues in the presentation of our IPs. Indeed, we had **several non-specific signals in the IPs** from unstimulated cells shown in the previous version of the manuscript and we thank the reviewer for pointing this out. This, in particular, concerns the IPs with anti-RIPK3, anti-pRIPK3 and anti-MLKL antibodies. Moreover, as already commented in the answers to reviewer 1, there are many bands around 50 kDa, which correspond to the heavy chains of the immunoprecipitating antibody and at the same time could indicate the presence of the specific proteins if they have a molecular mass around 50 kDa, which is the case for both RIPK3 and MLKL. In the current version of the manuscript, we have paid particular attention to the correct reporting of signals corresponding to the antibody's heavy chain, non-specific signals and specific signals. We have indicated for for RIPK3, pRIPK3 and MLKL in the pRIPK1-IPs that those are likely the heavy chains of the antibodies. We fully agree that we were not perfect in this respect in the previous version of the manuscript and we hope that the reviewer will be satisfied with the current version.

We critically reviewed **our sentencin on MLKL** presence in the IPs through the text and we apologize for mentioning it in a way as reviewer highlighted, when describing the Western Blot analysis of pRIPK1-IPs. Indeed, in these IPs, we have a strong signal from the immunoprecipitated antibodies, which likely overlaps the signal of MLKL. Hence, we comment on it in the text:

p.6:

To initiate the necrosome assembly, the investigated cell lines were treated with CD95L/BV6/zVAD-fmk (Figs. 2, 3). For each cell line, we selected the time interval, corresponding to the initial detection of pRIPK1 in total cellular lysates, assuming that these conditions would correspond to the assembly of the active necrosome complex, *e.g.* shortly after activation of RIPK1. After co-treatment with CD95L/BV6/zVAD-fmk, the necrosome was immunoprecipitated using anti-pRIPK1, anti-RIPK1, anti-FADD, anti-FLIP or anti-caspase-8 antibodies (Figs. 2, 3). Indeed, pRIPK1 was detected in these immunoprecipitations (IPs), indicating the assembly of the active necrosome complex in these

experiments. Besides pRIPK1, the other core components of the necrosome such as RIPK1, FADD, procaspases-8, -10 and c-FLIP were detected in these IPs (Figs. 2, 3). Due to the aforementioned lack of optimized antibodies, RIPK3 was detected at very low levels, if at all, in these experiments. Eventually, the used anti-RIPK3 antibodies showed the specificity against RIPK3 in the analysis of cellular lysates (Fig. EV2F). However, these antibodies showed a low sensitivity for RIPK3 detection in the IPs, which may be due to overlap of the anti-RIPK3 antibodies's epitope with the RIPK3 domains interacting with the other proteins in the necrosome as well as the overlap of the signal with the heavy chain of the immunoprecipitating antibodies. The latter was probably also a reason for the poor detection of MLKL in these IPs, as the signal for pMLKL and MLKL is likely to overlap with a signal from the antibody's heavy chain.

For the evidence for the presence of MLKL in the CD95L/BV6/zVAD-induced necrosome, we present quantitative mass spectrometry analysis of **MLKL-IP** (Fig. 4 and 5), which shows the presence of RIPK1, RIPK3, FADD, c-FLIP and caspase-8 in MLKL-IP upon CD95L/BV6/zVAD stimulation. This data set fully confirms the presence of MLKL in this complex, as in the absence of MLKL in complex, the necrosome components cannot be detected in the MLKL-IP. We very much hope that the reviewer will find this data set convincing. Please, see the results of quantitative mass spectrometry analysis of MLKL-IP below.

Part of the Figure 5 for the reviewer. HT29 cells were pretreated for 1 h with 5 μ M BV6 and 50 μ M zVAD-fmk and subsequently treated with 500 ng/ml CD95L for 5 h 6 h. The MLKL-IPs were carried out using anti-MLKL antibodies. The IPs were analyzed by AQUA mass spectrometry analysis. The amount of proteins detected is shown in fmols.

Furthermore, we suggest that the necrosome IP **via MLKL** has a **very low efficiency**, in addition to the sub-optimal quality of the antibody, as it may also co-immunoprecipitate the oligomerised MLKL from the cell membrane. We comment on this in the text. Please, see two new paragraphs:

p.8:

MLKL-, pRIPK3- and RIPK3-IPs also led to the pulldown of some core necrosome components from HT29, Jurkat 282 and SUIT-020 cells upon CD95L/BV6/zVAD-fmk stimulation (Appendix Figs. S1 and

S2). However, these IPs were less efficient compared to pRIPK1-IP. The latter may be related to the overlapping binding sites of the antibodies and the necrosome core components, as well as the lack of optimized antibodies for RIPK3, as pointed out by others (Samson *et al.*, 2021). In addition, MLKL-IP may largely immunoprecipitate oligomerised MLKL at the later stages of necroptosis, when it is already released from the necrosome complex and therefore this particular IP may not result in the efficient pulldown of the core necrosome components (Garnish *et al.*, 2021; Meng *et al.*, 2021; Meng *et al.*, 2023).

p. 12:

The absolute numbers of core necrosome components in MLKL-IP were in general lower than in the other three IPs (Figs. 4D, 5C). This fits to the hypothesis that anti-MLKL antibodies may largely immunoprecipitate oligomerised MLKL at the later stages of necroptosis, when it is already released from the necrosome complex and therefore this particular IP may not result in efficient pulldown of the core necrosome complex (Garnish *et al.*, 2021; Meng *et al.*, 2021; Meng *et al.*, 2023).

We have also moved the RIPK3-IP, pRIPK3-IP and MLKL-IPs to the Appendix Supplementary figures 1 and 2 following the reviewer's comments and we fully agree that antibodies used in these IPs were not good tools for this analysis.

We hope that the reviewer will be satisfied with the quality of the IPs using anti-pRIPK1, FADD, Caspase-8 and c-FLIP antibodies, shown in the current Figures 2 and 3, which show co-immunoprecipitation of the necrosome components.

General Remark 3

Fig. 6 uses quantitative Mass Spectrometry to look at the levels of RIPK1, FADD and RIPK3. Basically it shows that there is more FADD and RIPK1 in the complex than RIPK3. Since there are different ratios of FADD and RIPK1 depending on whether FADD or RIPK1 is pulled down whereas if there was a single complex one would expect ratios to stay the same, it's very unclear what solid quantitative conclusions we can draw from this data.

Answer 3

We appreciate the reviewer's comment. We have added the specific comments to the text, as our quantification strategy was indeed not well explained in the previous version of the manuscript.

In our quantitative assessment, we considered the following points:

1. When we are using the antibody against one protein as a bait, we do not consider this protein for quantification as we cannot exclude that also the proteins not bound to the complex might be co-immunoprecipitated. The only exception is pRIPK1-IP as in this case we perform high

molecular weight fractionation and show that all pRIPK1 is present only in the highmolecular weight complex and not in the low molecular weight fractions.

2. Despite this rule we still, indeed, have different ratios of RIPK1 compared to RIPK3 and FADD in different IPs. Importantly, the same tendency of more RIPK1 in the complex compared to RIPK3 was observed using immunoprecipitation with different antibodies, e.g. anti-pRIPK1, anti-FADD, anti-MLKL and anti-FLIP. However, the absolute values are different between these IPs as the antibodies recognise different parts of the complex and epitopes for some antibodies may be masked by the other components of the complex leading to less efficient pulldown. Hence, the next factor contributing to the amounts of the protein in the complex as judged by IP: the antibody used for the particular IP, as the latter may interfere with the binding sites of the proteins in the complex
3. Finally, the abundance of the protein in the same complex may differ between different cell types due to the different abundances of the protein in these cells.

We added the specific comments to the text, as this was indeed not well explained in the previous version of the manuscript. Please, see **the new text describing RIPK1 and RIPK3 ratio results** from mass spectrometry:

Page 11:

The quantitative mass spectrometry analysis of pRIPK1-IPs from SUIT-020 and HT29 cells revealed that RIPK3 is present in the lower amounts compared to RIPK1 (Figs. 4B, EV3E). As discussed above, we assumed that pRIPK1 was only present in the necrosome complex, which was supported by HMW fractionation experiments. Therefore, we suggested that RIPK1 amounts obtained from pRIPK1-IPs can be considered for estimating its ratio to RIPK3. Furthermore, these results were supported by quantitative mass spectrometry analysis of FADD-, c-FLIP- and MLKL-IPs, which also showed the lower levels of RIPK3 compared to RIPK1 (Fig. 4C-E, EV3F). It should be noted that the absolute numbers of RIPK1 detected in these IPs differed between the four IPs, probably due to the unequal efficiency of the immunoprecipitating antibodies. The latter might be because some epitopes for the immunoprecipitating antibodies might be interfering with protein-protein interactions in the complex. However, in all these experiments, the higher numbers of RIPK1 compared to RIPK3 were detected, strongly suggesting that RIPK1 is the major core component of the necrosome.

Page 12:

The levels of FADD in the pRIPK1-IPs from HT29 and SUIT-020 cells were lower than those of RIPK1 (Figs. 5A, EV3E,F, EV4B). Similar to pRIPK1-IPs the absolute numbers of FADD differed between the

pRIPK1-, c-FLIP- and MLKL-IPs, probably due to the unequal efficiency of the immunoprecipitating antibodies and also due to the fact that some epitopes might be interfering with interactions of FADD with the other proteins in the complex (Fig. 5). The number of FADD molecules in FADD-IP was much higher than the amounts of other necrosome components, as anti-FADD-IP co-immunoprecipitates all cellular FADD. Hence, FADD numbers from FADD-IPs cannot be used to make quantitative assumptions. Similarly, we assumed that absolute numbers of c-FLIP and procaspase-8 from FADD- and c-FLIP-IPs could not be used for quantification (Figs. 5B, D and EV4B) because c-FLIP was used as a bait in c-FLIP-IP and these IPs may contain proteins from procaspase-8/c-FLIP or procaspase-8/c-FLIP/FADD complexes formed independently of DL stimulation (Golks *et al*, 2006; Yang *et al*, 2024).

In addition, we would like to underline that the main message of the manuscript is that **RIPK1 aggregation** drives the activation of the complex. We believe that this is the fundamental finding, e.g. that aggregates or filaments drive a particular process, like DED filaments drive CD95 DISC activation or protein aggregation drives phase separation. In addition, we argue that there are many possible stoichiometries and that RIPK1 oligomer composition is very dynamic. We draw this conclusion from both mass spectrometry and AlphaFold modelling experiments. Therefore, our main finding here is that RIPK1 aggregates form the core structure that drives necrosome activation, which is further emphasised in the manuscript text. Specifically, we added the paragraph to the Discussion on this:

We show that the major component of the CD95/BV6/zVAD-fmk-induced necrosome is RIPK1, which forms the core of the complex, and that RIPK1 aggregation drives necrosome activation. Importantly, AlphaFold modeling indicated that necrosome assembly is likely to be a dynamic process with multiple possible stoichiometries involving the formation of RHIM and DD oligomers. The particular stoichiometry may be determined by the strength of stimulation, the endogenous expression of key necrosome components and their post-translational modifications. This type of complex assembly will be consistent with other oligomeric structures such as DED filaments, for which mathematical models have shown that the length of the filaments is strongly dependent on the abundance of the core components of the DISC complex in particular cell type and the stimulation strength (Schleich *et al.*, 2012).

Importantly, the role of proper ratios of RIPK1 and RIPK3 for cell death induction in cancer cells have been shown recently in the study using state of the art Protac technology, which allowed to trigger cell death in cancer cells. We think this study is an important evidence for the proper stoichiometry of the necrosome of the cell death induction.

Please, see the new text in the Discussion:

Finally, the role of the balance between RIPK1 and RIPK3 in inducing cell death in cancer cells has recently been highlighted by the study using Protac targeting RIPK1, which was able to induce cell death in cancer cells (Mannion *et al*, 2024). This study provides important evidence for the correct stoichiometry of RIPK1 and RIPK3 in the necrosome to drive cell death induction.

General Remark 4

The RNA levels (Fig. 6E, F), while no doubt of interest to someone, are peripheral to this figure since level of mRNA doesn't reliably correlate with protein level.

Answer 4

We appreciate the reviewer's comment. We have moved mRNA levels to the supplementary Figures (currently EV3G) as we fully agree that level of mRNA does not reliably correlate with the protein level.

General Remark 5

Fig. 7 uses AlphaFold to predict the structure of a "minimal" necrosome containing a FADD and RIPK1 DD. Since it contains only the DDs of these proteins and relies to some extent on the stoichiometry data generated in Fig. 6 to choose which structure is most accurate, it fails to convince.

Answer 5

In the time of revision AlphaFold3 was released that allowed to include RHIM-containing and DED-containing proteins to the model of the necrosome. Please, see the model of the necrosome generated using AlphaFold3 below and in the new Figure 9E, F:

Parts of Figure 9. Necrosome model reconstructed with AlphaFold3. RIPK1 is shown in gold. RIPK3 is shown in red. Procaspase-8 is shown in dark blue. c-FLIP is shown in light blue. For simplicity we didn't include the kinase domains of the RIPK1 and RIPK3 proteins.

AlphaFold (AlphaFold3) offers more precise reconstruction of molecular complexes. The previous

version, AlphaFold2, was limited to predicting only the structure of the DD oligomer complex. In contrast, AlphaFold3 can now predict not only the DD oligomer but also the structures of **RHIM amyloids** and DD and **DED oligomers** within the necrosome complex. This significantly enhances the quality of our model and its utility in advancing our understanding of the molecular mechanisms underlying necrosome assembly. The updated model further validates the findings presented in our previous submission while providing new data, which are well-supported by experimental and mutagenesis studies. Overall, the AlphaFold3 model of the necrosome was validated by mutagenesis data of DD domains, clinically relevant mutations, and the MS-derived stoichiometry presented in this manuscript.

The DD stoichiometry of FADD and RIPK1 still plays the key role in the assembly of this complex therefore this part is presented more in detail and validated by the mutants. Here predictions are made by AlphaFold2 (Current Figure 6 and EV4C, D). Importantly, AlphaFold3 predictions were fully fitting to AlphaFold2 with regard to the structure of the DD part of the complex. However, we have largely rewritten this part of the manuscript and we also added modeling of RHIM domain part of the necrosome complex to this experimental chapter. This modeling chapter is followed by the chapter describing the generation of the mutants of DD in FADD and RIPK1 interaction domains and their functional validation. The full model of the necrosome generated by AlphaFold3, shown above, is presented at the Discussion section.

General Remark 6

Fig. 9 is where the manuscript becomes more interesting as it tests the predicted mutants, however the cell viability assays do not sufficiently investigate the effect of the mutations on complex formation etc to prove that signalling has been affected in the predicted manner. i.e. earlier stoichiometry and IP experiments are needed to validate.

Answer 6

In the revised version of the manuscript, we have used the model of HeLa cells overexpressing RIPK3 (HeLa-RIPK3 cells), which is a very good experimental model for performing transfection followed by biochemical analysis. We also used FADD- and RIPK1-deficient HeLa-RIPK3 cells to introduce FADD and RIPK1 mutants, respectively. Measurement of cell viability after introduction of the mutants into these cell lines also demonstrated the lack of sensitivity to CD95L/BV6/zVAD-fmk treatment. We have performed biochemical analyses in these HeLa RIPK3 cells and have shown that upon introduction of a particular mutation there is a reduction in the phosphorylation of RIPK1 and MLKL, indicating a reduction in necrosome formation, as shown in the new Figure EV5. Please see below a panel from Figure EV5. It should be noted that since the introduction of mutants did not lead to a loss of cell viability upon CD95L/BV6/zVAD stimulation and phosphorylation of MLKL and RIPK1, in contrast to WT, we did not continue with IPs as the induction of necroptosis was blocked.

We have also significantly expanded the text of this chapter, as it was rather short in the previous version. We present this text below in response to specific comment 2.

Please see the new text and Figures 8, 9 and EV5.

Please also see the part of Figure EV5 below:

Figure EV5 for the reviewer. HeLa RIPK1 ko RIPK3 cells were transfected with empty vector, RIPK1 wt or RIPK1 AD (M637A, I641D). Transfected cells were pretreated with 5 µM BV6 and 50 µM zVAD-fmk for 1 h and afterwards stimulated with 500 or 1000 ng/mL CD95L for 3 h. Total cellular lysates were analyzed using Western Blot with the indicated antibodies. Actin served as loading control. One representative Western Blot out of three is shown. Phosphorylation of RIPK1 and MLKL upon introduction of wt is highlighted in red.

Specific Remarks

1.

Fig. 8 legend states, "Residues with somatic mutations observed located on type II protein-protein interaction interfaces are indicated in cyan, while other residues with somatic mutations observed are shown in cyan." I think one of these should be green.

1. Answer. This was corrected.

2.

Fig. 9D & E Were RIPK1 mutants used in these panels? The legend doesn't say although the text on page 14 says "the introduction of mutations into RIPK1 inhibited CD95L/BV6/zVAD-fmk-induced cell death". If mutants were not used I fail to see the logical connection to the rest of this figure which is about the point mutants. If they were used I do not see the positive control of death with wild type RIPK1.

2. Answer. We thank the reviewer for pointing this out. Please, see the new text, page 17:

To validate the key role of type II interactions in the necroptosis induction several amino acid residues of RIPK1 and FADD involved in the interaction of IIa and IIb DD interfaces were mutated. In particular, RIPK1 ((M637A/I641D) termed RIPK1-AD), FADD ((L172D/D175R/L176S) termed FADD-DRS) and FADD ((M170A/L172D/D175R/L176N), termed FADD-ADRN) mutants were constructed (Fig. 6C). The mutations were designed to inhibit type II DD protein-protein interactions as well as preserve the stability of the proteins, *e.g.* of FADD and RIPK1, according to estimation based on Rosetta scoring function.

RIPK1-AD was reconstituted into Jurkat RIPK1 ko (Figs. 8B, C and EV5A-C), while FADD-DRS and FADD-ADRN were introduced into FADD-deficient Jurkat (Jurkat FADD ko) (Fig. 8D). Furthermore, FADD- and RIPK1-deficient HeLa cells were also used for testing the effects of FADD-DRS, FADD-ADRN and RIPK1-AD (Figs. 9A-D and EV5D-G). Specifically, the constructed mutants were reconstituted into RIPK1-deficient RIPK3-overexpressing HeLa (HeLa RIPK1 ko RIPK3) and FADD-deficient-RIPK3-overexpressing HeLa (HeLa FADD ko RIPK3) cells (Figs. 9A-D and EV5D-G). WT-RIPK1 and WT-FADD were also reconstituted into these RIPK1- and FADD-deficient cells, respectively.

The loss of a cell viability upon CD95L/BV6/zVAD-fmk co-stimulation was observed in parental Jurkat A3 cells (Fig. 8B) as well as upon reconstitution of WT-RIPK1 and WT-FADD into RIPK1- and FADD-deficient cells (Figs. 8C, D and 9A-D). Addition of Nec-1s to these cells rescued the CD95L/BV6/zVAD-fmk-induced loss of their viability (Figs. 8B, C and 9A-D). Upon reconstitution of RIPK1-AD or FADD-DRS or FADD-ADRN into RIPK1- or FADD-deficient cells, CD95L/BV6/zVAD-fmk-induced loss of their viability was inhibited (Figs. 8C, D, 9A-D). This was observed upon both short-term and long-term CD95L/BV6/zVAD-fmk stimulation (Fig. 9A-D). Importantly, CD95L alone stimulation led to a loss of cell viability in both Jurkat RIPK1 ko and HeLa RIPK1 ko RIPK3 cells, but not in FADD-deficient cells (Figs. 8, 9A-D and EV5A-C). This is fully consistent with previous reports that CD95-mediated apoptotic cell death is FADD-dependent but not RIPK1-dependent (Abramson *et al.*, 2024; Feoktistova *et al.*, 2011; Geserick *et al.*, 2009). Reconstitution of RIPK1-AD and FADD-DRS/FADD-ADRN into HeLa RIPK1 ko RIPK3 and HeLa FADD ko RIPK3 cells, respectively, was also accompanied by the reduction in phosphorylation of RIPK1 and MLKL (Fig. EV5D-G). These results strongly indicate that introduction of mutations into RIPK1 and FADD, specifically disrupting type II DD interactions, leads to the inhibitory effects on necrosome assembly and necroptosis induction. Taken together, the designed mutations provide support for the suggested structural model of the necrosome and support the role of type II interactions of DDs in the necrosome assembly.

We would like to thank reviewer 2 for taking the time to read this manuscript and his excellent critical remarks, which allowed us to improve the quality of our work.

Referee #3 (Report for Author)

Ivanisenko, König, Hillert-Richter and colleagues present a really interesting and well formulated dissection of the chronology of FasL signaling and the componentry of the necrosomes underpinning induction of necroptotic death. FasL is poorly characterized as a necroptotic ligand and this study fills a massive void in knowledge. The authors take quantitative data, validating blots, and apply contemporary protein modelling approaches to understand complex assembly. The strength is in the use of multiple cell lines, multiple time points, and validating the models with mutational studies.

The work is extensive and is far more substantive than their prior submission, which I feel merits publication. The only major shortcoming for me is that the authors have not used their IP/MS approach to report new possible necrosome components (or MLKL) even though they are likely present in their existing datasets. On upload to an appropriate database for inspection (i.e. PRIDE), these will be revealed, so why not discuss here? There's an opportunity here for them to elaborate on the identified interactors in the AQUA MS, their stoichiometries, and contextualize how they might contribute to necroptotic signaling/necrosome stability or assembly. I do not expect them to validate new interactions using an orthogonal method as this is already a massive study, but I would appreciate some commentary about what other proteins they identified. Here I make a number of suggestions for additional analyses and data presentation, but do not expect further experiments.

Answer

We thank the reviewer for appreciating our work and we are very grateful for further very helpful, great and thoughtful comments. We would just like to point out that we still had to do further experiments despite the comments of the reviewer. Since the mass spectrometry analysis for identification of the interaction partners was not done in the previous version of the manuscript. Specifically, in the previous version, we only did quantitative AQUA peptide mass spectrometry analysis of our immunoprecipitations, which does not involve the protein identification. Hence, in the course of revisions, we performed **interactome analysis**, which is presented in the current **Figure 4A**. We hope that the reviewer will appreciate our efforts and once again, we would like to thank the reviewer for the very valuable comments.

Major comments

1. Please ensure the MS dataset is uploaded to the PRIDE database. Please comment on other components beyond those blotted that appear in the IPs. Whether they are novel or not is immaterial, they'd be important data for the field. Especially interesting if there are FasL specific interactors, although this may be fantasy. If there are, however, this would add enormously to the citability of this work. I'm especially interested to know if ZBP1 popped up in these interactomes. This is a highly topical area and would benefit the authors to discuss regardless or not of occurrence in their necrosome IPs.

Answer 1.

We have uploaded the Mass spectrometry data into **Pride database**. Please, see the corresponding citation in the text, page 31:

.. Mass spectrometry are available via ProteomeXchange with identifier PXD059873...

As highlighted above, in the previous version we did not analyse the interactome of these IPs, therefore we performed the additional mass spectrometry analysis requested by the reviewer in the

current round of manuscript revisions. We present these data in Figure 4a and source data. We did not detect ZBP1 in this analysis. We mention this fact in the text. We got all the core components of the necrosome in these pulldowns, which supports the specificity of these IPs for necrosome analysis. The other proteins of the cell death pathways were found at low levels in those IPs. We also performed the analysis using the GO database of all mass spectrometry hits and found that the components of the necrosis pathway have a high relevance in this analysis (Fig. EV4D and source data).

Please, see below the new Figure 4a, as well as corresponding manuscript text, page 9:

Figure 4A. The proteins identified in FADD-, FLIP- and pRIPK1-IPs. Heat Map from FADD IP, c-FLIP IP and pRIPK1 IP after qualitative mass spectrometry analysis. HT29 cells were pretreated for 1 h with 5 μM BV6 and 50 μM zVAD-fmk and subsequently treated with 500 ng/ml CD95L for 5 h, which was followed by IPs. CD95/BV6/zVAD-fmk (CBZ)-treated and untreated IP samples were analysed by mass spectrometry. Representative proteins are shown for CBZ-treated cells (FADD IP/CBZ, FLIP IP/CBZ, pRIPK1/CBZ) and untreated cells (FADD IP/medium control, FLIP IP/medium control, pRIPK1 IP/medium control, and beads-only control). Necrosome components are highlighted in red color. The full list of detected proteins can be found in the source data for Figure 4A.

Text, page 9:

Further evidence of the efficiency of the IPs used for necrosome co-immunoprecipitation was obtained by mass spectrometry analysis of the interactome in pRIPK1-, FADD- and c-FLIP-IPs (Fig. 4A). Importantly, all core components of the necrosome such as procaspase-8, c-FLIP, FADD, RIPK1 and RIPK3 were detected in these analyses upon CD95L/BV6/zVAD-fmk stimulation, further supporting the specificity of these approaches for necrosome co-immunoprecipitation (Fig. 4A). CD95 or CD95L were not detected in these IPs, suggesting that upon CD95L/BV6/zVAD-fmk co-treatment the complex II presents the most abundant complex among the CD95-induced cellular complexes (Fig. 4A). In addition to the core components of the necrosome, highly abundant and so-called 'sticky' proteins were detected in this screen with the high scoring, such as heat shock proteins (HSP7C, HS90B, HS90A), chaperones (BiP), cytoskeletal proteins (ACTG, ACTB, TBB4B, TBB5, MYH9, MYH14), components of peroxiredoxin family (PRDX1), DNA repair machinery (PRKDC) and RNA binding proteins (NONO, DDX5, DDX6, DDX3X, YBOX1, MSI2H, TCOF) (Fig. 4A). Importantly, most of these proteins were found in the beads-only sample supporting their non-specific character of binding. There were also several proteins detected in the IPs upon CD95L/BV6/zVAD-fmk stimulation that were reported to play a role in cell death pathways such as TRAP1, VDAC2, HIPK2/3, BIRC6, JAK2 albeit with a very low scoring (Fig. 4A). Interestingly, some potential interactors such as ZBP1 were not detected (Imai *et al*, 2024). Bioinformatic GO analysis also confirmed the presence of a statistically significant group of proteins associated with cell death pathways, and specifically the necroptosis network had the high relevance in this analysis (Fig. EV3D). This further supports the specificity of this approach for necrosome IP and confirms that these IPs can be used to pull down the necrosome complex.

2. Citations. The key references are listed for the necroptosis pathway in the second last paragraph of page 3, although over the past 4 years, knowledge has moved enormously on the underlying mechanism of MLKL activation. I encourage the authors to contemporize their reference list in the intro. In Discussion on page 16, Orozco et al. is one of many that have contributed to knowledge of the apoptosis-necroptosis switch. This was published in concert with other papers and there are others from, for example the Vucic lab, that add to our understanding on the chronology of PTMs and signaling events.

Answer 2.

We thank the reviewer for this comment. We have added the corresponding citations to the manuscript text and discuss them. Please, see the new text.

Specifically, we largely, increased citations describing **MLKL activation** in Introduction. The list of citations includes the following manuscripts:

(Alvarez-Diaz *et al*, 2016; Czabotar & Murphy, 2015; Davies *et al*, 2024; Li *et al*, 2012; Meng *et al*, 2021; Meng *et al*, 2023; Newton *et al*, 2019; Samson *et al*, 2020; Sun *et al*, 2012; Wang *et al*, 2014; Yuan & Ofengeim, 2024).

For **MLKL citations**, we have specifically added: (Garnish *et al.*, 2021; Meng *et al.*, 2021; Meng *et al.*, 2023)

Regarding the citation on page 16, Orosco *et al*, the reviewer is right that there are many more papers that need to be cited here, which we have done in the current version. However, here we are discussing the studies using overexpression of RIPK3, so the excellent work from the Vucic lab on the chronology of PTMs and signalling events at the TNF-induced necrosome is cited not here but in the part of the Discussion where we discuss ubiquitinylation/deubiquitinylation as the driving force of the necrosome and dynamic features of the necrosome composition :

Text, page 22:

Importantly, ubiquitinylation status of RIPK1 is a pivotal step, required for the formation of the necrosome (**de Almagro *et al.*, 2015**; Feltham & Silke, 2017; Li *et al*, 2020). Therefore, the priming of necrosome formation by deubiquitylation of RIPK1 by BV6, leading to the assembly of RIPK1/FADD oligomers upon CD95L stimulation, also plays a key role in our model. However, the structural background of how deubiquitinylation of RIPK1 promotes the formation of RIPK1/FADD oligomers in the course of the CD95-mediated necroptosis induction has to be delineated in the future studies. One of the molecular mechanisms might involve the steric hindrance of DD interaction interfaces due to polyubiquitinylation and/or subsequent conformational changes. Indeed, three residues within the RIPK1 DD have been reported to undergo ubiquitination in mouse RIPK1 (Hou *et al*, 2024), and these residues are conserved in the human RIPK1 sequences (K604, K627, and K648). Specifically, the ubiquitination of K648 and K627 may influence type I and other DD interactions within the necrosome, according to our model (Fig. 7B).

In addition, we cite the excellent work from the Vucic lab in the other part of the discussion when discussing the composition of the necrosome:

Text, page 19:

We show that the major component of the CD95/BV6/zVAD-fmk-induced necrosome is RIPK1, which forms the core of the complex, and that RIPK1 aggregation drives necrosome activation. Importantly, AlphaFold modeling indicated that necrosome assembly is likely to be a dynamic process with multiple possible stoichiometries involving the formation of RHIM and DD oligomers. The particular stoichiometry may be determined by the strength of stimulation, the endogenous expression of key necrosome components and their post-translational modifications (de Almagro *et al*, 2015; Hoblos *et al*, 2024).

3. The idea of RIPK1 as the nucleating force in human necrosomes has been proposed in the period between the last submission and this one - e.g. Jacobsen CDDis and Pierotti Biochem J. This could be suitably acknowledged to reflect the idea that pRIPK1 is a scaffold for these complexes in human cells. I feel it is also worth noting this is probably peculiar to human over mouse cells, which is the focus of the submission in question.

Answer 3.

We added the corresponding citations to the manuscript text, discuss them and make the corresponding notations. Text, page 20:

Overall our findings suggest that RIPK1 forms the major structural scaffold of the necrosome, just as caspase-8 forms the major core at the DED filaments (Dickens *et al.*, 2012; Schleich *et al.*, 2012), and all other components are recruited to RIPK1 homooligomers (Fig. 10). In this way, RIPK1 oligomer forms the structural platform for the assembly of this complex. It has to be noted that the possibility for RIPK3 homotypic interactions was suggested before by a number of biochemical studies (Mompean *et al.*, 2018; Orozco *et al*, 2014). Importantly, our model does not exclude RIPK3 oligomerization within these RIPK1 amyloids, like making the small RIPK3 insertions, but the amount of RIPK1 at the necrosome appears to be higher than RIPK3 (Fig. 10). Furthermore, the hypothesis that RIPK1 serves as a major scaffold for necrosome assembly is consistent with recent reports on the assembly of this complex (Jacobsen *et al*, 2022; Pierotti *et al*, 2023).

Moreover, we cite these two publications when discussing pharmacological targeting of the necrosome in the last paragraph of the discussion:

The structural model of the necrosome core predicted by AlphaFold offers unexpected insights into potential mechanisms of its assembly. It also lays the groundwork for the rational design of small molecules and synthetic proteins aimed at regulating the necrosome complex structure (Jacobsen *et al*, 2022; Pierotti *et al*, 2023).

4. The shortcomings of RIPK3 reagents are duly noted. This has advanced somewhat for monoclonals in the years since first submission, however. I do not feel the authors need to revisit this but I wanted to note that hRIPK3 antibodies from Petrie Cell Rep 2019 and Chiou EMBO Mol Med 2024 have now been vetted for WB and IF/IHC, respectively. I hope that the availability of vetted and ever-improving reagents will overcome the difficulties with detection of hRIPK3 in future studies.

Answer 4.

We have added the citation by Chiou *et al* to the manuscript text. Please, see the new addition:

However, the signals for pRIPK3 were rather weak due to the reagents not yet being optimized (Figs. 1A and EV1A, B) (Samson *et al*, 2021), which is currently being improved (Chiou *et al*, 2024; Petrie *et al*, 2019).

5. Please plot all replicate data points on bar charts.

Answer 5.

We have done the corresponding modifications of the bar charts.

6. In Figure 6, if the kinetics are anything like TNF on HT29 cells, the authors should see MLKL in the necrosome at 3+ hours. It would be of broad interest to compare the abundance of RIPK3 and MLKL in the IPs. Meng Nat Comm 2021 proposed that RIPK3 and MLKL reside in a 1:1 complex in the cytosol under basal conditions. It would be of enormous interest to know how this plays out in the real world inside cells, and quantitative mass spec seems like the ideal opportunity to define this.

Answer 6.

We thank the reviewer for this important point. We have seen background levels of RIPK3 in the unstimulated conditions in the IPs. In fact, there are several bands that support this hypothesis. However, we cannot make this statement based on the quality of our antibodies used for IP analysis by Western Blot, as highlighted in the response to the other reviewers. We also obtained the signal for RIPK3 in the unstimulated conditions in MLKL-IP by mass spectrometry, which would also support this hypothesis, but we feel that without proper Western Blot confirmation we cannot make a strong statement here. We hope the reviewer would understand our point of view here.

Minor

7. The authors have now too few articles in the text, although this not a concern since I am sure the copyeditors/typesetters will fix this

8. Second last line, page 10. "the ones" could be replaced with "those"

9. Discussion. I'm not sure what the statute of limitations is but I feel that saying necroptosis is recently discovered might be an oversell. It is certainly more recent than apoptosis, but I think it'd be more useful to put a time stamp on necroptosis, such as by saying only the past 24 (Holler) or 19 (Degterev) years.

10. I might have missed it, but I'm unclear why antibodies towards mouse RIPK3 and MLKL are included in the methods. My reading (apologies if I've missed it) is that the study is on a bunch of human lines.

11. In Figure 2, there are 2 fonts used for the key and the other labels. I suggest harmonizing.

Answer to 7-11. We fully agree with these comments we fixed these items in the new version of the manuscript and we hope that the reviewer was satisfied.

We would like to thank the reviewer for constantly checking our manuscript and making brilliant suggestions for improvement. We hope we have answered all the points.

Dear Inna,

Thank you again for submitting your revised manuscript (EMBOJ-2024-118145R) to The EMBO Journal, and for addressing the most important of our editorial requests. Your revised manuscript has now been seen by the three original referees who had previously assessed the initial version of your manuscript, and we have received the complete set of their comments, which you can find below.

I am very pleased to say that all three reviewers acknowledge that the revised manuscript is significantly improved and their previously raised concerns successfully addressed, and they are now all supportive of publication of the manuscript in The EMBO Journal. There are only two minor remaining concerns/suggestions regarding the title of the manuscript (ref. #2) and the presentation of molecular weight markers in the blots (ref. #3), which I kindly ask you to address in a final version of your manuscript. Please include a description of any changes in a brief point-by-point response letter.

There are also a few editorial requests/formatting changes that we need from you to address in the final version of your manuscript, before we can proceed with formal acceptance of the manuscript for publication in The EMBO Journal. Please briefly describe how they are addressed in your point-by-point letter.

- Please note that the same funding information should be provided both in the online manuscript tracking system (eJP) and in the Acknowledgements section of the revised manuscript; currently, there are some inconsistencies between the two, namely missing info in eJP: START-Program, Faculty of Medicine of RWTH Aachen university (Az. 57/22); the fellowship of OvGU; there is a mis-match for the grant number for Wilhelm Sander-Stiftung (in ms: 2017.008.02 vs. in eJP: 2007008.02).

- Please change the heading "Summary" to "Abstract".

- Please note that a maximum of 5 keywords can be listed after the Abstract (you currently list 9).

- Thank you for providing the databases and identifiers of your deposited datasets in the Data availability statement of your manuscript. Could you please also include the permanent, specific URLs (links) to the datasets?

- The author contributions statement should be removed from the manuscript file. Instead, we use CRediT to specify the contributions of each author in the journal submission system. Please feel free to use the free text box to provide more detailed descriptions during submission. See also our guide to authors for more information: <https://www.embopress.org/page/journal/14602075/authorguide#authorshipguidelines>.

- We noticed that figure callouts for Fig. 2A-C, and 3A-D are missing. Please make sure that all Figure panels are called out, and that all callouts are listed sequentially in your revised manuscript.

- Please choose the positive response ("Yes") from the drop-down menu in your Author checklist (in the last section "Data Availability") regarding the first question of deposition of primary datasets, and indicate in the last column that the information is available in the Data availability statement.

- Regarding Expanded View (EV) Dataset legends: source file names, titles, legends, and manuscript callouts all need to be updated to "Dataset EV1-EV2" instead of Supplementary Tables 1 and 2. Their legends should be uploaded as a separate tab/sheet in each Excel file.

- Appendix Tables need to be renamed to "Appendix Table S1-S3" (instead of Table S1-S3) and included in the Table of Contents of the Appendix PDF file. All callouts throughout the manuscript should be corrected to "Appendix Figure Sx" (in some cases, the word "Figure(s)" is missing). Please also add callouts for "Appendix Table Sx". Please use the heading "Appendix methods" for the protocols and scripts in your Appendix, and also add an appropriate callout in your manuscript. Finally, please remove the line numbering from the final version of your Appendix file, as this file will be published online without further modifications.

- Our journal encourages inclusion of data citations in the reference list to directly cite datasets that were obtained from public databases. Data citations in the article text are distinct from normal bibliographical citations and should directly link to the database records from which the data can be accessed. In the main text, data citations are formatted as follows: "Data ref: Smith et al, 2001" or "Data ref: NCBI Sequence Read Archive PRJNA342805, 2017". In the Reference list, data citations must be labeled with "[DATASET]". A data reference must provide the database name, accession number/identifiers, and a resolvable link to the landing page from which the data can be accessed at the end of the reference. Further instructions are available at: <https://www.embopress.org/page/journal/14602075/authorguide#referencesformat>.

- During our standard Figure checks, we detected possible image/structure re-use in two of your Figures:
1. In Figure 1H: the cell "6721" (brightfield, AVF, PI channels) appears twice.

2. In Appendix Figure S7: The structures in the left and right panels look to be the same.

We kindly request you to double-check these two Figures and correct them if necessary, or -if the re-use is intentional- describe it explicitly in the respective Figure legends.

- During our routine pre-acceptance checks, our data editors have raised the following queries regarding figures, data, and legends. Please make sure that all requests below are completely addressed in the final version of your manuscript:

1. Please provide the exact p values in the legends of Figures 1D-G; 8A-C; EV1 C-H; EV2 A-C; EV9 A-C.

2. Please note that the box plots need to be defined in terms of minima, maxima, centre, bounds of box and whiskers, and percentile in the legends of Figures 4B-E; 5A-D; EV6 A-D, EV7A.

3. Please note that information related to "n" is missing in the legends of Figures EV6 C, D; EV7A.

- The Source Data (SD) for Fig. 4 need to be zipped and labeled as "Figure 4 Source Data" (all SD folders need to be zipped, not uploaded as Excel files). The Source Data for Fig. 1H-I (not labeled in the zip folder for Fig. 1 and not clarified in the SD checklist) seem to be missing. Please upload these Data or clarify in your SD checklist.

- The manuscript section order should be corrected as follows: Title page - Abstract & Keywords - Introduction - Results - Discussion - Methods - Data Availability - Acknowledgements - Disclosure and Competing Interests Statement - References - Figure Legends - main Table(s) (if there are any) - Expanded View Figure Legends.

- Please note that EMBO press papers are accompanied online by:

A) a short (2 sentences) summary of the findings and their significance,

B) 2-5 short bullet points highlighting the key results, and

C) a synopsis image in .jpg or .png format that is exactly 550 pixels wide and 300-600 pixels high (the height is variable). Please note that the text needs to be legible at the final size.

Please upload this information along with your revised manuscript (the text for A and B should be provided in a separate Word file).

Please also note that as part of the EMBO publications' Transparent Editorial Process, The EMBO Journal publishes online a Peer Review File along with each accepted manuscript. This File will be published in conjunction with your paper and will include the referee reports, your point-by-point response and all pertinent correspondence relating to the manuscript. You can opt out of this by letting the editorial office know (contact@embojournal.org). If you do opt out, the Peer Review File link will point to the following statement: "No Peer Review File is available with this article, as the authors have chosen not to make the review process public in this case."

We look forward to seeing a final version of your manuscript as soon as possible. Please let us know if you have any questions and use this link to submit your revision: <https://emboj.msubmit.net/cgi-bin/main.plex>.

Best wishes,

Ioannis

Referee #1:

Authors have comprehensively addressed all the points I raised.

Referee #2:

The authors have done a good job in responding to my comments and I believe the manuscript is suitable for publication ... I would however suggest a change in the title ... in my opinion it would be better to highlight RIPK1 and the results than the

techniques. I feel it would attract more readers ... but it is ultimately the authors' choice.

Referee #3:

I feel the authors have done an enormous amount of work and this study meets the standard I'd expect of EMBO J. The only remaining concern I hold is that the authors have estimated protein MW in the new blots but should have used marker positions instead. This should be corrected before acceptance.

Point-to-Point Response

Referee #1:

Authors have comprehensively addressed all the points I raised.

Answer

We would like to thank reviewer 1 for the time spend with our manuscript and valuable comments. They were of a tremendous help. We are really grateful.

Referee #2:

The authors have done a good job in responding to my comments and I believe the manuscript is suitable for publication ... I would however suggest a change in the title ... in my opinion it would be better to highlight RIPK1 and the results than the techniques. I feel it would attract more readers ... but it is ultimately the authors' choice.

Answer

We would like to thank reviewer 2 for the time and very important critical but fair comments. They were very helpful and allowed us to improve our manuscript.

We are grateful for the suggestion to change the title of the manuscript, which has been done accordingly.

Referee #3:

I feel the authors have done an enormous amount of work and this study meets the standard I'd expect of EMBO J. The only remaining concern I hold is that the authors have estimated protein MW in the new blots but should have used marker positions instead. This should be corrected before acceptance.

Answer

We would like to thank reviewer 3 for the time spend with our manuscript and valuable comments. We highly appreciated them and they helped us enourmously.

The question with the molecular markers was addressed.

All editorial and formatting issues were resolved by the authors.

Dear Inna,

Congratulations on an excellent work! I am very pleased to inform you that your manuscript has now been accepted for publication in The EMBO Journal. Thank you for comprehensively addressing the referees' comments as well as the editorial and formatting requests.

If you have any questions, please do not hesitate to contact the Editorial Office. Thank you for your contribution to The EMBO Journal. Working with you has been a pleasure!

Best wishes,

Ioannis
